Manuscript prepared for Geosci. Model Dev. Discuss.
with version 2014/09/16 7.15 Copernicus papers of the LATEX class copernicus.cls.
Date: 7 July 2016

# Evaluation of NorESM-OC (versions 1 and 1.2), the ocean carbon-cycle stand-alone configuration of the Norwegian Earth System Model (NorESM1)

**Jörg Schwinger**[1], **Nadine Goris**[1], **Jerry F. Tjiputra**[1], **Iris Kriest**[2], **Mats Bentsen**[1], **Ingo Bethke**[1], **Mehmet Ilicak**[1], **Karen M. Assmann**[1,a], **and Christoph Heinze**[3,1]

[1]Uni Research Climate, Bjerknes Centre for Climate Research, Bergen, Norway
[2]GEOMAR Helmholtz-Zentrum für Ozeanforschung, Kiel, Germany
[3]Geophysical Institute, University of Bergen, Bjerknes Centre for Climate Research, Bergen, Norway
[a]now at: Department of Marine Sciences, University of Gothenburg, Gothenburg, Sweden

Correspondence to: J. Schwinger (jorg.schwinger@uni.no)

## Abstract

Idealised and hindcast simulations performed with the stand-alone ocean carbon-cycle configuration of the Norwegian Earth System Model (NorESM-OC) are described and evaluated. We present simulation results of three different model configurations (two different model versions at different grid resolutions) using two different atmospheric forcing data sets. Model version NorESM-OC1 corresponds to the version that is included in the fully coupled model NorESM-ME1, which participated in CMIP5. The main update between NorESM-OC1 and NorESM-OC1.2 is the addition of two new options for the treatment of sinking particles. We find that using a constant sinking speed, which has been the standard in NorESM's ocean carbon cycle module HAMOCC (HAMburg Ocean Carbon Cycle model) does not transport enough particulate organic carbon (POC) into the deep ocean below approximately 2000 m depth. The two newly implemented parameterisations, a particle aggregation scheme with prognostic sinking speed, and a simpler scheme that uses a linear increase of the sinking speed with depth, provide better agreement with observed POC fluxes. Additionally, reduced deep ocean biases of oxygen and remineralised phosphate indicate a better performance of the new parameterisations. For model version 1.2, a re-tuning of the ecosystem parameterisation has been performed, which (i) reduces previously too high primary production in high latitudes, (ii) consequently improves model results for surface nutrients, and (iii) reduces alkalinity and dissolved inorganic carbon biases at low latitudes. We use hindcast simulations with prescribed observed and constant (pre-industrial) atmospheric $CO_2$ concentrations to derive the past and contemporary ocean carbon sink. For the period 1990–1999 we find an average ocean carbon uptake ranging from 2.01 to 2.58 Pg C yr$^{-1}$ depending on model version, grid resolution and atmospheric forcing data set.

## 1 Introduction

Earth system models (ESMs) have been developed to take into account feedbacks between the physical climate and biogeochemical processes in projections of climate change (Bretherton, 1985; Flato, 2011). However, due to the complexity of feedback processes, it
can prove useful to run one or several submodels of an ESM independently by using prescribed data at the boundary between submodel domains, e.g. by using prescribed atmospheric conditions at the air–sea boundary to force the ocean and ice models of an ESM. Such "stand-alone" model configurations are useful to conduct idealised experiments, to perform hindcast simulations in which boundary conditions reflect the observed variability
and trends, or to save computer time in cases where certain feedbacks are not expected to be important. Idealised experiments include set-ups where (purposefully manipulated) atmospheric output from fully coupled ESM runs is used to force the stand-alone configuration of the same or another model.

Here, we describe and evaluate the stand-alone ocean carbon-cycle configuration of the
15 Norwegian Earth System Model (NorESM-OC). The Norwegian Earth System Model version 1 (NorESM1, Bentsen et al., 2013; Tjiputra et al., 2013) is derived from the Community Earth System Model version 1 (CESM1, Gent et al., 2011; Lindsay et al., 2014), using the same sea-ice and land models as well as the same coupler, but a modified atmospheric model (CAM4-Oslo, Kirkevåg et al., 2013), and a different ocean model. NorESM's physical
ocean component originates from the Miami Isopycnic Coordinate Ocean Model (MICOM; Bleck and Smith, 1990; Bleck et al., 1992) but has been updated with modified numerics and physics as described in Bentsen et al. (2013). This model has been extended by Assmann et al. (2010) to include the HAMburg Ocean Carbon Cycle model version 5.1 (HAMOCC5.1, Maier-Reimer, 1993; Maier-Reimer et al., 2005). In the NorESM-OC configuration MICOM-
HAMOCC is coupled to CESM's sea-ice model version 4 (CICE4, Holland et al., 2012) and a "data atmosphere" which provides atmospheric forcing fields to the ocean and sea-ice components.

The NorESM-OC model configuration allows to study the global ocean carbon cycle and related biogeochemical cycles (phosphorus, nitrogen, silicate, iron, and oxygen) in idealised or hindcast simulations. We pursue two main objectives with this setup. First, it provides a simplified and computationally (relatively) inexpensive framework for the implementation and testing of new or updated model parameterisations. For example, we present the implementation and evaluation of new parameterisations for the sinking of particulate organic carbon (POC) in this work. In a fully coupled model setup atmospheric $CO_2$ concentration is sensitive to the parameterisation of the biological pump (e.g. Marinov et al., 2006; Kwon et al., 2009), and eventually we aim to evaluate this sensitivity and associated feedbacks in the fully coupled version of NorESM. It is convenient, however, to perform a first evaluation without any feedback between biogeochemistry and physical climate (i.e., ocean circulation is unchanged in all test cases).

Second, hindcast simulations can provide insight into the response of biogeochemical cycles to observed variability and trends in atmospheric forcing. Of particular interest is the estimation of past and contemporary anthropogenic carbon uptake by the oceans using meteorological reanalysis data to force the ocean model. Such information is needed to better constrain the Earth system's contemporary carbon budget, an effort undertaken by e.g. the Global Carbon Project (GCP, Le Quéré et al., 2015). Hindcast simulations performed with a first version of this model configuration (NorESM-OC1) contributed to the annual update of this carbon budget for the years 2011 to 2013. NorESM-OC1 corresponds to the version included in the fully coupled NorESM1-ME that is described in Tjiputra et al. (2013), and that participated in the phase 5 of the Coupled Model Intercomparison Project (CMIP5, Taylor et al., 2012).

Although NorESM-OC is computationally less expensive than the fully coupled NorESM, computational constraints limit the time-scale, for which the model can be applied, to a few hundred to thousand years on current hardware. We note, that this limitation is mainly due to the physical ocean model and the costly transport of tracers. By using an efficient offline method (e.g., Khatiwala et al., 2005; Kriest et al., 2012) it would be possible to apply HAMOCC on time-scales at least one order of magnitude longer, as long as all relevant

external inputs are provided (e.g., fluxes due to continental weathering). Similar versions of HAMOCC have been applied on time-scales of 50 000 years (e.g., Heinze et al., 2003, who use an offline setup with annually averaged ocean circulation fields).

In this manuscript we describe NorESM-OC1 next to an updated version of the model sys-
tem (NorESM-OC1.2), which is configured on a numerically more efficient grid at either $1°$ or $2°$ nominal resolution, includes additional parameterisations for the sinking of POC, and features several updates and improvements as described in Sect. 2. Spin-up followed by hindcast simulations forced by two slightly different forcing data sets have been performed for the different model versions and configurations. We evaluate the model results against
available observation based climatologies (Sects. 3.1 to 3.3), and we calculate ocean car-
bon sink estimates and anthropogenic carbon storage (Sect. 3.4). The newly implemented particle sinking schemes are evaluated using observation based estimates of global POC fluxes and of remineralisation as well as sediment trap data (Sect. 3.5). We conclude by discussing the current status of the model system and lines of further development.

## 15  2  Model description and configuration

The components of NorESM discussed here have been described and evaluated in papers published previously in this special section (Bentsen et al., 2013; Tjiputra et al., 2013) and elsewhere (Danabasoglu et al., 2014; Griffies et al., 2014; Downes et al., 2015; Farneti et al., 2015). Therefore, we only give a brief description of the NorESM-OC model compo-
20 nents and focus on documenting the specific model configurations and the updates made
for NorESM-OC1.2.

### 2.1  The physical ocean model MICOM

The main benefit of an isopycnal model is the good control on the diapycnal mixing (less numerical diffusion across isopycnic surfaces) that helps to preserve water masses during
long model integrations. This is of particular interest for a coupled physical-biogeochemical
model, since sharp gradients in tracer concentrations between water masses (e.g. in the

thermocline region) can potentially be better represented by the model. The isopycnic framework provides a smooth representation of bottom topography, and therefore overflow of water masses is modelled without numerical obstacles (although mixing at steep slopes must be parameterised carefully).

As mentioned above, the model is based on MICOM as described by Bleck and Smith (1990) and Bleck et al. (1992). Key aspects retained from this model version are a mass conserving formulation (non-Boussinesq), Arakawa C-grid discretisation, leap-frog and forward-backward time stepping for the baroclinic and barotropic mode, respectively, and a potential vorticity/enstrophy conserving scheme for the momentum equation. Different from the original MICOM, we use the incremental remapping algorithm of Dukowicz and Baumgardner (2000) for transport of layer thickness, potential temperature, salinity, and tracers. The second order accurate algorithm is expressed in flux form and thus conserves mass by construction. Furthermore, it guarantees monotonicity of tracers (i.e., the scheme does not create new minima or maxima) for any velocity field.

Originally, MICOM had a single bulk surface mixed layer while in NorESM the mixed layer is divided into two model layers with freely evolving density and equal thickness when the mixed layer is shallower than 20 m. The uppermost layer is limited to 10 m when the mixed layer is deeper than 20 m. The main reason for this is to allow for a faster ocean surface response to surface fluxes. We achieve reduced mixed layer depth biases (compared to original MICOM) by using a turbulent kinetic energy (TKE) model based on Oberhuber (1993), extended with a parameterisation of mixed layer restratification by submesoscale eddies (Fox-Kemper et al., 2008). To improve the representation of water masses in weakly stratified high latitude halocline, the static stability of the uppermost layers is measured by in-situ density jumps across layer interfaces, thus allowing for layers that are unstable with respect to potential density referenced at 2000 dbar to exist. The diagnostic version of the eddy closure of Eden and Greatbatch (2008) as implemented by Eden et al. (2009) is used to parameterise the thickness and isopycnal eddy diffusivity. Further, to reduce sea surface salinity and stratification biases at high latitudes, salt released during freezing of sea ice can be distributed below the mixed layer.

The ocean component exchanges no mass with the other components of NorESM. Thus, freshwater fluxes are converted to virtual salt and tracer fluxes before they are applied to the ocean.

## 2.2 The sea ice model CICE

NorESM-OC employs the CESM sea ice component, which is based on version 4 of the Los Alamos National Laboratory sea ice model (CICE4, Hunke and Lipscomb, 2008). Important modifications of this model that are utilised in CESM and NorESM1 are the delta-Eddington short-wave radiation transfer as well as melt pond and aerosol parameterisations, all described by Holland et al. (2012). The sea-ice model shares the same horizontal grid with the ocean component of NorESM.

## 2.3 The marine biogeochemistry model HAMOCC5.1

The HAMburg Ocean Carbon Cycle (HAMOCC) model is based on the original work of Maier-Reimer (1993) and subsequent refinements (Maier-Reimer et al., 2005). The model simulates the marine biogeochemical cycles of carbon, nitrogen, phosphorus, silicate, iron, and oxygen, including the fluxes of these elements across the air–sea interface. HAMOCC5.1 was coupled with the isopycnic MICOM model by Assmann et al. (2010). We describe a version of the code that evolved from this point and note that the original model has also been further developed by the biogeochemistry group at Max Planck Institute in Hamburg (Ilyina et al., 2013). The model described by Assmann et al. (2010) was the starting point for the development of the ocean biogeochemistry component of NorESM1. Main differences between this model and NorESM-OC1 are updates of mixed layer and eddy diffusivity parameterisations, the use of CICE as ice model, and the details of atmospheric forcing, which followed Bentsen and Drange (2000), wheras NorESM-OC uses the approach of Large and Yeager (2004). The version of HAMOCC employed by Assmann et al. (2010) is very similar to the one in NorESM-OC1 with the notable difference of an update of the carbon chemistry scheme (see below).

The HAMOCC code is embedded in MICOM, and hence runs at the same spatial and temporal resolution as the ocean model. The model has 19 prognostic tracers (Table 1), all of which are advected and diffused with the ocean circulation provided by MICOM. Two additional tracers are needed if the particle aggregation scheme (prognostic sinking speed, see Sect. 2.3.3) is switched on. There are also three "preformed" tracers for oxygen, phosphate, and alkalinity, which are set to the corresponding concentration values in the surface mixed layer at each time-step, and are otherwise passively advected. Preformed tracers have been introduced in version 1.2 to enhance the diagnostic capabilities of the model.

When HAMOCC was implemented as the biogeochemistry module of MICOM, the biogeochemical tracers were only defined at one of the two time-levels of the leap-frog time stepping scheme to increase computational efficiency (Assmann et al., 2010). To prevent amplification of numerical noise, the MICOM leap frog time-stepping includes a time-smoothing applied to the mid time-level of temperature, salinity and layer thickness fields at each time step, i.e. $x_{\mathrm{mid}} = w_1 x_{\mathrm{mid}} + w_2 (x_{\mathrm{old}} + x_{\mathrm{new}})$ with $w_1 = 0.875$ and $w_2 = 0.0625$. Since the physical fields undergo this time smoothing, but in NorESM-OC1 the biogeochemical tracers do not, there is an inconsistency between physical fields and biogeochemical tracers introduced by this simplification, which results in a non-conservation or tracer mass. This non-conservation is accounted for in model version 1 by a correction factor applied after the time-smoothing to the biogeochemical tracer fields. Although simulation results were not significantly affected on integration time scales of $\approx 1000$ years we decided to re-implement the tracer transport scheme to make it fully consistent with the physical fields in version 1.2 by defining the tracer fields on two time-levels and apply the same time-smoothing procedure as for the physical fields. We anticipate that this improvement will become important on integration time scales of 10 000 years or longer.

## 2.3.1 Air–sea gas exchange and inorganic carbon chemistry

HAMOCC calculates the exchange of various gases through the air–sea interface. This exchange is determined by three components: The gas solubility in seawater, the gas transfer rate, and the gradient of the gas partial pressure between the atmosphere and the

ocean surface. The solubilities of $O_2$ and $N_2$ in seawater as functions of surface ocean temperature and salinity are taken from Weiss (1970), and solubilities of $CO_2$ and $N_2O$ from Weiss and Price (1980). Solubilities of sulfur hexaflouride ($SF_6$) and the two chlorofluorocarbons CFC-11 and CFC-12 are calculated according to Warner and Weiss (1985) and Bullister et al. (2002). The gas transfer rates are computed using the empirical relationship $k = aU_{10}^2(660/Sc)^{1/2}$ derived by Wanninkhof (1992), where $U_{10}$ is the surface wind speed at 10 m reference height and $a$ is a gas independent constant. The Schmidt number *Sc* is the kinematic viscosity divided by the diffusivity of the respective gas component, and 660 is the Schmidt number of seawater at 20 °C. Schmidt numbers for $CO_2$ and CFCs have been taken from Wanninkhof (1992). Since the polynomial used there to fit the experimental data displays non-physical behaviour outside the validity range of the fit for temperatures $> 30$ °C (very small and negative Schmidt numbers), we use the re-fitted temperature dependence recommended by Gröger and Mikolajewicz (2011) in model version 1.2. We note that the updated air-sea gas exchange formulation provided by Wanninkhof (2014) also includes re-fitted Schmidt numbers, avoiding problems at high SST. The Schmidt number for $O_2$ is taken from Keeling et al. (1998), and the same value is assumed also for $N_2$ and $N_2O$. The model has no gas exchange through ice-covered surface areas of a grid cell, and for partly ice-covered grid cells the gas exchange flux is scaled to the ice free fraction of the cell.

Compared to the original HAMOCC (Maier-Reimer et al., 2005) and to the version used by Assmann et al. (2010), our model includes a revised inorganic seawater carbon chemistry following Dickson et al. (2007). Total alkalinity (TA) is defined to include contributions of boric acid, phosphoric acid, hydrogen sulphate, hydrofluoric acid, and silicic acid,

$$
\begin{aligned}
\text{TA} =& [\text{HCO}_3^-] + 2[\text{CO}_3^{2-}] + [\text{B(OH)}_4^-] + [\text{OH}^-] + [\text{HPO}_4^{2-}] + 2[\text{PO}_4^{3-}] + [\text{SiO(OH)}_3)^-] \\
& - [\text{H}^+] - [\text{HSO}_4^-] - [\text{HF}] - [\text{H}_3\text{PO}_4].
\end{aligned}
\tag{1}
$$

We use an iterative carbonate alkalinity correction method (Follows et al., 2006; Munhoven, 2013) to calculate the oceanic partial pressure of $CO_2$ ($pCO_2$) prognostically as a function of temperature, salinity, dissolved inorganic carbon (DIC), and TA. Using an initial guess for the hydrogen ion concentration $[\text{H}^+]_0$, all non-carbonate contributions to TA are calculated

according to Dickson et al. (2007) and added to or subtracted from (1) in order to obtain an initial guess for carbonate alkalinity $CA_0$ ($CA = [HCO_3^-] + 2[CO_3^{2-}]$). Given CA and DIC and the equilibrium constants of the carbonate system $K_1$ and $K_2$, the corresponding hydrogen ion concentration can be calculated:

$$[H^+] = \frac{K_1}{2\,CA} \left( DIC - CA + \sqrt{(DIC - CA)^2 + 4\,CA\,K_2/K_1(2\,DIC - CA)} \right). \tag{2}$$

The hydrogen ion concentration $[H^+]_1$ thus obtained is used to re-iterate these calculations until convergence is reached. Here, we use $[H^+]$ from the previous time step as an initial guess for the calculation of CA, and stop iterations once the relative change ($([H^+]_{i+1} - [H^+]_i)/[H^+]_{i+1}$) becomes smaller than $\epsilon = 5 \times 10^{-5}$. Only at the first time step of integration we use $[H^+]_0 = 10^{-8}$ mol kg$^{-1}$. As pointed out by Follows et al. (2006), the changes in $[H^+]$ from one time step to the next are small, and one or a few iterations of this procedure usually suffice. The effect of pressure on the various dissociation constants is calculated according to Millero (1995).

### 2.3.2 Ecosystem model

HAMOCC5.1 employs an NPZD-type (nutrient, phytoplankton, zooplankton and detritus) ecosystem model, extended to include dissolved organic carbon (DOC). The ecosystem model was initially implemented by Six and Maier-Reimer (1996). The nutrient compartment is represented by three macronutrients (phosphate, nitrate, silicate), and one micronutrient (dissolved iron). A constant Redfield ratio of $P : C : N : O_2 = 1 : 122 : 16 : -172$ following Takahashi et al. (1985) is used. That is, the ratios of carbon to phosphorus and nitrogen to phosphorus in organic matter are $R_{C:P} = 122$ and $R_{N:P} = 16$, respectively, and $R_{O_2:P} = 172$ moles of oxygen are consumed during remineralisation of 1 mole phosphorus.

The phytoplankton growth rate $J(I,T)$ follows the light-saturation curve formulation of Smith (1936)

$$J(I,T) = \frac{\alpha \sigma I(z) f(T)}{\sqrt{(\alpha \sigma I(z))^2 + f(T)^2}} \tag{3}$$

with temperature dependence parameterised according to Eppley (1972), $f(T) = \mu_{\text{phy}} 1.066^T$. Here, $\alpha = 0.02$ is the initial slope of the photosynthesis vs. irradiance curve, $\sigma = 0.4$ is the fraction of photosynthetically active radiation, and $\mu_{\text{phy}} = 0.6\,\text{d}^{-1}$. The available light is calculated based on incoming solar radiation prescribed through the atmo-
5 spheric forcing ($I_0$) and attenuation through seawater ($k_\text{w}$) and chlorophyll ($k_{\text{chl}}$)

$$I(z) = I_0\,e^{-(k_\text{w} + k_{\text{chl}}\text{Chl})z}. \tag{4}$$

Chlorophyll (Chl) is calculated from phytoplankton mass assuming a constant carbon to chlorophyll ratio $R_{\text{C:Chl}} = 60$.

A disadvantage of an isopycnic model is the low vertical resolution in weakly stratified
regions. The mixed layer in MICOM was represented by only one model layer in Assmann et al. (2010). For NorESM a second model layer has been added to improve the simulation of mixed layer processes. The top model layer varies in thickness between 5 and 10 m while the second model layer represents the rest of the mixed layer. Consequently, for deep mixed layers, the second model layer will often have a vertical extent of 100 m and more,
which still results in a rather poor resolution for the simulation of biological processes. We therefore decided to re-implement Eq. (4) for NorESM-OC1.2 by calculating the average light availability $\overline{I}$ (instead of using the light availability at the centre of layers), which reads for layer $i$:

$$\overline{I}_i = I_{0,i} \frac{1}{(k_\text{w} + k_{\text{chl}}\text{Chl}_i)\Delta z_i} \left(1 - e^{-(k_\text{w} + k_{\text{chl}}\text{Chl}_i)\Delta z_i}\right) \tag{5}$$

In addition to light limitation, phytoplankton growth is co-limited by availability of phosphate, nitrate, and dissolved iron. The nutrient limitation is expressed through a Monod function $f_{\text{nut}} = X/(K + X)$, where $K$ is the half-saturation constant for nutrient uptake ($K = 4 \times 10^{-8}\ \text{kmol P m}^{-3}$), and

$$X = \min\left(\text{PO}_4, \frac{\text{NO}_3}{R_{\text{N:P}}}, \frac{\text{Fe}}{R_{\text{Fe:P}}}\right). \tag{6}$$

$R_{\text{N:P}} = 16$ and $R_{\text{Fe:P}} = 366 \times 10^{-6}$ are the (constant) nitrogen to phosphorus and iron to phosphorus ratios for organic matter used in the model. Aerial dust deposition to the surface ocean is prescribed based on Mahowald et al. (2006). A fraction of the dust deposition (1 %) is assumed to be iron, and part of it is immediately dissolved and available for biological production. To mimic the process of complexation with ligands, iron concentration is relaxed towards a value of 0.6 $\mu$mol m$^{-3}$ when modelled iron is larger than this value. We note that this parameterisation of the iron cycle is rather simplistic. The spatial pattern of iron in the surface mainly reflects the aeolian input and upwelling of iron in the Southern Ocean. At depth, iron concentration is determined by accumulation of remineralised iron and the assumed complexation. Therefore, the iron concentration approaches a constant value of 0.6 $\mu$mol m$^{-3}$ at depth larger than approximately 700 m. We find that the resulting iron limitation in our model is rather weak, and given these limitations we will not focus on the iron cycle in this manuscript.

In nitrate-limited regions, i.e. if $[\text{NO}_3] < R_{\text{N:P}}[\text{PO}_4]$, the model assumes nitrogen fixation by cyanobacteria in the surface layer. The distributions of calcium carbonate and biogenic silica export productions depend on the availability of silicic acid, since it is implicitly assumed that diatoms out-compete other phytoplankton species when supply of silicic acid is sufficient. This is parameterised by assuming that a fraction $[\text{Si}]/(K_{\text{Si}} + [\text{Si}])$ of export production contains opal shells while the remaining fraction contains calcareous shells, where $K_{\text{Si}} = 1$ mmol Si m$^{-3}$ is the half-saturation constant for silicate uptake (Martin-Jézéquel et al., 2000). Phytoplankton loss is modeled by specific mortality and exudation rates as well as zooplankton consumption. DOC is produced by phytoplankton and zooplankton (through constant exudation and excretion rates) and is remineralised at a constant rate (whenever the required oxygen is available). The full set of differential equations defining the ecosystem can be found in Maier-Reimer et al. (2005).

Between model version 1 and 1.2 we performed a re-tuning of the ecosystem parameterisation (see Table 2 for old and new parameter values). The objective of this effort was to (i) reduce previously overestimated primary production in high latitudes, (ii) increase production in the low latitude oligotrophic gyres, and (iii) reduce a low latitude bias in alkalinity and

DIC found in model version 1. The tuning was not based on an objective procedure, but relied on the authors' experience and a series of "trial and error" simulations. Points (i) and (ii) were addressed by increasing zooplankton abundance (decreasing zooplankton mortality $\lambda_{\text{zoo}}$ and increasing zooplankton assimilation efficiency $\omega_{\text{zoo}}$), which helps to limit primary production in the high latitude "high-nutrient, low-chlorophyll" regions. Further, phytoplankton growth is stimulated mainly at low latitudes by reducing the half-saturation constant $K_{\text{PO}_4}$ and by increasing available nutrients in the surface by slightly increasing the remineralisation of detritus and DOC. The alkalinity and related DIC biases (iii) were addressed by reducing the surface concentration of silicate in order to increase $CaCO_3$ production. This was done by increasing the opal to phosphorus uptake ratio $R_{\text{Si:P}}$.

### 2.3.3 Particle export and sinking

The standard HAMOCC scheme for the treatment of sinking particles assumes a prescribed constant sinking speed for three classes of particles. Particulate organic carbon associated with dead phytoplankton and zooplankton is transported vertically at $5\,\text{m}\,\text{d}^{-1}$. As POC sinks, it is remineralised at a constant rate and according to oxygen availability. When oxygen concentration falls below a threshold of $0.5\,\mu\text{mol}\,\text{L}^{-1}$, POC is remineralised by denitrification. Without explicitly modeling this process, it is assumed that $2/3 \times R_{\text{O}_2:\text{P}} \approx 115$ moles of nitrate are consumed by denitrifying bacteria to remineralise an amount of detritus corresponding to one mole of phosphate (i.e., it is assumed that the oxygen from two moles of nitrate substitutes three moles of oxygen during denitrification). Particulate inorganic carbon (calcium carbonate, PIC) and opal shells (biogenic silica) sink with a fixed speed of $30\,\text{m}\,\text{d}^{-1}$. Biogenic silica is decomposed at depth with a constant dissolution rate while the dissolution of calcium carbonate shells depends on the saturation state with respect to $CaCO_3$ in the surrounding seawater. Non-remineralised particulate materials reaching the sea floor sediment undergo chemical reactions with the sediment pore waters and vertical advection within the sediment. The 12-layer sediment model (Heinze et al., 1999; Maier-Reimer et al., 2005) is primarily relevant for long-term simulations ($> 1000\,\text{years}$). Nevertheless, the sediment model was activated for all simulations presented here. The current

model version neither includes a parameterisation of weathering fluxes nor does it take into account the influx of carbon and nutrients through continental discharge. The material lost to the sediments is therefore not replaced by some mechanism in the model configurations presented here. The model drift caused by these losses to the sediment is small on the time-scales considered in this manuscript, particularly at the surface. Nevertheless, this simplification limits the applicability of NorESM-OC1 and 1.2 to time-scales of the order of 1000 years. Input of DIC and nutrients in particulate and dissolved forms through rivers is currently implemented into NorESM.

In NorESM-OC1.2 we provide two alternative options for the treatment of particle sinking. First, a scheme which calculates the sinking speed as a linear function of depth as $w_{POC} = \min(w_{min} + a\,z, w_{max})$ has been implemented. Here, $w_{min}$ is the minimum sinking speed found at $z = 0$, and $a$ is a constant describing the increase of speed with depth. The sinking speed can be limited by setting the parameter $w_{max}$. We refer to this scheme as WLIN in the following text, and we use $w_{min} = 7\,\mathrm{m\,d^{-1}}$, $w_{max} = 43\,\mathrm{m\,d^{-1}}$, and $a = 40/2500\,\mathrm{m\,d^{-1}m^{-1}}$ in this study. Note that for $w_{min} = 0$ and $w_{max} = \infty$, this parameterisation is equivalent to the widely used Martin-curve formulation (Martin et al., 1987; Kriest and Oschlies, 2008).

The second new option is a scheme with variable (prognostic) sinking speed which is calculated according to a size distribution of sinking particles. Since an explicit representation of sinking particles through a large number of discrete size classes would not be feasible in a large scale Earth system model due to computational constraints, we implemented the particle aggregation and sinking scheme devised by Kriest and Evans (1999) and Kriest (2002) as a cost effective alternative. This model (hereafter abbreviated KR02) assumes that phytoplankton and detritus form sinking aggregates and that the size distribution of aggregates obeys a power law formulation

$$n(d) = A\,d^{-\varepsilon}, \quad l < d < \infty, \tag{7}$$

where $d$ is the particle diameter, and $A$ and $\varepsilon$ are parameters of the distribution. The lower bound of diameters $l$ conceptually corresponds to the size of a single cell. Large values of $\varepsilon$ ( a steeper slope of the distribution) correspond to a larger number of slow sinking small

particles. By integration over the size range from $l$ to $\infty$ the total number of aggregates (NOS) is obtained

$$\text{NOS} = A \int\limits_{l}^{\infty} d^{-\varepsilon} \mathrm{d}d = A \frac{l^{(1-\varepsilon)}}{(\varepsilon - 1)}, \tag{8}$$

provided that $\varepsilon > 1$. The mass of an aggregate is described by $Cd^{\zeta}$ and consequently, the
total mass of aggregates reads

$$M = A\, C_l \frac{l^{(1-\varepsilon)}}{(\varepsilon - 1 - \zeta)} = \text{PHY} + \text{POC}, \tag{9}$$

where $C_l = Cl^{\zeta}$ is the mass of a single cell. If particles were spheres with constant density, then $\zeta = 3$. Since the mass of aggregates grows more slowly than the cube of their diameter, we adopt the value $\zeta = 1.62$ used in Kriest (2002). Finally, it is assumed that the sinking
speed of aggregates depends on the diameter according to $w(d) = Bd^{\eta}$ with $w_l = Bl^{\eta}$ being the sinking speed of a single cell. The total phytoplankton and detritus mass $M = \text{PHY} + \text{POC}$ and the number of aggregates NOS are the prognostic variables of the KR02-scheme as implemented in HAMOCC. Given $M$ and NOS, the parameters $\epsilon$ and $A$ of the particle size distribution can be determined, and from these the average sinking speed for
mass and numbers are obtained.

NOS is treated like other particulate tracers in the model, i.e. NOS is advected and diffused by ocean circulation and treated in HAMOCC's sinking scheme (see below) using the average sinking speed for particle numbers as given in Kriest (2002). Additionally, aggregation of particles decreases NOS, while photosynthesis, egestion of fecal pellets, and
zooplankton mortality increase NOS. The size distribution is affected by these process as follows. Sinking removes preferentially the large particles and leaves behind the smaller ones, thereby steepening the slope $\varepsilon$ of the size spectrum in surface layers. Aggregation "flattens" the slope of the size spectrum, because it reduces the number of particles, but not mass. These processes are parameterised as in scenario "pSAM" of Kriest (2002), but with

stickiness set to 0.25, the factor for shear collisions set to 0.75 d$^{-1}$, and a maximum particle size for size dependent aggregation and sinking of 0.5 cm (see Table 3 for a summary of parameter values). We assume that all other biogeochemical processes do not impact $\varepsilon$ (e.g., photosynthesis increases $M$ and NOS proportionally, such that the slope of the size

distribution is not affected), the exception being zooplankton mortality, which flattens the size spectrum through the addition of (large) zooplankton carcasses.

An implicit scheme is used to redistribute mass (and the number of particles in the case of KR02) through the water column

$$C_i^{t+dt} = C_i^t + \frac{dt}{\Delta z_i}(w_{i-1}C_{i-1}^{t+dt} - w_i C_i^{t+dt}), \tag{10}$$

where d$t$ is the time-step length, $\Delta z_i$ the thickness of layer $i$, $C$ stands for the concentration of POC, PIC, opal or NOS, and $w_i$ is the corresponding sinking speed at layer $i$. This scheme is used with prognostic or prescribed sinking speed options. Note that in the KR02 scheme $w$ is a function of time and would be evaluated at time-level $t + dt$ in a fully implicit discretisation. This, however, is practically impossible and we resort to the common simpli-

fication of evaluating $w$ at the old time-level ("lagging the coefficients", Anderson, 1995) for the purpose of solving Eq. (10). We further note that PIC and opal are sinking at a fixed prescribed rate of 30 md$^{-1}$ if the standard or WLIN scheme is used, but are assumed to sink as a component of the phytoplankton-detritus aggregates in the KR02 scheme (i.e. at the prognostic sinking speed calculated by the scheme). For a more detailed discussion of the

KR02 scheme in comparison to the assumption of constant or linearly increasing sinking speed, we refer the reader to Kriest and Oschlies (2008).

## 2.4 Model configuration

We discuss three different model grid configurations, one for NorESM-OC1, which runs on a displaced pole grid with 1.125° nominal resolution and with grid singularities over Antarc-

tica and Greenland. NorESM-OC1.2 has been set up on a numerically more efficient tripolar grid in 1 and 2° nominal resolution. The tripolar grid has its singularities at the South Pole,

in Canada, and in Siberia. The nominal resolutions given here indicate the zonal resolution of the grid while the latitudinal resolution is finer and variable. For the displaced pole grid of NorESM-OC1, the latitudinal grid spacing is $0.27°$ at the equator gradually increasing to $0.54°$ at high southern latitudes. The tripolar grid of NorESM-OC1.2 is optimised for isotropy

of the grid at high latitudes, and the latitudinal spacing varies from $0.25°$ ($0.5°$) at the equator to $0.17°$ ($0.35°$) at high southern latitudes for the $1°$ ($2°$) nominal resolution. Note that over the northern hemisphere both grid types are distorted to accommodate the displaced pole over Greenland or the dual pole structure over Canada and Siberia. Due to the more evenly distributed grid spacing of the tripolar grid in the northern hemisphere, the time-step can be

increased from 1800 to 3200 s, and a timestep of 5400 s can be used for the $2°$ configuration. We use the abbreviations Mv1, Mv1.2, and Lv1.2 for the three versions/configurations, where "M" and "L" refer to medium ($1°$) and low ($2°$) resolution, respectively, followed by the version number of the model. All variants of NorESM-OC discussed here are configured with 51 isopycnic layers referenced at 2000 db and potential densities ranging from 28.202

to 37.800 kg m$^{-3}$. The reference pressure at 2000 dbar provides reasonable neutrality of model layers in large regions of the ocean (McDougall and Jackett, 2005).

### 2.4.1 Forcing

The atmospheric forcing required to run NorESM-OC comprises air temperature, specific humidity and wind at 10 m reference height as well as sea level pressure, precipitation

and incident short wave radiation fields. Two variants of the data sets developed for the Coordinated Ocean-ice Reference Experiments (COREs, Griffies et al., 2009), the CORE normal year forcing (CORE-NYF, Large and Yeager, 2004) and the CORE interannual forcing (CORE-IAF, Large and Yeager, 2009) can be used to provide the atmospheric forcing for NorESM-OC. Both data sets are based on the NCEP reanalysis (Kalnay et al., 1996)

and a number of observational data sets. Corrections are applied to the NCEP surface air temperature, wind, and humidity to remove known biases while precipitation and radiation fields are derived entirely from satellite observations (see Large and Yeager, 2004, 2009 for details). The near surface atmospheric state has a 6 hourly frequency, while the radiation

and precipitation fields are daily and monthly means, respectively. The normal year forcing has been constructed to represent a climatological mean year with a smooth transition between the end and start of the year (Large and Yeager, 2004). This forcing can be applied repeatedly for as many years as needed to force an ocean model without imposing inter-annual variability or discontinuities. We use this forcing data set to spin up our model. The CORE-IAF consists of the corrected NCEP data for the years 1948–2009. Radiation and precipitation data sets are daily and monthly climatological means before 1984 and 1979, respectively, and vary interannually for the time period thereafter.

Since we wish to run hindcast simulations up to present date, but the CORE-IAF only covers the years 1948 to 2009, we devised an alternative data set, which is identical to the CORE-IAF for surface air temperature, wind, humidity, and density (i.e., the same corrections as for the CORE-IAF are applied to the NCEP reanalysis data), but the radiation and precipitation fields are taken from the NCEP reanalysis without any corrections. NCEP surface air temperature and specific humidity are re-referenced from 2 to 10 m reference height following the same procedure as used for the CORE forcings (S. Yeager, personal communication, 2012). We refer to this forcing data set as NCEP-C-IAF in the following text.

Continental freshwater discharge is based on the climatological annual mean data set described by Dai and Trenberth (2002). It has been modified to include a 0.073 Sv contribution from Antarctica as estimated by Large and Yeager (2009). This freshwater discharge is distributed evenly along the coast of Antarctica as a liquid freshwater flux. All runoff fluxes are mapped to the ocean grid and smeared out over ocean grid cells within 300 km of each discharge point to account for unresolved mixing processes.

In order to stabilise the model solution, we apply salinity relaxation towards observed surface salinity with a restoring time scale of 365 days (version 1) and 300 days (version 1.2) for a 50 m thick surface layer. The restoring is applied as a salt flux which is also present below sea ice. In model version 1.2 balancing of the global salinity relaxation flux was added as an option, which allows to keep the global mean salinity constant over long integration times. That is, positive (negative) relaxation fluxes (where "positive" means a salt flux into the ocean) are decreased (increased) by a multiplicative factor if the global total of the

relaxation flux is positive (negative). The somewhat shorter relaxation time scale applied in version 1.2 was chosen to approximately counteract the weakening of the relaxation flux due to this balancing procedure. Another option introduced in version 1.2 is a weaker salinity relaxation over the Southern Ocean south of 55° S with a linear ramp between 40

and 55° S. This measure can slightly improve the simulated Southern Ocean hydrography and tracer distributions. Both new options have been activated for all simulations with model version 1.2 presented here, and a time scale of 1050 days has been used for SSS relaxation south of 55° S.

#### 2.4.2 Initialisation and simulation set-up

The physical ocean model is initialised with zero velocities and the January mean temperature and salinity fields from the Polar Science Center Hydrographic Climatology (PHC) 3.0 (Steele et al., 2001). Initial concentrations for the biogeochemical tracers phosphate, silicate, nitrate and oxygen are taken from the gridded climatology of the World Ocean Atlas 2009 (WOA, Garcia et al., 2010a, b). DIC and alkalinity initial values are derived from the

GLODAP climatology (Key et al., 2004), using the estimate of anthropogenic DIC included in the data base to obtain a preindustrial DIC field. In regions of no data coverage a global mean profile of the respective data is used to initialise the model. Dissolved iron concentration is set to $0.6\,\mu\mathrm{mol\,m^{-3}}$ on initialisation. In the sediment module, the pore water tracers are set to the concentration of the corresponding tracer at the bottom of the water column,

and the solid sediment layers are filled with clay only on initialisation.

The model is spun up for 800 (version 1) and 1000 years (version 1.2) using the CORE normal year forcing and a prescribed preindustrial atmospheric $CO_2$ concentration of 278 ppm to get the oceanic tracers into a near-equilibrium state. The spin-up time span is still too short to attain full equilibrium, but the strong initial transient changes in

tracer concentrations following the initialisation flatten out after about 500 years simulation time. During the last 100 years of the spin-up carbon fluxes (positive into the ocean) stabilised at 0.26, $-0.005$, and $-0.004\,\mathrm{Pg\,C\,yr^{-1}}$, with only small trends of 0.00007, 0.021, and $-0.048\,\mathrm{Pg\,C\,yr^{-1}\,century^{-1}}$ for Mv1, Mv1.2, and Lv1.2, respectively. The relatively large up-

take of carbon in Mv1 at the end of the spin-up is due to a lower $CaCO_3$ to POC production ratio found in this configuration. A full equilibration of the model with respect to this process would require a considerably longer spin-up. The re-tuning of the ecosystem parameterisation, which increases $CaCO_3$ production in version 1.2, leads to carbon fluxes much closer to zero at the end of the spin-up period. The larger trends in Mv1.2 and Lv1.2 compared to Mv1 despite longer spin-up time is due to larger decadal to centennial scale internal variability in these configurations (i.e., the systematic long term drift is much smaller). We attribute this to the details of the salinity relaxation (balancing of the restoring flux, weaker relaxation south of $40^\circ$ S). We finally note that even these larger trends are tiny compared to the changes in ocean carbon uptake due to anthropogenic carbon emissions, and that we calculate our estimates of anthropogenic carbon uptake relative to a control run to account for offsets and trends due to a not fully equilibrated model.

Following the spin-up, we initialise two hindcast runs for each model version and configuration. The "historical" run uses prescribed atmospheric $CO_2$ concentrations taken from the data set provided by Le Quéré et al. (2015). We start model integrations on 1 January 1762, since atmospheric $CO_2$ begins to exceed the spinup value of 278 ppm in this year. The second hindcast, which we call "natural" run in this text, is continued with the constant preindustrial $CO_2$ concentration of 278 ppm. Northern and Southern Hemisphere tropospheric concentrations of CFC-11, CFC-12, and $SF_6$ are prescribed according to Bullister (2014). The historical and natural simulations continue to be forced by the CORE normal year forcing until 1947. At 1 January 1948 we branch off historical and natural runs that are forced by CORE and NCEP-C interannually varying forcing (two historical and two natural runs for each model version and configuration). In the following sections we focus mainly on the simulations forced with the NCEP-C-IAF data set, and all results presented have been obtained with this forcing unless explicitly stated otherwise.

## 2.5 Technical aspects

The HAMOCC code was originally written in FORTRAN 77. It was later re-written to take advantage of Fortran 90 elements (e.g. ALLOCATE statements) and to conform to the For-

tran 90 free source code format. Certain model options, such as, for example, the selection of a POC sinking scheme, are implemented via C-preprocessor directives. HAMOCC should compile on any platform that provides a FORTRAN compiler and a C-preprocessor.

MICOM is parallelised by dividing the global ocean domain horizontally into (logically)
rectangular tiles which are processed on one processor core each. Communication between cores is implemented using the Message Passing Interface (MPI) standard. Since HAMOCC is integrated into MICOM as a subroutine call it inherits this parallelism. Note that HAMOCC, in addition, has been parallelised for shared memory systems using the OpenMP standard. This feature is, however, not tested and not supported for the current
set-up. For the simulations presented in this manuscript, the ocean component has been run on 190 (Mv1), 309 (Mv1.2), and 155 (Lv1.2) cores on a Cray XE6-200 system, yielding a model throughput of 11 (Mv1), 20 (Mv1.2), and 64 (Lv1.2) simulated model years per (wall-clock) day.

## 3 Results and discussion

### 3.1 Physical model

#### 3.1.1 Temperature and salinity

Zonal mean temperature ($T$) and salinity ($S$) differences between model and the observation based climatology from WOA 2009 (Locarnini et al., 2010; Antonov et al., 2010) for the Atlantic and Pacific basins are shown in Figs. 1 and 2. The general patterns of $T$ and
$S$ deviations are similar across the different model versions and configurations. While all three configurations include salinity relaxation, this is not balanced in the case of Mv1, with the result that average salinity falls by 0.2 units during the course of the integration. The predominantly negative salinity bias for the Mv1 configuration is visible in Fig. 2. The mid-latitude and tropical regions have a too strong temperature gradient in the upper 700 m, that
is, a warm bias in the upper thermocline and a cold bias below. While magnitude and extent

of the warm bias are similar for the three model configurations (up to $\approx 4\,°C$), the cold bias is weakest for Mv1, and strongest (up to $-4\,°C$) for the low resolution configuration Lv1.2.

In the Southern Ocean south of $50°\,S$, the model is generally biased towards too cold and fresh conditions (about $-1$ to $-2\,°C$ and up to $-0.5$ units on the Practical Salinity Scale)
below a slightly too warm surface layer. The layer of upwelled Atlantic deep water, which is warmer and more saline than Southern Ocean surface waters, is not or only weakly preserved in the model. At depths below 3000–4000 m the cold and fresh bias extends northwards into the Atlantic basin up to the equator (a weak cold bias extends into the North Atlantic at depth). Above these bottom water masses and below the thermocline the
Atlantic is generally too warm by 1 to $2\,°C$ and too saline by 0.2 to 0.5 units (again, salinity is biased low in Mv1 due to the unbalanced salinity restoring flux). In contrast, a warm and saline bias at intermediate depths in the Pacific is mostly confined to the Southern Hemisphere, whereas the North Pacific is biased cold and fresh at all depths below 1000 m for all model configurations. We note that the cold and fresh bias in the Southern Ocean
water column is specific to the stand-alone configuration of the model and is not found in the fully coupled version of NorESM (e.g. Bentsen et al., 2013, Fig. 14).

### 3.1.2 Atlantic Meridional Overturning Circulation

The Atlantic Meridional Overturning Circulation (AMOC) for the ocean-ice only configuration of Mv1.2 has been compared to results from other models in Danabasoglu et al. (2014).
The forcing protocol for this latter study was to run the model through five cycles (1948–2007) of the CORE-IAF. The AMOC strength in our model, measured as the maximum of the annual mean meridional stream-function at $26.5°\,N$, varied between 12.5 and 19 Sv and showed an increasing trend of roughly 1 Sv century$^{-1}$. Under the spin-up with the CORE-NY forcing performed in the present work, the AMOC shows a long transient increase in
strength for about 300 years before stabilising. The average overturning for the 30 years before switching to inter-annual forcing (1918–1947) is 22.3, 24.8, and 21.5 Sv for Mv1, Mv1.2, and Lv1.2 respectively (Fig. 3, left panel). Aside from the absolute values we find similar curves of AMOC strength under the forcing protocol applied here (Fig. 3,right panel)

compared to the results presented in Danabasoglu et al. (2014): first a 10 to 15 year decrease by 2 to 4 Sv followed by a relatively stable phase until the early 1980s, an increase by 4 to 7 Sv towards a maximum in the late 1990s and another decrease until the end of the simulation period (2007 or 2014). The Lv1.2 configuration forced by NCEP-C-IAF is an outlier in our small model ensemble. Compared to the other model simulations the annual and decadal scale variability of AMOC strength appears to be similar, but superimposed onto a negative trend of 8 Sv over the simulation period (see below).

In model configuration Mv1 we find only minor differences in AMOC strength between the simulation forced with CORE-IAF and the one forced with NCEP-C-IAF. For Mv1.2 and Lv1.2, however, the CORE-IAF simulations show a weaker initial decrease and a generally larger overturning than the corresponding NCEP-C-IAF simulations. These differences can be traced back to a peculiarity of the salinity relaxation scheme when the balancing of the relaxation flux is activated (not available in model version 1). Since too much salt is taken out of the surface ocean globally by the unbalanced scheme, negative salt fluxes are reduced by a multiplicative factor when balancing of the salinity relaxation flux is activated. Interestingly, the global restoring salt flux imbalance is larger when the model is forced with the CORE-IAF. As a consequence of correcting this imbalance, we find a stronger reduction of the restoring salt flux out of the surface ocean in the simulations performed with CORE-IAF compared to those forced with NCEP-C-IAF. In the Atlantic north of $40°$ N a positive salinity bias is therefore reinforced in the CORE-IAF simulations with model version 1.2, driving an increase of the AMOC relative to the simulations forced with NCEP-C-IAF. This effect is particularly pronounced in the Lv1.2 configuration leading to the negative AMOC trend of 8 Sv described above. We note that the AMOC strength has been estimated as 17.2 Sv for the time period from April 2004 to October 2012 based on observations from the Rapid Climate Change programme (RAPID) array (McCarthy et al., 2015, diamond in Fig. 3). Compared to this value our model overestimates the AMOC strength by about 4 to 9 Sv, except for the special case of Lv1.2 forced with NCEP-C-IAF discussed above, where AMOC is lower than the observational estimate by about 2 Sv for the period from 2004 to 2012.

### 3.1.3 Mixed layer depth and CFC-11

Seasonal cycles of modeled average bulk mixed layer depth (MLD) compared to an observation based MLD climatology (de Boyer Montégut et al., 2004) are shown for several regions in Fig. 4. The climatology uses a threshold criterion for density, that is, MLD is de-
fined by the depth where density has increased by $0.03 \, \mathrm{kg \, m^{-3}}$ relative to its near surface value. We note that the depth of the bulk mixed layer in our model is calculated based on energy gain and dissipation in the surface ocean, and that modelled and observed quantities are therefore not directly comparable. A MLD climatology based on a density criterion is nevertheless suitable for comparison with our model, since the criterion measures strat-
ification directly. We find a good agreement of modelled and observation based MLD in terms of phasing and summer minimum depth, whereas during autumn and winter modelled average MLDs are up to 40 to 100 m larger outside the tropics. The largest differences are found in the Southern Ocean south of $60° \, \mathrm{S}$ (Fig. 4f) during austral winter. However, the observation based climatology relies on very few profiles during May–October in this
region. Further, our calculation of model average MLD excluded ice covered grid cells while we have no information about how many observed profiles taken under ice cover in the Antarctic entered the climatology. We therefore have little confidence in the model–data comparison for the southernmost Southern Ocean during winter. In the tropics modelled average MLDs are roughly 20 m larger than observed year round (i.e. about 60 vs. 40 m)
with a small annual cycle that corresponds well to the data based estimate. Generally, the three model versions and configurations show very similar average MLD, but the low resolution configuration Lv1.2 has a less pronounced seasonal cycle (i.e. the winter maximum MLD is shallower and hence closer to the observation based estimate).

MLD averaged over large regions and time periods is not necessarily a useful indicator
of upper ocean ventilation since spatially and temporarily localised convection events can transport vertically large amounts of heat and matter. In fact, the modelled maximum MLD is frequently larger than 800 m over extended areas of the Southern Ocean (not shown). The zonal mean distribution of CFC-11 in model version 1.2 (Fig. 5) indicates a deeper than

observed mixing compared to CFC-11 profiles from the GLODAP data base. In the Atlantic sector of the Southern Ocean between 500 to 1000 m depth we find relatively high average CFC-11 concentrations of up to 3 nmol m$^{-3}$, whereas the corresponding observed CFC-11 concentration is 0.44 nmol m$^{-3}$. These high concentrations are mainly caused by large fluxes occurring during Antarctic winter north of the ice edge. We note that forcing the model with the NCEP-C-IAF data set tends to attenuate the deep mixing in the Southern Ocean compared to simulations carried out with the CORE-IAF (see Fig. 5c). In the North Atlantic we find frequent deep convection with maximum MLDs as deep as 1400 m in the Labrador and Irminger Sea as well as in the Greenland and Norwegian Sea. The comparison with GLODAP CFC-11 data reveals a tendency for high CFC-11 concentrations located at too large depths also in the Labrador and Irminger Sea (GLODAP does not cover the Nordic Seas). In mid latitudes we find less CFC-11 in central and intermediate water masses, best seen as a negative bias in the modelled zonal mean integrated CFC-11 content (Fig. 5e) between 40° S and 40° N.

## 3.2 Primary and export production

Differences between version 1 and 1.2 of NorESM-OC are to a large extent due to differences in the ecosystem parameter settings. We compare model results with satellite derived estimates of primary production (PP), which are based on Moderate-Resolution Imaging Spectroradiometer (MODIS) data and three different processing algorithms, the Vertically Generalized Production Model and its "Eppley" variant (VGPM and Eppley-VGPM, Behrenfeld and Falkowski, 1997) and the Carbon based Production Model (CbPM, Westberry et al., 2008). We processed ten years of these data (2003–2012; obtained from www.science.oregonstate.edu/ocean.productivity) into three climatologies as detailed in the Appendix. Since continental shelf regions are only poorly resolved in the model, particularly in the lower resolution configuration, we focus on evaluating the large scale open ocean productivity. We therefore exclude data from shelf regions from the following analysis (see Appendix for details).

Figure 6 shows the mean PP over the years 2003–2012 for the model configurations Mv1, Mv1.2, and Lv1.2 and the mean and range of the three satellite based climatologies. As noted in Sect. 2.3.2, model version 1 has a very strong PP at high latitudes, a feature that is not found in the satellite derived estimates. Modelled values are high in excess of 10 mol C m$^{-2}$ yr$^{-1}$ almost everywhere south of 40° S and in large areas north of 40° N (Fig. 6a and e). These large discrepancies are reduced in model version 1.2 through the re-tuning of the ecosystem model described in Sect. 2.3.2. Sensitivity tests with the same physical model but both versions of the biogeochemistry module (not shown) indicate that changes in the physical fields between model versions do not contribute significantly to this result. A positive bias with respect to the MODIS-based estimates is still found south of 40° S, but differences are reduced to less than 5 mol C m$^{-2}$ yr$^{-1}$. North of 40° N large scale average PP in model version 1.2 generally compares well with the observation based estimates, although values are now at the lower end of the range given by the three satellite climatologies.

At low latitudes (40° S to 40° N) outside of equatorial and coastal upwelling regions, all three model versions and configurations show a rather low production. Since this region represents a large area of the world ocean, the global open ocean integrated PP is lower than the mean of the MODIS-estimates of 46.3 Pg C yr$^{-1}$ for all model configurations (Table 4; 43.3 for Mv1, 35.1 for Mv1.2, and 29.6 Pg C yr$^{-1}$ for Lv1.2). Figures 6e and 7 show that this discrepancy indeed originates in the latitudes between 40° S and 40° N. Despite the ecosystem re-tuning and the lower nutrient consumption in the Southern Ocean in model version 1.2, PP is only slightly higher in Mv1.2 compared to Mv1 (and about equal in Lv1.2). We conclude that rather than caused by inadequate ecosystem parameter settings, the modelled low PP at low latitudes is caused by a too low nutrient supply from below the euphotic zone, due to too stably stratified water masses (warm bias at the surface, cold bias in the lower thermocline, Fig. 1). The lower PP in the low resolution is consistent with the stronger cold bias found in this configuration.

The mean seasonal cycle of vertically integrated PP (averaged over the years 2003–2012) is shown in Fig. 8 for six regions. The too strong productivity in model Mv1 is reflected

in a very pronounced productivity peak in late spring or early summer everywhere south of 40° S and north of 40° N. Model version 1.2 shows a significantly reduced annual cycle of PP in these regions (approximately 40–80 % reduction of peak production), much better in line with the MODIS based PP estimates. In the North Atlantic and North Pacific the peak

production of 30 mol C m$^{-2}$ yr$^{-1}$ in the updated model comes close to the satellite derived value. In the southern high latitude region (60 to 40° S) the modelled PP also peaks at about 30 mol C m$^{-2}$ yr$^{-1}$, but here the corresponding MODIS estimates are much lower (peaking at 10 mol C m$^{-2}$ yr$^{-1}$). In late summer, autumn, and winter modelled PP at high latitudes is lower than the observational estimate.

In the tropics outside the Indian Ocean there is no significant annual cycle in the model as well as in observations. The seasonal variation in PP caused by the monsoon in the Indian Ocean is well captured by the model although there appears to be a small phase shift of about one month relative to the MODIS based PP estimates (Fig. 8c). The subtropical annual cycle of PP shows little difference between model versions. In the southern sub-

tropics (20 to 40° S, Fig. 8d, very similar results for the northern subtropics) we find a too strong seasonal cycle, with production peaking in early Southern Hemisphere spring (October/November) in the model, in line with the mean of the three MODIS based climatologies.

### 3.2.1 Export production of POC, $CaCO_3$, and opal

Figure 9 shows the mean export production of POC, as well as the $CaCO_3$ and opal to

organic carbon ratios in exported matter. The spatial pattern of POC export closely resembles the pattern of PP, since the fraction of PP that is exported as POC (about 15–25 %, not shown) shows only small spatial variations. Total annual carbon export (averaged over 2003 to 2012, Table 4) is 8.8 (Mv1), 7.1 (Mv1.2), and 6.1 Pg C yr$^{-1}$ (Lv1.2).

Since the partitioning between opal and $CaCO_3$ export production is parameterised de-

pendent on available silicate in our model, $CaCO_3$ export dominates over opal export only in regions depleted of surface silicate, which are the subtropical gyres in the Atlantic and the south Pacific. Due to the increased silicate to phosphorus uptake ratio $R_{Si:P}$ in model version 1.2 (clearly visible in Fig. 9 g to i) the $CaCO_3$ production is maintained or even slightly

expanded into the western North Pacific despite the much lower PP (and surface silicate consumption) at high latitudes in this model version. We note that the simple parameterisation of opal and $CaCO_3$ export production is qualitatively supported by opal to particulate inorganic carbon (PIC) ratios derived from sediment traps. For example, Honjo et al. (2008,
see their Fig. 7) show that high opal/PIC ratios are constrained to ocean regions with high surface silica concentrations, a pattern that is qualitatively reproduced by our model.

Total modelled opal export (between 95 and 120.5 Tmol Si yr$^{-1}$, Table 4) is within the uncertainty range of the estimate of 105$\pm$17 Tmol Si yr$^{-1}$ given by Tréguer and De La Rocha (2013). The ratio of $CaCO_3$ export to organic carbon export of 7.1 to 7.9 % is within the
10 range estimated by Sarmiento et al. (2002, 6$\pm$3 %).

### 3.3 Biogeochemical tracers

#### 3.3.1 Nutrients

Maps of surface phosphate concentrations for the three model versions and configurations Mv1, Mv1.2, and Lv1.2 as well as the observation based global climatology from WOA 2009
(Garcia et al., 2010b) are shown in Fig. 10. The reduced high latitude PP in model version 1.2 leads to larger surface phosphate concentrations almost everywhere north of 40° N and south of 40° S compared to Mv1. This improves the agreement with observations (i.e. reduces the negative PO$_4$ bias found in Mv1) in the Southern Ocean and in the North Pacific. In contrast, a positive PO$_4$ bias is introduced in the North Atlantic north of 40° N in
model version 1.2. In the Southern Ocean simulated PO$_4$ surface concentrations are closest to observations in the configuration Lv1.2.

In all model versions/configurations there is too much phosphate trapped at intermediate depth in the tropical oxygen minimum zones, while there is a negative bias below approximately 2000 m depth as well as in the whole water column of the Southern Ocean. This
general pattern is shown for the Atlantic in Fig. 11 but is also found in the other ocean basins. Besides the broad similarities, a considerable redistribution of phosphate at depth can be observed across the different model configurations. Due to the lower PP in the

Southern Ocean in model version 1.2 less phosphate is trapped south of 50° S and more phosphate is found in the oxygen minimum zones of the tropical oceans. A similar redistribution of nutrients in the world oceans in response to a changed export production in the Southern Ocean has been reported for other models earlier (e.g. Sarmiento and Orr, 1991; Marinov et al., 2006; Primeau et al., 2013).

To investigate the different phosphate distributions further, we plot preformed phosphate in Fig. 12. Here, we approximate preformed phosphate by $PO_4^{pref} \approx PO_4 - AOU/R_{O_2:P}$ (Broecker et al., 1985, where $R_{O_2:P} = 172$ is the stoichiometric oxygen to phosphorus ratio used in the model), since a preformed phosphate tracer was not available in version 1 of the model, and because we can estimate preformed phosphate from WOA observations in the same manner. The differences between the approximation and the explicitly simulated preformed tracer (available in version 1.2) are relatively small in the Atlantic where anaerobic remineralisation of POC is not abundant in the model. The preformed phosphate concentrations are very similar in Mv1 and Mv1.2 south of 50° S. Hence, the stronger negative bias in total $PO_4$ found in Mv1.2 is indeed caused by a smaller remineralised fraction. The too strong PP in Mv1 in the Southern Ocean also results in too low preformed phosphate concentrations in Antarctic Intermediate Waters. This bias is clearly reduced in model version 1.2 since less nutrients are stripped out of surface waters by biological production in the Southern Ocean. Likewise, the increased surface $PO_4$ concentration in model version 1.2 in the Atlantic north of 40° N leads to higher than observed preformed phosphate concentrations in intermediate and deep waters in the Atlantic basin.

The modelled distributions of nitrate and phosphate are similar in terms of biases and general spatial structure since the model uses a fixed stoichiometric ratio ($R_{N:P} = 16$) for the composition of organic matter. The difference $[PO_4] - [NO_3]/R_{N:P}$ (Fig. 13) is positive everywhere in broad agreement with observations, indicating that nitrate is depleted relative to phosphate with respect to the canonical N:P ratio of 16. Large values of this difference are found in the tropical Pacific, in the model as well as in observations, but more pronounced in the model. This pattern is due to the tropical Pacific oxygen minimum zone (OMZ) where $NO_3$ is consumed during denitrification to remineralise organic matter and

release PO$_4$. The oxygen minimum zones are excessively large in our model, particularly in the tropical Pacific (see Sec. 3.3.2). These results show, that the simple parameterisation of nitrogen fixation, which occurs in the surface ocean as soon as $[NO_3] < R_{N:P}[PO_4]$ in our model, is active over the whole surface ocean, which is probably unrealistic. This simple parameterisation should be viewed more as a means to keep the model ocean close to the assumed stoichiometric N:P ratio than a realistic parameterisation of nitrogen fixation. At depth, we find major deviations from the similarity of nitrate and phosphate distributions in the OMZ of the tropical Pacific. Here, our model shows a local minimum of nitrate (due to the too strong denitrification caused by too low oxygen values) instead of a local maximum as observed and as it is found for phosphate.

Surface silicate concentrations (Fig. 14) in the Southern Ocean are much lower than observed in Mv1. Larger Si concentrations and hence a better agreement with observations are simulated with model version 1.2 due to the reduced PP in the Southern Ocean. Again (as for maximum PO$_4$ concentrations) the maximum Si values are closest to observations in Lv1.2. The lower surface silicate concentrations equatorward of approximately 35° in version 1.2 compared to version 1 is maintained (despite the much lower Si consumption in the Southern Ocean) due to the re-tuned silicate uptake (increased $R_{Si:P}$). Although the concentration differences are not large, particularly not relative to the half-saturation constant for silicate uptake ($K_{Si} = 1 \, \text{mmol Si m}^{-3}$), the effect over more than 1000 years of integration is a clear reduction of alkalinity in model version 1.2 (see below) due to a slightly but constantly larger fraction of calcareous shell formation.

Generally, the global spatial surface concentration pattern of silicate is less well reproduced than the surface pattern of PO$_4$ in all model configurations. The area of high silicate concentration around Antarctica is broader (i.e. extents further north) than observed and the observed north–south asymmetry with much smaller values in the northern high latitudes (north of 40° N) is not well reproduced either. We note that the pattern of elevated silicate concentrations in the North Pacific and Arctic is similar to the pattern of PO$_4$ concentration in the model (Fig. 10). While this is in good agreement with observations for phosphate, observed maximum Si concentrations are 50% smaller than modelled maximum concen-

trations. This might indicate that our ecosystem model is not tuned well enough or that its structure is oversimplified with respect to silica cycling (e.g., fixed Si:P ratio, fixed constant sinking speed).

A summary of model skill for surface and interior distributions of nutrients is given in the Taylor-diagrams shown in Fig. 15. Surface ocean nitrate and phosphate show high correlations with observed fields ($R > 0.9$ except for $PO_4$ in Mv1 where $R = 0.87$) and a good agreement of spatial variability. Spatial variability is lowest in Mv1 due to the underestimation of surface $PO_4$ and $NO_3$ concentrations at high latitudes. Highest skill scores are found for Lv1.2 consistent with the fact that this model configuration comes closest to observed surface concentrations in the Southern Ocean. Simulated surface silicate fields show significantly lower correlations with observations (between $R = 0.5$ for Mv1 and $R = 0.8$ for Lv1.2). Spatial Si variability is too high for model version 1.2 because of the too large Si surface concentration north of $40°$ N, while Mv1 reproduces the observed surface Si variability, however with a wrong spatial pattern.

The improved simulation of surface nutrients in model version 1.2 comes at the cost of less model skill in the interior ocean (Fig. 15b–d). Apparently, the too strong PP in the Southern Ocean in Mv1 partly compensates for deficiencies in the circulation field. As discussed above, a larger Southern Ocean PP increases the amount of phosphate trapped in the water column south of $50°$ S and reduces the amount of phosphate trapped in the tropical oxygen minimum zones in version 1 compared to version 1.2 (Fig. 11). Hence, model version 1.2 shows less correlation and a larger variability of $PO_4$ at 500 m depth. Simulated nitrate fields at this depth level are modified by denitrification. Since the tropical oxygen minimum zones are considerably larger than observed, particularly in the Pacific, model skill with respect to $NO_3$ is reduced.

The dominant impact of the oxygen minimum zones on model skill at 500 m depth can be assessed by restricting the analysis to the extra-tropical ocean (i.e., by omitting grid points located between $20°$ S and $20°$ N, Fig. 15d). Then, phosphate variability is very close to observed, and the correlation of modelled nitrate with observations increases by almost 0.3 for all model configurations such that it becomes similar to that of phosphate.

At 2000 m depth skill scores for Mv1 and Mv1.2 are similar for the three nutrients. Lv1.2 reaches better skill scores (or about equal for silicate) than the higher resolution configurations. Silicate has a rather uniform distribution at this depth, and is reproduced well by all configurations, leading to relative high skill scores for Si.

### 3.3.2 Oxygen

As discussed in Sect. 3.1 there is too much deep convection and associated tracer transport to depth in the Southern Ocean in all NorESM-OC versions and configurations. Oxygen is particularly affected, since $O_2$ solubility at cold temperatures is high (this is also true for $CO_2$, but the upwelled water masses in the Southern Ocean tend to be rich in DIC and low in oxygen). During the long simulations (more than 1000 years) presented here the deep ocean fills up with oxygen rich waters originating from the Southern Ocean. The resulting zonal mean $O_2$ concentrations in the central and east Pacific basin and the differences with respect to the WOA 2009 climatology (Garcia et al., 2010a) are shown in Fig. 16. In the Southern Ocean south of 50° S and at depths deeper than 500 m we find an average positive bias of 0.05, 0.1, and 0.09 mol m$^{-3}$ in Mv1, Mv1.2, and Lv1.2, respectively. The much lower bias in model configuration Mv1 is due to the larger flux of organic matter from the surface ocean. We find average AOU values of 0.09, 0.04, and 0.05 mol m$^{-3}$ (average south of 50° S at depths deeper than 500 m), such that preformed $O_2$ is virtually exactly the same for all model versions and configurations. We note that part of the deep ocean oxygen bias is connected to low POC transport into the deep ocean when the standard sinking scheme is used. The large positive bias below 3000 m depth is reduced by 40% in Lv1.2 when using alternative sinking schemes. This is further discussed in Sect. 3.5.

The modelled oxygen minimum zones (OMZs) are excessively large in all model versions and configurations. This is particularly true for the Pacific basin (Fig. 16), and to a lesser degree for the tropical Atlantic and tropical Indian Ocean (not shown). The impact of the parameterisation of POC sinking on the modelled OMZs is discussed in Sect. 3.5. We finally speculate that the strong $O_2$ biases of opposite sign in adjacent water masses are probably more pronounced in our isopycnic model than they would be in a $z$-coordinate

model, since the strong $O_2$ gradients between these water masses are not alleviated by numerical diffusion.

### 3.3.3 Dissolved inorganic carbon and alkalinity

Surface maps of DIC and alkalinity for the three model configurations and the correspond-
ing data based GLODAP climatologies are shown in Figs. 17 and 18. Model version 1 has a strong positive bias in DIC and alkalinity, which is most pronounced between approximately $40°$ S and $50°$ N. This bias is driven by a too large production of alkalinity in these latitudes. In our model, alkalinity is (aside from advection and diffusion) governed by biological production, which adds one mole alkalinity per mole nitrate consumed, and calcium
carbonate production, which decreases alkalinity by 2 moles per mole $CaCO_3$ produced (Wolf-Gladrow et al., 2007). Since the partitioning between opal and $CaCO_3$ export production is parameterised dependent on available silicate, the reduction of surface silicate concentrations at low latitudes in model version 1.2 (Sect. 3.3.1) results in increased $CaCO_3$ production and a considerable reduction of the alkalinity and DIC biases. Since modelled
alkalinity at low latitudes is quite sensitive to $CaCO_3$ export (see also Kwiatkowski et al., 2014), we take the reduction of alkalinity biases as indirect evidence for an improved distribution of $CaCO_3$ export in Mv1.2 and Lv1.2.

Average surface DIC concentrations in the Southern Ocean are lowest for Mv1.2. Again, as in the case of phosphate, the reason for this is a lower remineralised concentration of DIC
in the upwelled water masses compared to Mv1, and a lower preformed DIC concentration compared to Lv1.2 (not shown, but compare Figs. 11 and 12).

### 3.4 Oceanic $p$CO$_2$, ocean carbon sink, and anthropogenic carbon

Compared to the observation based data products of Takahashi et al. (2009) and Land-
schützer et al. (2015a, b) we find that the general pattern of surface $CO_2$ partial pressure in
sea water ($p$CO$_2$, Fig. 19) is well reproduced by the model. The $p$CO$_2$ maximum at low latitudes is broader and less confined to the equatorial region than observed. The main factors

contributing to this mismatch are the very high primary production in the narrow eastern equatorial Pacific upwelling band (compare Fig. 6), which draws down $pCO_2$ in the model where it has its maximum in observations, and the higher than observed values in the upwelling areas along the west coast of America and Africa. In the southernmost Southern

Ocean modeled mean $pCO_2$ values are about 35 (Mv1.2) and 20 ppm (Lv1.2) lower than in the Landschützer et al. (2015a) data product. This is consistent with the DIC and alkalinity distributions found in this region (see Figs. 17 and 18). The differences between the Takahashi et al. (2009) $pCO_2$ climatology and a multiannual mean of the Landschützer et al. (2015a) data are only small.

We finally use the historical and natural runs to estimate the contemporary ocean carbon sink as well as the total storage of anthropogenic carbon ($DIC_{ant}$) in the ocean. The ocean carbon sink is defined here as the carbon flux into the ocean including its natural variability corrected for the mean and trend found in the natural simulation. This flux is calculated for the recent decades from 1959 to 2013 with the starting date 1959 chosen such that any

shock due to the change of atmospheric forcing in 1948 has vanished and does not influence the calculation of trends. Results for the different model versions and configurations forced with NCEP-C-IAF and CORE-IAF are shown in Fig. 20.

The simulated ocean carbon sink is largest for model version 1 with a mean carbon sink of 2.58 and 2.50 Pg C yr$^{-1}$ during 1990 to 1999 when forced with the NCEP-C-IAF and

20 CORE-IAF, respectively (Table 5). The accumulated uptake over 1959 to 2007 for this model version and the two forcing data sets differs in the same proportion (102 vs. 99 Pg C). Model version 1.2 shows a lower ocean carbon sink with a mean value of 2.33 Pg C yr$^{-1}$ over the 1990 to 1999 period for the medium resolution configuration (2.46 when forced with CORE-IAF). The corresponding numbers for the low resolution simulation forced with the NCEP-C

and CORE-IAF are 2.01 and 2.24 Pg C yr$^{-1}$, respectively. We compare these model results to the three independent data based estimates (grey symbols with error bars in Fig. 20) of McNeil et al. (2003), $2.0 \pm 0.4$ Pg C yr$^{-1}$ over 1990–1999, Manning and Keeling (2006), $1.9 \pm 0.6$ Pg C yr$^{-1}$ over 1990–2000, and Mikaloff Fletcher et al. (2006), $2.2 \pm 0.25$ Pg C yr$^{-1}$ for 1995, which are the basis for the ocean sink value given in the Intergovernmental Panel

on Climate Change Fourth Assessment Report (Denman et al., 2007), and which are also used as the data based ocean sink estimate in Le Quéré et al. (2015). The simulated ocean sink appears to be too large in Mv1 while for version 1.2 the sink estimates are within the error bars of the data based estimates. The closest match is found for the low resolution configuration forced with NCEP-C-IAF which has the lowest carbon uptake. The rather too high $CO_2$ uptake is confirmed when comparing the model results to the $CO_2$ flux estimates of Landschützer et al. (2015a). There is a relatively good agreement with the $CO_2$ sink simulated with Lv1.2 up to 1990 and towards the end of the time series (year 2011). From the early 1990s until 2001 the Landschützer et al. (2015a) data shows a marked decline followed by a steep increase of carbon uptake, which they attribute to a saturation followed by a reinvigoration of the Southern Ocean carbon sink. Although our model also shows a stagnation of the ocean carbon sink during the 1990s there is no pronounced minimum and hence a rather large deviation from the Landschützer et al. (2015a) data around the year 2000.

Also compared to results of similar ocean biogeochemical models forced with reanalysis data (Le Quéré et al., 2015, see Fig. 21 for names and references of individual models), the carbon uptake of NorESM-OC appears to be large. Figure 21 shows that after 1980, the carbon sink in Mv1 and Mv1.2 is above the range of other models, while Lv1.2 is at the upper end of the range.

The anthropogenic dissolved inorganic carbon ($DIC_{ant}$) stored in the ocean is displayed in Fig. 22 for the year 1994, and corresponding total numbers are given in Table 5. Consistent with the larger than observed carbon sink fluxes our model also shows higher anthropogenic carbon storage compared to the Sabine et al. (2004) estimate of $DIC_{ant}$ available from the GLODAP data base. In the North Atlantic modeled anthropogenic carbon is larger than observed, which is consistent with the too strong simulated AMOC. Model version 1.2 has a stronger AMOC when simulations are forced with the CORE-IAF (see Sect. 3.1.2), and we find a larger accumulation of $DIC_{ant}$ in the North Atlantic for this forcing (not shown) consistent with the larger (accumulated) uptake flux (Fig. 20). In the Southern Ocean the too high concentration of $DIC_{ant}$ is consistent with the too strong ventilation diagnosed from

the CFC-11 tracer (Sect. 3.1.3). Larger than observed $DIC_{ant}$ concentrations are also found north of $40°$ S where DIC is subducted in Antarctic Intermediate Waters. Compared to a recent synthesis of anthropogenic carbon storage estimates (Khatiwala et al., 2013, $155 \pm 31$ Pg C for the year 2010) modelled $DIC_{ant}$ storage of 186 (Mv1), 175 (Mv1.2), and 159 (Lv1.2) Pg C is higher, but still within the uncertainty range of the synthesis estimate.

## 3.5 Particle export and sedimentation

To evaluate the three different sinking schemes available in NorESM-OC1.2, we conducted a spin-up run over 1000 years for each scheme using the low-resolution configuration of the model. In addition we performed one sensitivity experiment employing the standard option, but with the constant POC sinking speed increased from 5 to $14$ m d$^{-1}$. These two runs are referred to as STD-slow and STD-fast, respectively, while the model runs employing the prognostic particle aggregation scheme (Kriest, 2002) and the linear increase of sinking speed are abbreviated KR02 and WLIN in the following text. The parameter settings used for the KR02 aggregation scheme can be found in Table 3, and we use $w_{min} = 7$ m d$^{-1}$, $w_{max} = 43$ m d$^{-1}$, and $a = 40/2500$ m d$^{-1}$m$^{-1}$ for the run WLIN. The global average sinking speed profiles for the four experiments and the resulting POC fluxes (normalised to the flux at 100 m depth) are shown in Fig. 23a and b, respectively.

Faster sinking particles provide a more efficient removal of carbon and nutrients from the surface ocean. Therefore, the amount of PP would drop considerably in STD-fast, KR02, and WLIN compared to the standard parameterisation if no other measures taken. Since already the PP of about 30 Pg C yr$^{-1}$ found for Lv1.2 is at the low end of observation based estimates (Carr et al., 2006), and since we wish to have a similar PP in the four configurations as a starting point for our evaluation, we tune the ecosystem parameterisation for the runs STD-fast, KR02, and WLIN towards less efficient export and more efficient recycling of nutrients in the euphotic zone. This is accomplished by reducing both the fraction of grazing egested $(1 - \epsilon_{zoo})$ and the assimilation efficiency of herbivores $\omega_{zoo}$ (see Table 2), a measure which reduces the fraction of grazing routed to detritus and increases the fraction that is recycled directly to phosphate (see Maier-Reimer et al., 2005, Eqs. 9 and 11). At the

end of the spin-up runs (years 1001–1010) we find globally integrated PP values of 29.6, 29.0, 28.4, and 30.4 Pg C yr$^{-1}$ for the experiments STD-slow, STD-fast, KR02, and WLIN. As mentioned above, the model is generally not in full equilibrium after 1000 years, but the remaining drift is small.

Figure 23c shows the average POC fluxes at 100 and 2000 m depth, and at the ocean bottom (where water depth is larger than 1000 m) for the four experiments and recent observation based estimates of these fluxes. The carbon export fluxes (POC flux through 100 m depth) of 4.0 to 5.6 Pg C yr$^{-1}$ found in the model are at the lower end of observation based estimates of 4.0 (Henson et al., 2012), 5.7 (Honjo et al., 2008), and 11.2 Pg C yr$^{-1}$, (Laws et al., 2000). The three runs employing the differently tuned ecosystem show a lower export than the STD-slow configuration. Note that in the KR02 scheme living phytoplankton is part of the sinking aggregates (in STD and WLIN phytoplankton does not sink), hence the relative large export despite the re-tuned ecosystem. The POC flux through 2000 m is significantly smaller than the observed range in STD-slow: 0.16 vs. 0.43 (Honjo et al., 2008) and 0.66 Pg C yr$^{-1}$ (Henson et al., 2012). It falls within or slightly above this range for the other three runs (0.45, 0.69, and 0.68 Pg C yr$^{-1}$ for STD-fast, KR02, and WLIN, respectively). A similar picture emerges for the bottom POC flux: In STD-slow it is again much smaller than the estimate of Seiter et al. (2005, 0.03 vs. 0.5 Pg C yr$^{-1}$), while the other sinking schemes yield values which are more comparable (though smaller: 0.17, 0.32, and 0.36 Pg C yr$^{-1}$ for STD-fast, KR02, and WLIN, respectively). We will evaluate the POC flux at 2000 m in more detail below.

Since the absolute POC fluxes depend on PP, and estimates of PP derived from satellite data differ widely, we plot the export ratio or export efficiency (EE, defined as $F_{100}^{POC}/PP$, where $F_{100}^{POC}$ is the POC flux at 100 m depth) as well as the transfer efficiency (TF, defined as $F_{2000}^{POC}/F_{100}^{POC}$) in Fig. 23d. The global average export efficiency in all four model runs (STD-slow 18 %, STD-fast 14 %, KR02 21 %, and WLIN 14 %) falls within the range estimated based on observations: 10 % (Henson et al., 2012), 16.3 % (Honjo et al., 2008), and 21 % (Laws et al., 2000). Consistent with the results for the absolute fluxes, the transfer efficiency obtained for STD-slow (1.7 %) is too small in comparison to the observational estimates of

7.5 % (Honjo et al., 2008) and 19 %(Henson et al., 2012), while the other runs fall within this range (STD-fast 9 %, KR02 10 %, and WLIN 16 %). These results indicate that the flux reaching the deep ocean below 2000 m and the sediments is likely too low in the STD-slow configuration. Regarding the three other schemes, a conclusion is difficult to draw, since the spread in the observation based estimates is large.

The global zonal mean and global mean profiles of remineralised phosphate ($PO_4^{remin} = PO_4 - PO_4^{pref}$, where $PO_4^{pref}$ is calculated based on AOU as above) shown in Fig. 24 and Fig. 25, respectively, reveal that there is a strong negative bias in the STD-slow simulations everywhere below 2000 m, and also at shallower depth in the Southern Ocean and north of 40° N. This bias is reduced if one of the three other sinking schemes is used, but the reduction is weaker in the deep ocean for STD-fast. A strong positive bias of $PO_4^{remin}$ is found in intermediate water masses at tropical and subtropical latitudes as well as in the tropical oxygen minimum zones for the STD-slow scheme. This bias is not significantly reduced in the STD-fast experiment, but there is a clear reduction found in KR02 and WLIN. Out of the three schemes with more efficient sinking KR02 appears to perform slightly better in terms of $PO_4^{remin}$ bias, while STD-fast clearly has less skill than KR02 and WLIN. We note that improvements in simulated $PO_4^{remin}$ directly translate into improved oxygen fields. The positive $O_2$ bias discussed in Sect. 3.3.2 is reduced in the deep ocean if one of the more efficient sinking schemes is employed. E.g., below 3000 m depth average $O_2$ biases are 0.1 mol m$^{-3}$ for STD-slow, 0.07 for STD-fast, and 0.06 for both KR02 and WLIN. Oxygen minimum zones are also better simulated in KR02 and WLIN, particularly in the Atlantic, whereas the improvement in the Pacific basin is modest. The absence of significant amounts of remineralised phosphate in the Arctic basin in our model (Fig. 24) is due to a combination of low PP and too strong ventilation (positive $O_2$ biases, compare Fig. 16).

We further evaluate the sinking schemes by comparison with a comprehensive synthesis of sediment trap data published by Honjo et al. (2008). These data have been normalised to 2000 m depth and comprise 134 stations of which we consider 120 here. We do not use ten stations for which the closest model grid point has a depth of less than 2000 m, and we disregard two stations in the high Arctic and two in the Mediterranean. Further, we

form the mean value of fluxes for stations situated in the same model grid cell leaving us with 102 model-data pairs. The scatter diagrams presented in Fig. 26 confirm the too low flux of POC in STD-low. POC fluxes at 2000 m in this model configuration are smaller than 0.05 mol C m$^{-2}$ yr$^{-1}$ for all but eight stations in the tropical Pacific. The latter stations are located in the narrow equatorial tongue of excessively high PP (compare Fig. 6). In this region we find the too large oxygen minimum zones at depth, where remineralisation by oxygen ceases and is replaced by less efficient processes (denitrification, sulphate reduction). As mentioned above the choice of the sinking scheme does not significantly improve the excessive oxygen minimum zones in the Pacific. Therefore, we also find some very high POC fluxes inconsistent with observations in STD-fast, KR02, and WLIN at tropical Pacific sites (Fig. 26b, c, and d).

STD-fast shows modeled POC fluxes through 2000 m depth that are still clearly too small. Except for a few sites located in the tropical Pacific fluxes are smaller than 0.2 mol C m$^{-2}$ yr$^{-1}$ whereas observed values range up to 0.6 mol C m$^{-2}$ yr$^{-1}$ in the Arabian Sea. One could argue that a further increase of the constant sinking speed would probably further improve the modeled fluxes. However, a constant sinking speed very efficiently removes POC and associated particular nutrients from the upper ocean. Already the 14 m d$^{-1}$ constant sinking speed is faster in the upper 500 m of the water column than the sinking speed in KR02 and WLIN (Fig. 23a). A significant increase of the constant sinking speed would therefore require a strong further tuning of ecosystem parameters towards more recycling and less efficient export to keep PP in the range of observed values. This would lead to even smaller values for export efficiency inconsistent with observation based estimates.

The KR02 scheme is able to reproduce the highest POC flux values, which have been observed in the Arabian Sea (orange dots in Fig. 26). This might be unsurprising, since the KR02 parameterisation has been originally developed and tested for application in this region. However, the scheme simulates also rather large flux values ($> 0.5$ mol C m$^{-2}$ yr$^{-1}$) for three nearby stations in the tropical Indian Ocean (yellow dots in Fig. 26) and one station in the subtropical Atlantic (blue dots) where the corresponding observed values are between

0.1 and 0.3 mol C m$^{-2}$ yr$^{-1}$. At the low end of observed flux values, the KR02 scheme is able to reproduce the lowest observed flux values well, but there are too many low POC flux values simulated by the scheme, mainly in the Southern Ocean south of 60° S (red dots), in the Atlantic north of 40° N (dark blue dots), and in the tropical Atlantic (light blue dots).

The WLIN scheme does not yield any fluxes larger than 3.2 mol C m$^{-2}$ yr$^{-1}$ outside the tropical Pacific, that is, the highest observed POC fluxes in the Arabian Sea are not well reproduced. Also, the scheme does not produce any fluxes smaller than 0.02 mol C m$^{-2}$ yr$^{-1}$. Hence also three of the four smallest observed flux values are better simulated by KR02 than WLIN. The correlations obtained with KR02 and WLIN are 0.32 and 0.22. However, if we omit the high flux stations in the Arabian Sea (station number 54 to 58 in Honjo et al., 2008), we find correlations of 0.23 and 0.28 for KR02 and WLIN, respectively. Hence, the relatively better correlation with sediment trap observations in KR02 compared to WLIN is solely due to better representation of high POC fluxes in the Arabian Sea. If we additionally disregard the stations in the tropical Pacific (stations 103 to 111 in Honjo et al., 2008), where the large discrepancy between model and sediment trap data is due to other model deficiencies than the POC sinking scheme, the correlations increase to 0.44 and 0.52 for KR02 and WLIN.

We have based the evaluation of the different POC sinking schemes on indirect (PO$_4^{remin}$) and direct (sediment trap) measurements. While the indirect method has inherent inaccuracies related to the calculation of AOU and the assumed stoichiometry of remineralisation, the direct measurement of POC fluxes by sediment traps also comes with large systematic uncertainties (see Honjo et al., 2008, and references therein). It is difficult to decide which of the two methods provides the most reliable evaluation and we therefore use both approaches here. We finally note that a comprehensive sensitivity analysis or a rigorous tuning of the different POC sinking schemes would require accelerated off-line integration techniques such as those applied by e.g. Kwon et al. (2009) or Kriest et al. (2012), which we to date have not available for our model.

## 4 Summary and conclusions

We have evaluated and compared two versions and several configurations of NorESM-OC, the stand-alone ocean carbon-cycle set-up of the Norwegian Earth System Model (NorESM). Model version NorESM-OC1 corresponds to the fully coupled NorESM1-ME,
which participated in CMIP5. Important updates made for NorESM-OC1.2 include the implementation of two new sinking schemes for particulate organic carbon (POC), and a re-tuning of the ecosystem parameterisation. We have presented results for different grid configurations (1.125° displaced pole grid for version 1, 1 and 2° tripolar grid for version 1.2) and two slightly different forcing data sets (CORE and NCEP-C).

The re-tuning of the ecosystem parameterisation reduces primary production (PP) in high latitudes in model version 1.2, which significantly improves agreement with observation based PP estimates. Also, simulation results for surface nutrients, dissolved inorganic carbon, and alkalinity are improved. On the other hand, model skill in the deeper ocean is slightly deteriorated. Particularly in the Southern Ocean, the stronger trapping of nutri-
ents and the larger consumption of oxygen for remineralisation due to the too high PP in version 1 partly compensate for deficiencies in the circulation field.

We find that the deep ventilation in the Southern Ocean is too strong in the model, such that, over the long integration times considered here, too much oxygen accumulates in the Southern Ocean and in deep waters ventilated from there. In strong contrast to these
20 positive $O_2$ biases, the model develops too large oxygen minimum zones along with a too strong accumulation of remineralised phosphate in the tropical oceans, particularly in the Pacific.

The ocean carbon sink as simulated with the model (using the standard POC sinking scheme) ranges from 2.01 to 2.58 Pg C yr$^{-1}$ for the time period 1990–1999. The higher
value stems from version 1 of the model and lies above the error bars of estimates based on observations. Version 1.2 of the model generally simulates a lower carbon uptake more in line with the observational estimates. We find only a weak sensitivity of our ocean sink estimate to the forcing data used, an exception being the low resolution configuration of

version 1.2 forced with the NCEP-C data set. For this configuration, a reduced ocean circulation (reduced AMOC) translates to a reduced carbon uptake (the lower end of the range given above).

Results of the simulations with different sinking parameterisations indicate that the standard sinking scheme used in our model so far (constant sinking speed, constant remineralisation rate) fails to transport enough carbon into the deep ocean, even if the sinking speed is increased considerably. The two new options implemented in version 1.2 (linear increase of sinking speed with depth, and a particle aggregation scheme, which calculates sinking speed prognostically) show much better agreement with sediment trap data at 2000 m depth. Deep ocean biases of remineralised phosphate (too low in the model) and $O_2$ (too high in the model) are reduced significantly, e.g., the average $O_2$ bias below 3000 m depth is reduced from 0.1 to 0.06 mol m$^{-3}$ for both new sinking parameterisations compared to the standard scheme.

Based on the evaluation presented in this work, there is no definite conclusion which of the two new POC sinking schemes performs better in our model. The examination of remineralised phosphate is slightly in favour of the particle aggregation scheme, which is also able to reproduce the highest recorded POC fluxes through 2000 m at stations in the Arabian sea. On the other hand, the bulk of sediment trap measurements is slightly better reproduced by simply assuming that the sinking speed increases linearly with depth. From a conceptual point of view, the aggregation scheme introduces additional parameters which are only poorly constrained by observational studies. Also, the prognostic particle aggregation depends on the number and mass of particles and consequently on primary production, which still is a quantity with large uncertainties in both the model and observational estimates. Obviously, the specification of a global sinking speed profile requires less parameters, but regional differences due to different ecosystem structure cannot be represented.

There are several directions for future model development that can be identified from our results. The parameterisations of iron cycling and nitrogen fixation are simplistic and should be improved for future model versions, particularly since the observational basis for both

processes has improved considerably in recent years. The tropical oxygen minimum zones are a long standing problem to be solved (e.g., Najjar et al., 1992; Dietze and Loeptien, 2013, and references therein). The hope that a more realistic parameterisation of POC sinking would also result in an improved simulation of oxygen minimum zones in our model,
particularly in the Pacific, has not been fulfilled. Part of this problem is the distribution of primary production which is too high in a narrow band along the upwelling water masses in the tropical Pacific and too low in the neighbouring oligotrophic subtropical gyres, leading to a wrong distribution of particle rain. Simulations with increased spatial resolution will help to elucidate which improvements can be made through a more accurate representation of
ocean currents in this region.

Future higher resolution model configurations will also offer the opportunity to study processes in shelf regions in more detail, where currently neglected processes are likely to be of importance. These include the parameterisation of tidal mixing induced sediment–water column interactions, and a fully consistent representation of the inflow of carbon and
15 nutrients through rivers.

In the Southern Ocean, we need to address temperature and salinity biases connected to deficiencies in the circulation field as well as the problem of too much deep ventilation. Due to the long integration time scales needed to spin-up the biogeochemistry module of our model, errors in the Southern Ocean circulation and ventilation can result in extended
biases in the interior ocean, e.g., the deep ocean $O_2$ bias seen in the model simulations presented here. Further, the rather high modeled anthropogenic carbon uptake originates (partly) in the Southern Ocean, and improvements of these estimates are urgently needed to help better constraining the carbon budget of the Earth system.

## Appendix A: Climatologies of satellite derived primary production

The three satellite climatologies were constructed from monthly gridded PP data for the years 2003–2012 that were obtained from the Ocean Productivity Website (www. science.oregonstate.edu/ocean.productivity). These data are derived from MODIS re-

trievals (MODIS.R2014 reprocessing) using the VGPM and Eppley-VGPM algorithms (Behrenfeld and Falkowski, 1997), as well as the CbPM algorithm (Westberry et al., 2008). From these three different PP estimates (original resolution $1/6$ of a degree) we first assembled monthly climatologies on a $1° \times 1°$ grid with the requirement that a grid point should have valid data for at least 3 out of 10 years. From these monthly climatologies we finally arrive at the annual climatologies by averaging over all months. In the last step we assume that grid points without data are not observed due to unfavourable light conditions, and therefore assume zero production there. This procedure might lead to an underestimation of PP in the satellite based climatologies at high latitudes ($> 50°$). Since we intend to validate our model with respect to the large scale open ocean PP, we exclude data from shelf regions from the analysis presented here. A $1° \times 1°$ grid cell was considered continental shelf where the average water depth (derived from the ETOPO2v2 bathymetry, National Geophysical Data Center, 2006) was shallower than 300 m.

**Code availability**

We are committed to share NorESM with the scientific community and to make the model available for scientific and educational purposes. As major model components are based on software developed by others, whose interests have to be protected, availability of the code is subject to signing two license agreements — one for the use of NorESM, and, additionally, for the use of HAMOCC signing of the MPI-ESM license agreement is required. Please mail to noresm-ncc@met.no for inquiries about obtaining the source code of NorESM. Signing of the MPI-ESM license agreement can be easily done through http://www.mpimet.mpg.de/en/science/models/model-distribution.html.

*Acknowledgements.* J. Schwinger, N. Goris, I. Bethke, M. Ilicak, and M. Bentsen are supported by the Research Council of Norway through project EVA (229771), and J. Tjiputra acknowledges the Research Council of Norway funded project ORGANIC (239965). M. Ilicak is supported by the SKD BASIC project and the project Ice2Ice that has received funding from the European Research Council under the European Community's Seventh Framework Programme (FP7/2007-2013) / ERC grant agreement no 610055. Supercomputer time and storage resources were provided by the Norwe-

gian metacenter for computational science (NOTUR, project nn2980k), and the Norwegian Storage Infrastructure (NorStore, project ns2980k). This work was supported by the Bjerknes Centre for Climate Research. We gratefully acknowledge the CESM project, which is supported by the National Science Foundation and the Office of Science (BER) of the US Department of Energy. Mixed layer depth climatologies based on observed profiles have been obtained from the IFREMER/LOS Mixed Layer Depth Climatology website www.ifremer.fr/cerweb/deboyer/mld. The authors thank two anonymous reviewers and the editor for their constructive and helpful comments, which improved this manuscript.

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

**Table 1.** Prognostic biogeochemical tracers in NorESM-OC.

| Tracer | Abbreviation | Unit |
|---|---|---|
| Carbonate system | | |
| Dissolved inorganic carbon | DIC | $kmol\ C\ m^{-3}$ |
| Total alkalinity | TA | $keq\ m^{-3}$ |
| Nutrients | | |
| Phosphate | $PO_4$ | $kmol\ m^{-3}$ |
| Nitrate | $NO_3$ | $kmol\ m^{-3}$ |
| Silicate | Si | $kmol\ m^{-3}$ |
| Dissolved iron | dFe | $kmol\ m^{-3}$ |
| Ecosystem state | | |
| Phosphorus in phytoplankton | PHY | $kmol\ P\ m^{-3}$ |
| Phosphorus in zooplankton | ZOO | $kmol\ P\ m^{-3}$ |
| Dissolved organic carbon | DOC | $kmol\ P\ m^{-3}$ |
| Particulate organic carbon | POC | $kmol\ P\ m^{-3}$ |
| Particulate inorganic carbon | PIC ($CaCO_3$) | $kmol\ C\ m^{-3}$ |
| Biogenic silica (opal) | BSi | $kmol\ S\ m^{-3}$ |
| Number of aggregates[a,b] | NOS | $cm^{-3}$ |
| Mass of dust in aggregates[a,b] | aDUST | $kg\ m^{-3}$ |
| Mass of non-aggregated dust | fDUST | $kg\ m^{-3}$ |
| Dissolved gases | | |
| Oxygen | $O_2$ | $kmol\ m^{-3}$ |
| Nitrogen | $N_2$ | $kmol\ m^{-3}$ |
| Nitrous oxide | $N_2O$ | $kmol\ m^{-3}$ |
| Trichlorofluoromethane[b] | $CFC-11$ | $kmol\ m^{-3}$ |
| Dichlorodifluoromethane[b] | $CFC-12$ | $kmol\ m^{-3}$ |
| Sulfur hexafluoride[b] | $SF_6$ | $kmol\ m^{-3}$ |
| "Preformed" tracers[b] (model diagnostics) | | |
| Preformed oxygen | $prO_2$ | $kmol\ m^{-3}$ |
| Preformed phosphate | $prPO_4$ | $kmol\ m^{-3}$ |
| Preformed alkalinity | prALK | $kmol\ m^{-3}$ |

[a]Only if particle aggregation (prognostic sinking speed) is activated.
[b]Available as of version NorESM-OCv1.2.

**Table 2.** Parameter values of the ecosystem parameterisation (see Maier-Reimer et al., 2005, for details) that have been changed between model version 1 and 1.2. Parameter values in brackets have been used for the model runs with the non-standard sinking schemes (Sect. 3.5).

| Parameter | Symbol | Version 1 | Version 1.2 | Unit |
|---|---|---|---|---|
| Half-saturation constant for $PO_4$ uptake | $K_{PO_4}$ | $1 \times 10^{-7}$ | $4 \times 10^{-8}$ | kmol P m$^{-3}$ |
| Zooplankton assimilation efficiency | $\omega_{zoo}$ | 0.5 | 0.6 (0.5) | – |
| Fraction of grazing egested | $1 - \epsilon_{zoo}$ | $1 - 0.8$ | $1 - 0.8 (1 - 0.9)$ | – |
| Zooplankton mortality rate | $\lambda_{zoo}$ | $5 \times 10^{-6}$ | $3 \times 10^{-6}$ | (kmol P m$^{-3}$ d)$^{-1}$ |
| Zooplankton excretion rate | $\lambda_{zoo,DOC}$ | 0.03 | 0.06 | d$^{-1}$ |
| Detritus remineralisation rate | $\lambda_{det}$ | 0.02 | 0.025 | d$^{-1}$ |
| DOC remineralisation rate | $\lambda_{DOC}$ | 0.003 | 0.004 | d$^{-1}$ |
| Opal to phosphorus uptake ratio | $R_{Si:P}$ | 25 | 30 | mol Si mol P$^{-1}$ |

**Table 3.** Parameter values adopted for the prognostic sinking speed scheme.

| Parameter | Symbol | Value | Unit |
|---|---|---|---|
| Minimum particle size | $l$ | 0.002 | cm |
| Maximum particle size | $L$ | 0.5 | cm |
| Mass of a single cell | $C_l$ | 0.0915 | nmol C |
| Sinking speed of single cell | $w_l$ | 1.4 | md$^{-1}$ |
| Exponent of diameter-velocity relation | $\eta$ | 0.62 | – |
| Exponent of diameter-mass relation | $\zeta$ | 1.62 | – |
| Shear parameter | Sh | 0.75 | d$^{-1}$ |
| Stickiness of particles | $S$ | 0.25 | – |

**Table 4.** Primary production and export production averaged over the years 2003 to 2012 simulated by the three model versions/configurations.

|  | Mv1 | Mv1.2 | Lv1.2 | unit |
|---|---|---|---|---|
| Primary production | 43.3 | 35.1 | 29.6 | $Pg\,C\,yr^{-1}$ |
| POC export | 8.8 | 7.1 | 6.1 | $Pg\,C\,yr^{-1}$ |
| $CaCO_3$ export | 0.62 | 0.56 | 0.47 | $Pg\,C\,yr^{-1}$ |
| $CaCO_3$ export to POC export ratio | 7.1 | 7.9 | 7.8 | % |
| Opal Export | 120.5 | 110.8 | 94.8 | $Tmol\,Si\,yr^{-1}$ |

**Table 5.** Estimates of the ocean carbon sink, accumulated carbon uptake, and anthropogenic DIC simulated by the three model versions/configurations. Numbers in parentheses are results from the model forced with the CORE-IAF forcing. The second to last column gives corresponding observation based estimates.

|  | year | Mv1 | Mv1.2 | Lv1.2 | Obs | unit |
|---|---|---|---|---|---|---|
| C sink | 1990–1999 | 2.58 (2.50) | 2.33 (2.46) | 2.01 (2.24) | $2.0 \pm 0.44$[1] | $Pg\,C\,yr^{-1}$ |
| acc. C uptake | 1959–2007 | 102 (99) | 93.0 (95.6) | 83.7 (88.4) | — | Pg C |
| $DIC_{ant}$ | 1994 | 141 (139) | 133 (134) | 122 (123) | $118 \pm 19$[2] | Pg C |
| $DIC_{ant}$ | 2010 | 186 | 175 | 159 | $155 \pm 31$[3] | Pg C |

[1] This is the weighted (by uncertainty) mean of the estimates of McNeil et al. (2003) ,Manning and Keeling (2006),and Mikaloff Fletcher et al. (2006). The given uncertainty is the root mean square of the individual error estimates.
[2] Sabine et al. (2004)
[3] Khatiwala et al. (2013)

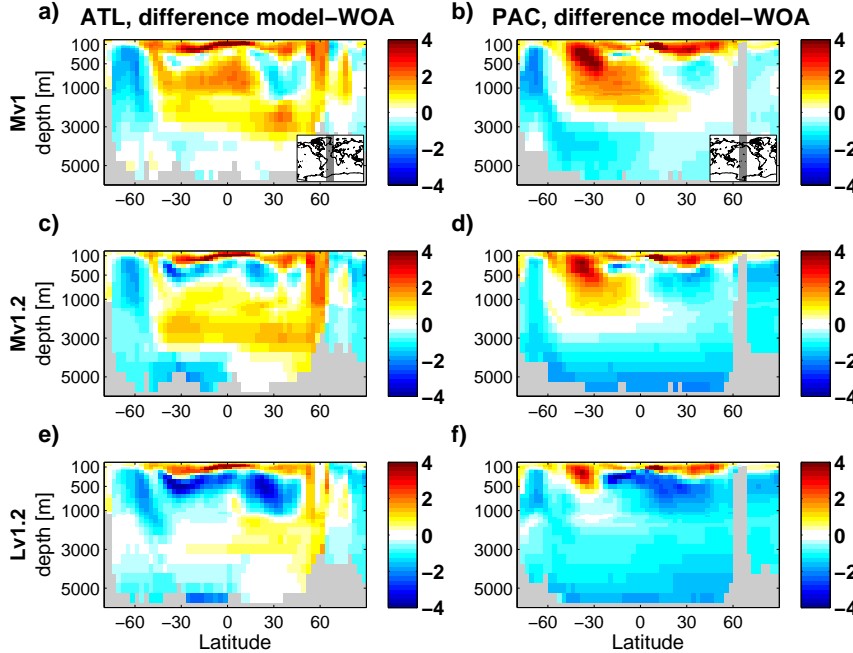

**Figure 1.** Temperature difference model-WOA averaged over 1965–2007 along zonal mean sections through the Atlantic/Southern Ocean **(a, c, e)** and the Pacific/Southern Ocean **(b, d, f)** for the model configurations Mv1 **(a, b)**, Mv1.2 **(c, d)**, and Lv1.2 **(e** and **f)**. The regions covered by the zonal mean calculations are indicated by the grey shaded area in the insets of panels **(a)** and **(b)**.

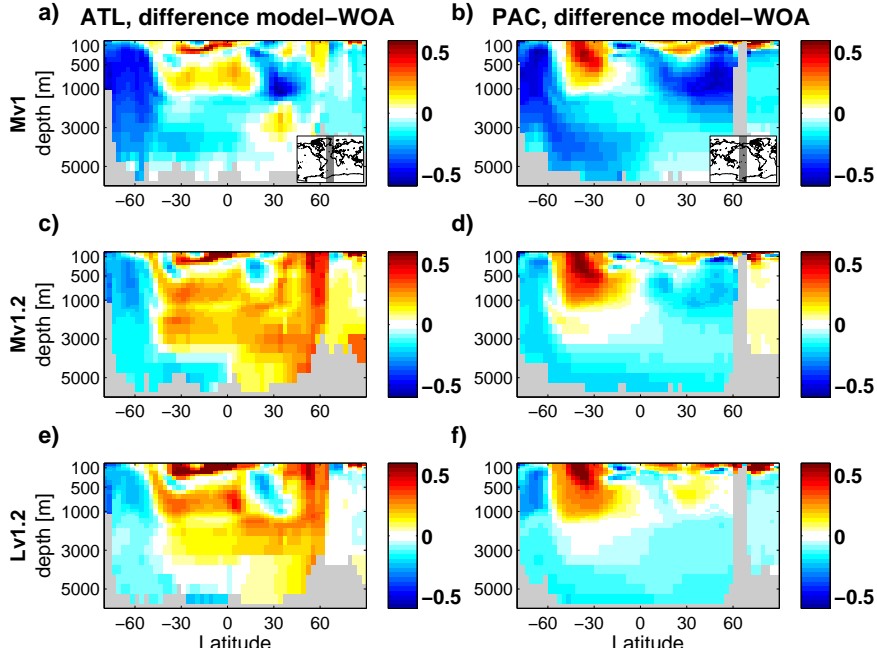

**Figure 2.** As Fig. 1 but for salinity.

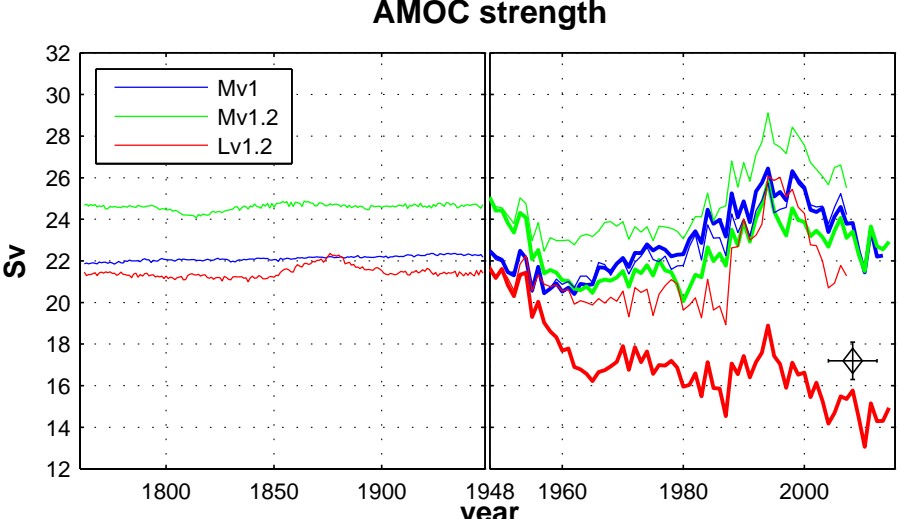

**Figure 3.** Atlantic Meridional Overturning Circulation (AMOC) measured as the maximum of the meridional stream-function at 26.5° N for Mv1 (blue lines), Mv1.2 (green), and Lv1.2 (red). The left panel shows the AMOC for the period 1762–1947 during which the model is forced with the CORE normal year forcing. In the right panel the years 1948–2014 are shown, and thick (thin) lines indicate the overturning simulated with the model forced by the NCEP-C-IAF (CORE-IAF) data set. The black diamond with error bars indicates the observational estimate provided by McCarthy et al. (2015).

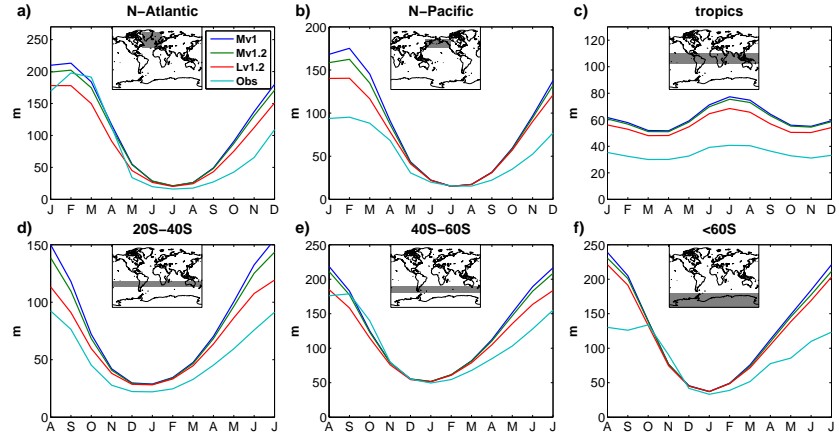

**Figure 4.** Mean seasonal cycle of mixed layer depth over the years 1961–2008 for **(a)** North Atlantic, **(b)** North Pacific, **(c)** tropics (20° S to 20° N), **(d)** southern subtropics (20 to 40° S), **(e)** latitudes between 40 to 60° S, and **(f)** Southern Ocean south of 60° S. Shown are results for the model configurations Mv1 (dark blue), Mv1.2 (green), and Lv1.2 (red). The light blue line is the observation based climatology of de Boyer Montégut et al. (2004), which uses a density threshold criterion of 0.03 kg m$^{-3}$ to define MLD.

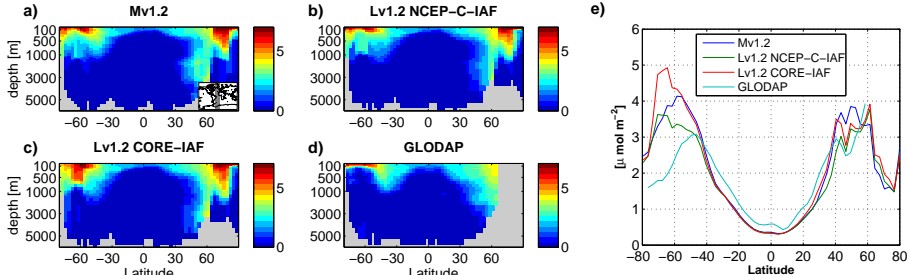

**Figure 5.** Zonal mean CFC-11 concentrations $(\text{nmol m}^{-3})$ along a section through the Atlantic/Southern Ocean (as indicated by the grey shaded area in panel **a**) averaged over the years 1988 to 1998 for the model configurations **(a)** Mv1.2, **(b)** Lv1.2 forced with NECP-C-IAF, **(c)** Lv1.2 forced with CORE-IAF, and **(d)** CFC-11 concentration taken from the GLODAP gridded data set. Panel **(e)** displays the global zonal mean CFC-11 column content $(\mu\text{mol m}^{-2})$.

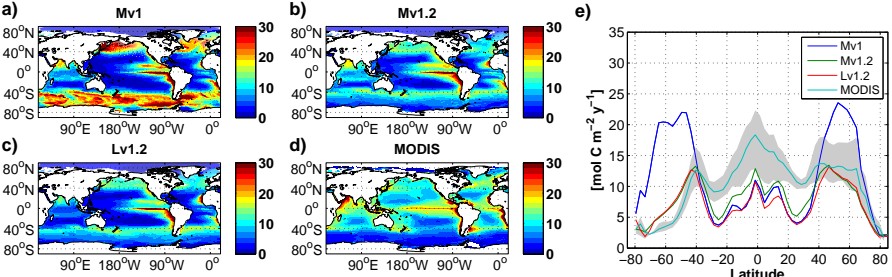

**Figure 6.** Vertically integrated primary production (mol C m$^{-2}$ yr$^{-1}$) averaged over the years 2003–2012 for the model configurations **(a)** Mv1, **(b)** Mv1.2, **(c)** Lv1.2, and **(d)** the mean of three satellite based climatologies (derived from MODIS retrievals, see text). Panel **(e)** displays the zonal means of each field presented in panels **(a–d)**. The grey shaded area represents the range of the zonal means of the three satellite based climatologies.

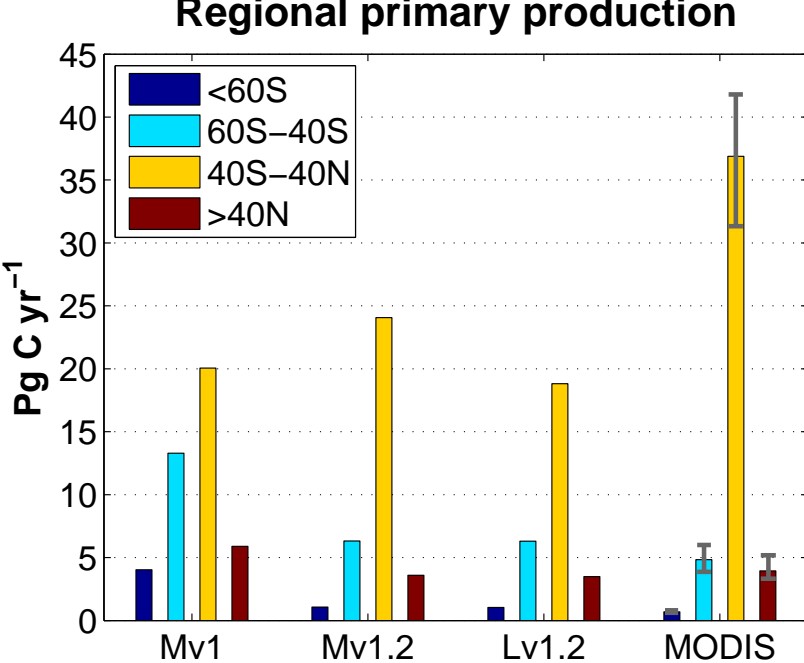

**Figure 7.** Mean values of total primary production over the years 2003–2012 simulated by the three model configurations and an estimate of PP based on satellite data (mean of three climatologies derived from MODIS retrievals, see text) for the regions indicated in the inset. The grey bars indicate the range of observation based estimates.

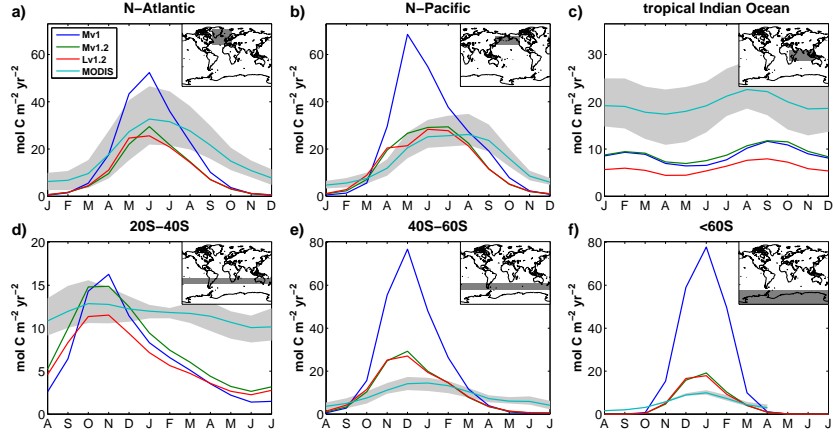

**Figure 8.** Mean seasonal cycle of vertically integrated primary production over the years 2003–2012 for **(a)** North Atlantic, **(b)** North Pacific, **(c)** tropical Indian Ocean (Indian Ocean between 20° S to 20° N), **(d)** southern subtropics (40 to 20° S), **(e)** latitudes between 60 to 40° S, and **(f)** Southern Ocean south of 60° S. Shown are the results for model configuration Mv1 (dark blue), Mv1.2 (green), and Lv1.2 (red) and the mean and range of the satellite derived seasonal cycle (light blue and grey shaded area).

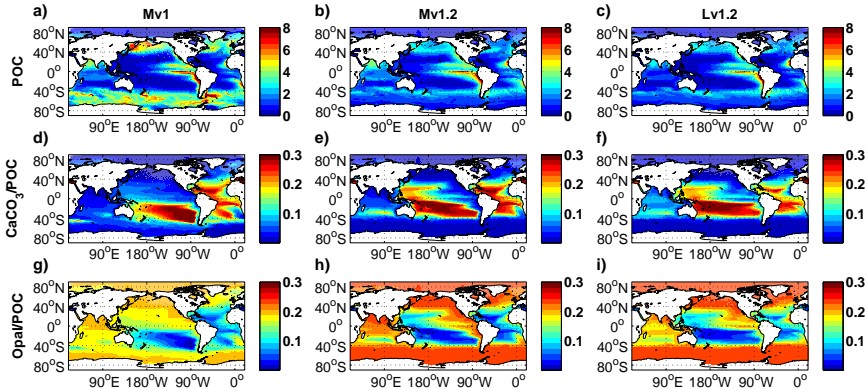

**Figure 9.** Export production of POC averaged over the years 2003–2012 for the model configurations **(a)** Mv1, **(b)** Mv1.2, **(c)** Lv1.2; corresponding $CaCO_3$ to organic carbon ratio in exported matter for **(d)** Mv1, **(e)** Mv1.2, **(f)** Lv1.2; corresponding opal to organic carbon ratio in exported matter for **(g)** Mv1, **(h)** Mv1.2, **(i)** Lv1.2.

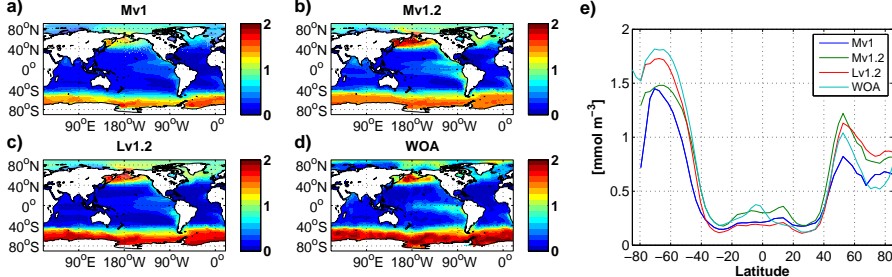

**Figure 10.** Surface phosphate concentration ($\mathrm{mmol\,m^{-3}}$) averaged over the years 1965 to 2007 for the model configurations **(a)** Mv1, **(b)** Mv1.2, **(c)** Lv1.2, and **(d)** surface $PO_4$ concentration taken from the World Ocean Atlas gridded data set. Panel **(e)** displays the zonal mean of each field presented in panels **(a–d)**.

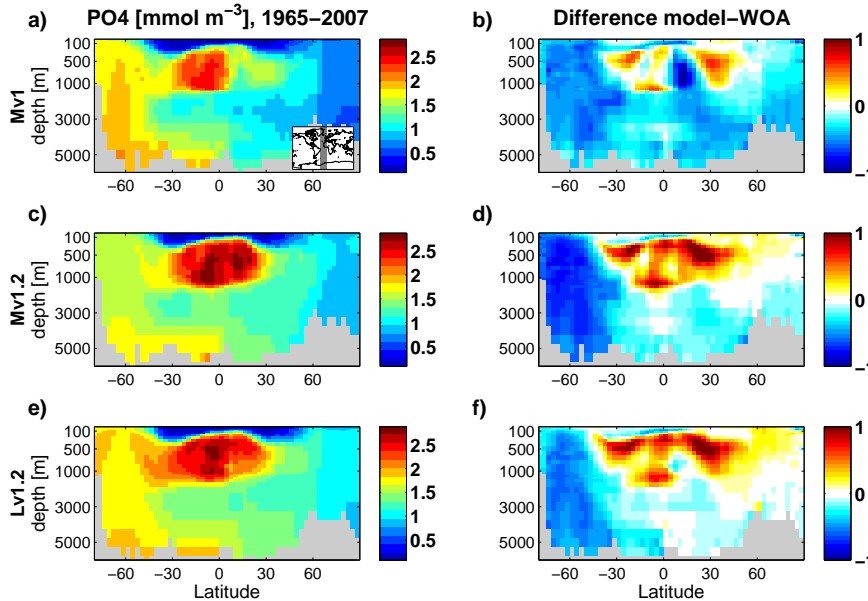

**Figure 11.** Zonal mean phosphate concentration ($\mathrm{mmol\,m^{-3}}$) along the grey shaded region in the Atlantic/Southern Ocean indicated in the inset in panel **(a)** averaged over the years 1965 to 2007 for the model configurations **(a)** Mv1, **(c)** Mv1.2, and **(e)** Lv1.2. Panels **(b)**, **(d)**, and **(f)** show the differences with respect to the World Ocean Atlas gridded data set.

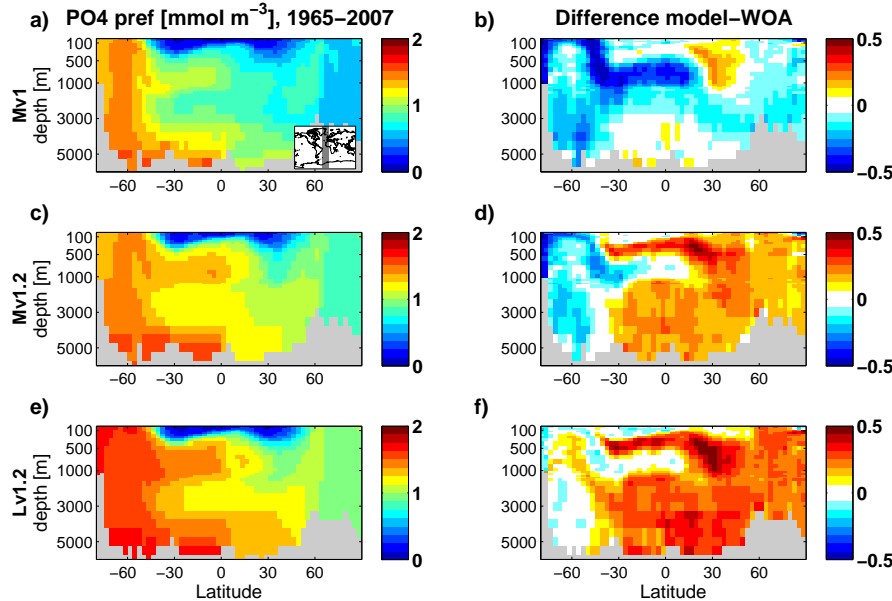

**Figure 12.** As Fig. 11 but for preformed phosphate ($mmol\,m^{-3}$). The remineralised fraction of phosphate is approximated by apparent oxygen utilisation (AOU) divided by the stoichiometric oxygen to phosphorus ratio $R_{O_2:P} = 172$ used in the model.

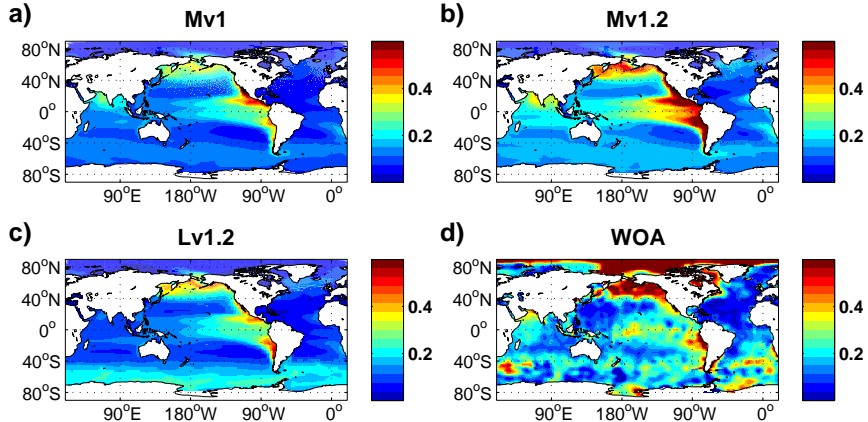

**Figure 13.** Difference $[PO_4] - [NO_3]/16$ (mmol m$^{-3}$) at the surface averaged over the years 1965 to 2007 for the model configurations **(a)** Mv1, **(b)** Mv1.2, **(c)** Lv1.2, and **(d)** surface $[PO_4] - [NO_3]/16$ as calculated from the World Ocean Atlas gridded data set.

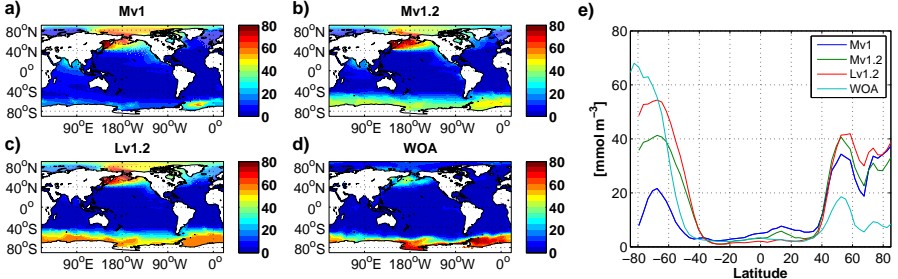

**Figure 14.** As Fig. 10 but for surface silicate concentrations (mmol m$^{-3}$).

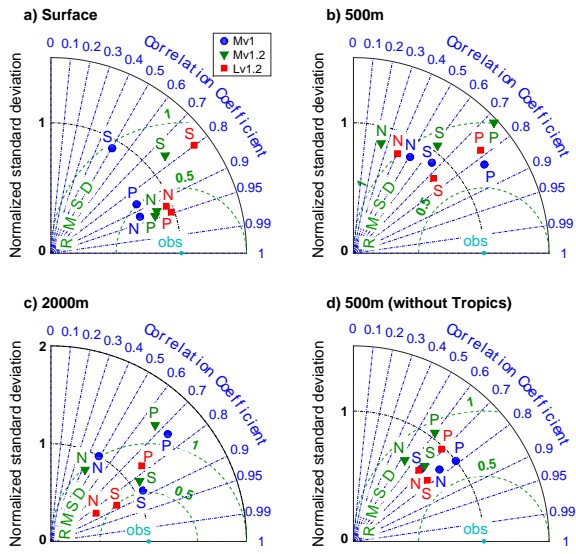

**Figure 15.** Taylor diagrams for phosphate (P), nitrate (N), and Silicate (S) for model configurations Mv1 (blue circles), Mv1.2 (green triangles), and Lv1.2 (red squares) at **(a)** surface, **(b)** 500 m depth, **(c)** 2000 m depth. Panel **(d)** shows results for the 500 m depth level with error prone grid points located in the tropics (between 20° N and 20° S) omitted from the analysis. Observations are the objectively analysed climatologies from WOA 2009 (Garcia et al., 2010b). Model fields have been averaged over the time period during which the bulk of observations in WOA 2009 has been acquired (1965 to 2007).

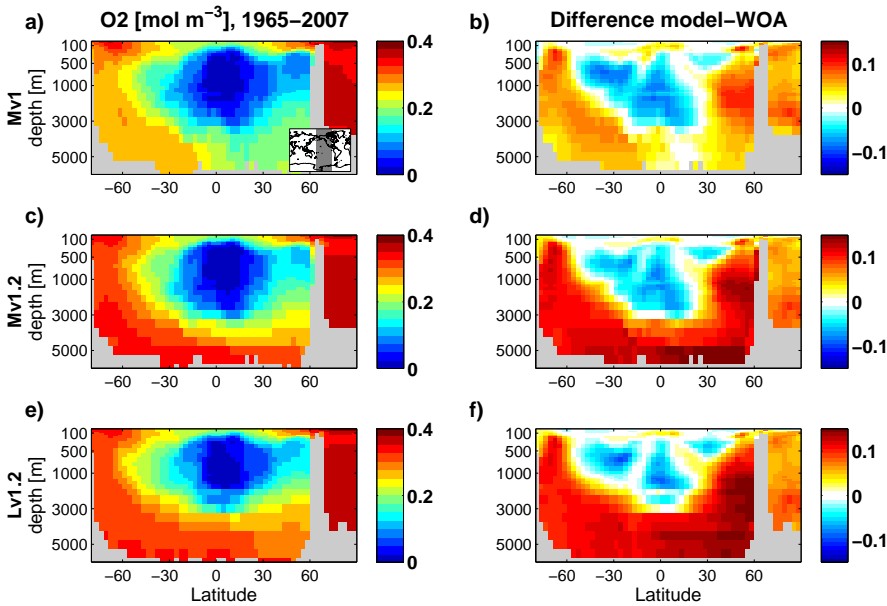

**Figure 16.** As Fig. 11, but for oxygen $(\mathrm{mol\,m^{-3}})$ along the grey shaded region in the Pacific/Southern Ocean indicated in the inset in panel **(a)**.

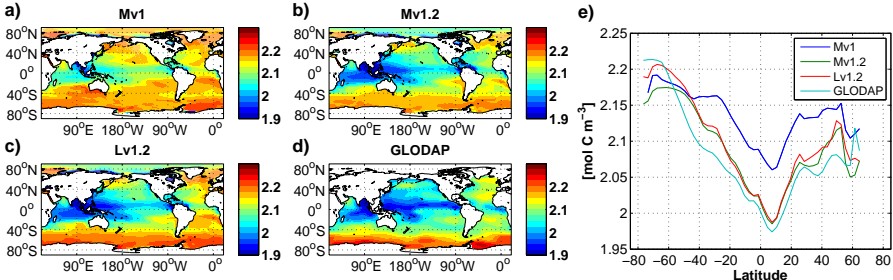

**Figure 17.** Surface concentration of dissolved inorganic carbon (mol m$^{-3}$) averaged over the years 1988 to 1998 for the model configurations **(a)** Mv1, **(b)** Mv1.2, **(c)** Lv1.2, and **(d)** corresponding surface DIC from the observation based GLODAP climatology. Panel **(e)** displays the zonal mean of each field presented in panels **(a–d)**.

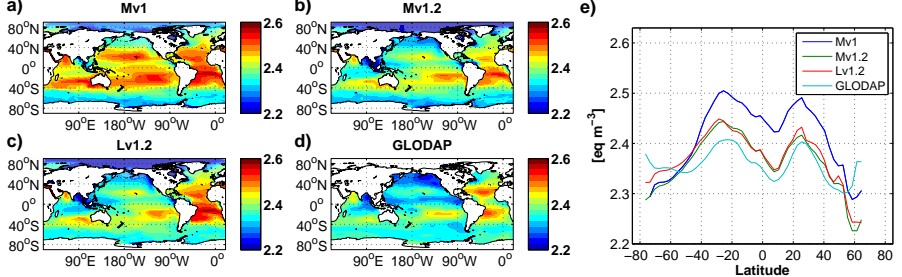

**Figure 18.** As Fig. 17 but for surface total alkalinity (eq m$^{-3}$).

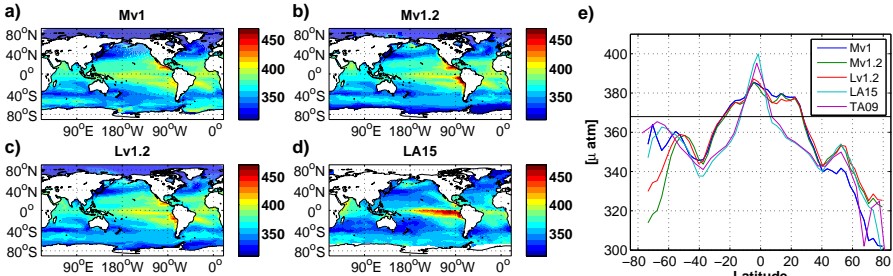

**Figure 19.** Surface partial pressure of $CO_2$ (µatm) averaged over the years 1995 to 2005 for the model configurations **(a)** Mv1, **(b)** Mv1.2, **(c)** Lv1.2, and **(d)** the observation based surface $pCO_2$ data set of Landschützer et al. (2015a). Panel **(e)** displays the zonal mean of each field presented in panels **(a–d)**, and additionally the zonal mean of the data product by Takahashi et al. (2009). The black line indicates the mean atmospheric $pCO_2$ over the averaging period.

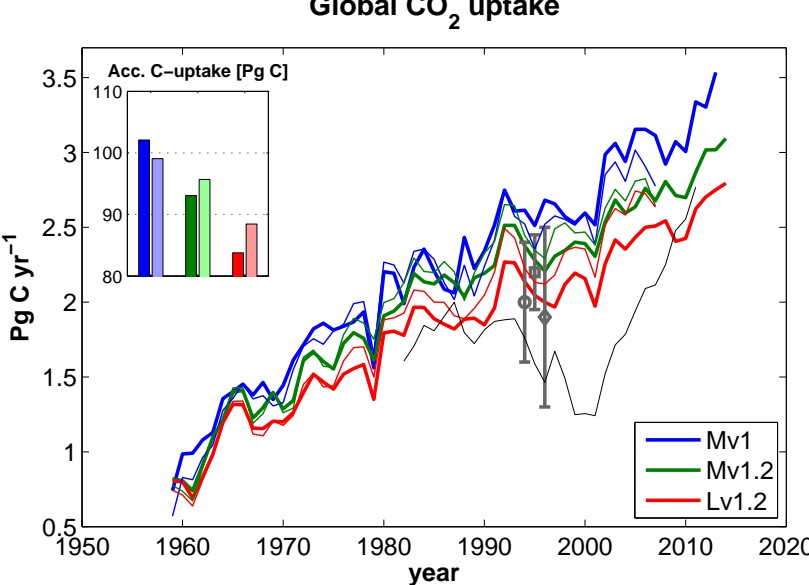

**Figure 20.** Globally integrated annual ocean carbon sink flux 1959–2014 for Mv1 (blue lines), Mv1.2 (green), and Lv1.2 (red). Thick (thin) lines indicate carbon fluxes simulated with the model forced by the NCEP-C-IAF (CORE-IAF) data set. The total integrated flux from 1959 to 2007 is given in the inset, where the paler shading indicates results for simulations with CORE-IAF. The thin black line is the carbon flux estimate by Landschützer et al. (2015a) assuming a constant $0.45\,\mathrm{Pg\,yr^{-1}}$ contribution of riverine outgassing flux as in Le Quéré et al. (2015). Symbols with error bars are the carbon uptake estimates by McNeil et al. (2003) (circle), Mikaloff Fletcher et al. (2006) (square), and Manning and Keeling (2006) (diamond). Note that the estimates of McNeil et al. (2003) and Manning and Keeling (2006) are mean uptake values over the 1990s.

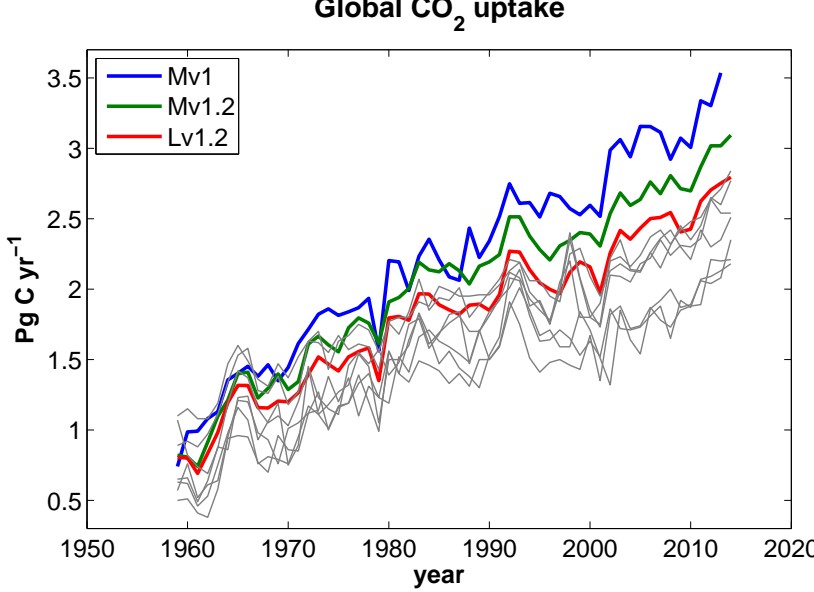

**Figure 21.** Globally integrated annual ocean carbon sink flux 1959–2014 for Mv1 (blue line), Mv1.2 (green), and Lv1.2 (red) forced by the NCEP-C-IAF data set. Thin grey lines indicate the ocean carbon sink fluxes calculated by the models used in Le Quéré et al. (2015): NEMO-PlankTOM5 (Buitenhuis et al., 2010), NEMO-PISCES (LSCE version, Aumont and Bopp, 2006), CCSM-BEC (Doney et al., 2009), MPIOM-HAMOCC (Ilyina et al., 2013), NEMO-PISCES (CNRM version, Séférian et al., 2013), CSIRO (Oke et al., 2013), and MITgcm-REcoM2 (Hauck et al., 2013).

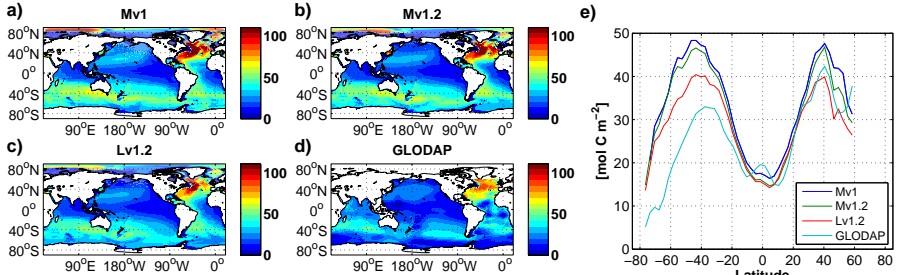

**Figure 22.** Vertically integrated anthropogenic carbon ($mol\,m^{-2}$) for the year 1994 for the model configurations **(a)** Mv1, **(b)** Mv1.2, **(c)** Lv1.2, and **(d)** corresponding $DIC_{ant}$ from the observation based GLODAP climatology (Sabine et al., 2004). Panel **(e)** displays the zonal mean of each field presented in panels **(a–d)**.

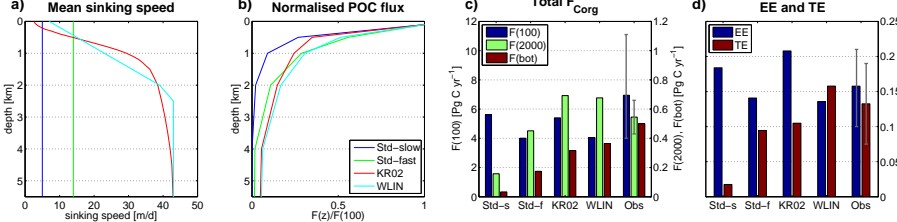

**Figure 23. (a)** Mean profiles of sinking speed with depth for the four experiments STD-slow (dark blue), STD-fast (green), KR02 (red), and WLIN (light blue); **(b)** as panel a, but for mean profiles of POC fluxes (averaged over areas with water depth >4000 m) normalised to the flux at 100 m depth; **(c)** globally accumulated fluxes of particulate organic carbon (POC) through 100 m depth (dark blue), through 2000 m depth (green), and the downward POC flux at the ocean bottom for depths larger than 1000 m (brown); **(d)** export efficiency (EE, dark blue) and transfer efficiency (TE, brown). The observation based estimates ("Obs") in panels **(c)** and **(d)** are derived from Henson et al. (2012), Honjo et al. (2008), and Laws et al. (2000) for POC flux at 100 m and EE, from Henson et al. (2012) and Honjo et al. (2008) for POC flux at 2000 m and TE, and from Seiter et al. (2005) for the bottom POC flux. The coloured bars indicate the mean of the observation based estimates while the grey error bars give the maximum and minimum estimate. Note that the Honjo et al. (2008) EE and TE are calculated as ratios of total PP, total export and total flux at 2000 m and not as the mean of gridded EE and TE values as the rest of the values presented here. We include these data nevertheless since the differences between the two methods of calculation are small compared to the spread of the observation based data.

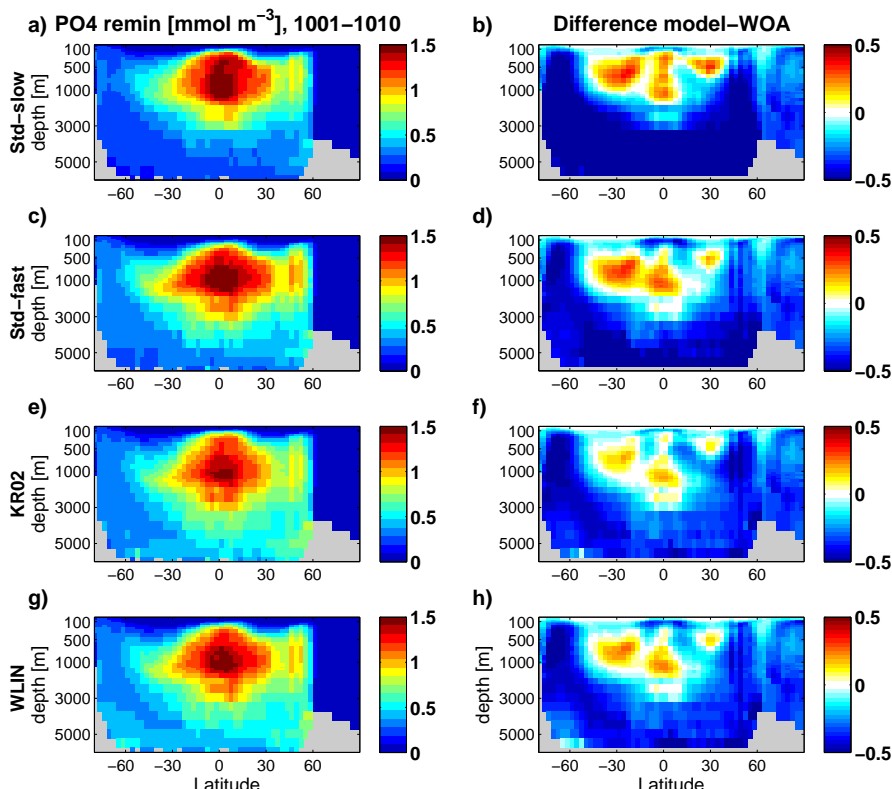

**Figure 24.** Global zonal mean concentrations of remineralised phosphate ($\mathrm{mmol\,m^{-3}}$) averaged over the last 10 years of the spin-up simulations employing the **(a)** STD-slow, **(c)** STD-fast, **(e)** KR02, and **(g)** WLIN particle sinking parameterisations. Panels **(b, d, f, h)** show the differences with respect to the World Ocean Atlas gridded data set. The remineralised fraction of phosphate is approximated by apparent oxygen utilisation (AOU) divided by the stoichiometric oxygen to phosphorus ratio $R_{\mathrm{O_2:P}} = 172$ used in the model.

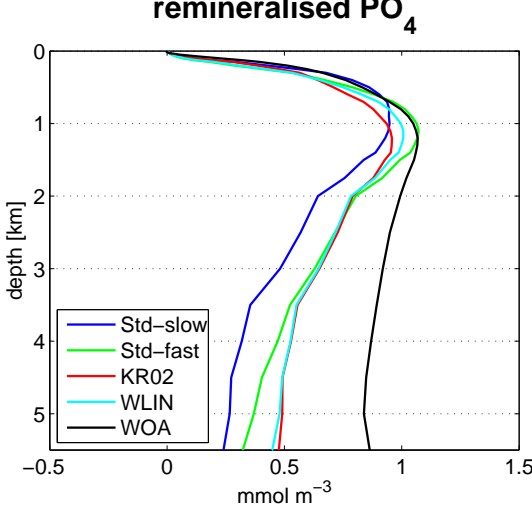

**Figure 25.** Global mean profiles of remineralised phosphate ($mmol\,m^{-3}$) averaged over the last 10 years of the spin-up simulations employing the STD-slow (dark blue), STD-fast (green), KR02 (red), and WLIN (light blue) particle sinking parameterisations. The remineralised fraction of phosphate is approximated by apparent oxygen utilisation (AOU) divided by the stoichiometric oxygen to phosphorus ratio $R_{O_2:P} = 172$ used in the model.

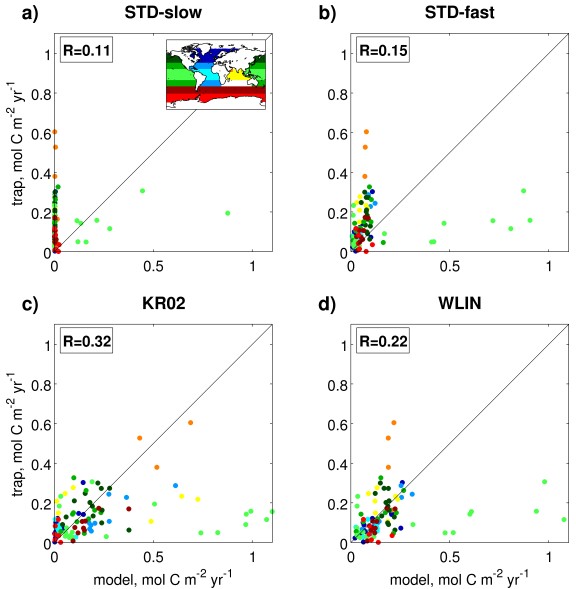

**Figure 26.** Scatter plots of modelled vs. observed POC fluxes at 2000 m depth for the POC sinking parameterisations **(a)** STD-slow, **(b)** STD-fast, **(c)** KR02, and **(d)** WLIN. Observed data are taken from a synthesis of sediment trap POC fluxes compiled by Honjo et al. (2008). Data from several stations are averaged if they are located in the same model grid cell. Stations are discarded if the depth in the corresponding model grid cell is shallower than 2000 m. Model-data pairs are colour coded according to ocean basin and latitudinal range: Atlantic north of 40° N (dark blue), subtropical Atlantic (blue), tropical Atlantic (light blue), Pacific north of 40° N (dark green), subtropical Pacific (green), tropical Pacific (light green), tropical Indian Ocean (yellow), Arabian Sea (orange), Southern Ocean between 40 and 60° S (brown), and Southern Ocean south of 60° S (red). Here, the term subtropical refers to latitudes between 20 and 40° N as well as 20 and 40° S, and the term tropical refers to latitudes between 20° N and 20° S. Colour codes are also shown in the inset in panel a.