# Peer review of "Evaluation of NorESM-OC (versions 1 and 1.2), the ocean carbon-cycle stand-alone configuration of the Norwegian Earth System Model (NorESM1)"

_Geoscientific Model Development, 2015_

## Referee Comment (RC1) · Anonymous Referee #1 · 24 Feb 2016

Overview

This manuscript presents a description of an updated version of a previously published earth system model. The focus here is on the ocean physics and marine biogeochemistry components, and the work presented uses these in forced mode under reanalysis atmospheric forcing. The configurations presented span old to new, and cover resolution differences, physics updates and biogeochemistry tuning and updates. Evaluation is performed for select fields between the model configurations and observational climatologies. The manuscript includes an evaluation of four schemes for the water col-

umn remineralisation of export production, though similarities in performance preclude a definitive selection.

Summary

In general, the manuscript does well at describing the model and outlining its performance. I have a number of relatively minor specific comments dotted throughout the manuscript (see below). My only major comment would be that, while intercomparing the various versions of the model, the manuscript does nothing to contextualise the performance of NorESM-OC within the context of other Earth system models. The CMIP5 archive is something of a treasure trove on this point, and most of the evaluations made in the manuscript could be repeated with output from it. However, I would only suggest adding an overview of this (e.g. Taylor diagrams?), and not amending the manuscript throughout – the main task here, and I would agree with the authors on this, is to evaluate performance between traceable versions of NorESM-OC.

My recommendation is publication following addressing of these comments.

—–

Pg. 2, ln. 4: "We present simulation results . . ." – this sentence could perhaps be a little clearer on how many configurations are examined; the mention of multiple resolutions makes it a little opaque

Pg. 3, ln. 12: The authors simulate the ocean component of an ESM under reanalysis forcing; did they consider running under atmospheric output from the ESM instead?

Pg. 8, ln. 1: "We anticipate . . . of 10,000 years . . ." – would I be correct in assuming this "anticipation" is either too computationally expensive to test, or is not actually possible because v1.2 of the model rolls in several changes and not just this one (i.e. it is not possible to separate differences due to the time-stepping change from those from other sources)?

Pg. 8, ln. 13: "derived by Wanninkhof (1992)" – it might be worth examining Wanninkhof (2014) for updates here; I believe that some of the issues mentioned subsequently are tackled there

Pg. 10, ln. 15: A remark about the shape of the nutrient limitation relationship would be useful; also something about the relationship between nutrients (e.g. Leibig's Law or something else?)

Pg. 10, ln. 23: Presumably this means that opal "production" is associated with export only, and is not representative of all of the opal production by diatoms (some of which is dissolved before it can be exported)?

Pg. 11, ln. 4: "we performed a re-tuning" – by eye?

Pg. 11, ln. 16: Is there a potential problem here because Si:P ratios in the real world are not constant?; in iron-limited HNLC regions, for instance, Si:P is typically elevated (since diatom cell cycle length is increased affording more time to uptake silicon); the global nature of the "solution" here to DIC / TA biases potentially promises trouble regionally

Pg. 14, ln. 16: It would perhaps be helpful to illustrate to the reader the depth distribution of sinking particles, as well as remineralisation, produced by these different schemes (e.g. for the same quantity of POC at 100m, what's its fate down a static water column?)

Pg. 16, ln. 10: "... a restoring time scale of 365 days (version 1) and 350 days (version 1.2) ..." – any reason why this is 15 days shorter?; presumably a parameterisation oversight?

Pg. 17, ln. 13: "with only small trends of 0.00007, 0.021, and $-0.048$ PgCyr$-1$ century$-1$ for Mv1, Mv1.2, and 15 Lv1.2, respectively" – any explanation for why the longer duration simulations with v1.2 have markedly higher CO2 trends than the shorter duration v1 model?; one might instinctively expect smaller values for longer simulations; perhaps plot up the net CO2 flux with time?

Pg. 18, ln. 8: "Since the unbalanced salinity relaxation flux removes salt . . ." - this sentence reads as if it is saying that the reduction in S in Mv1 is due to the relaxation flux being applied across *all* model configurations; I think this sentence and the preceding one should be combined into: "While all three configurations include salinity relaxation, this is not balanced in the case of Mv1, with the result that average salinity falls by 0.2 psu during the course of the integration."

Pg. 18, ln. 14-16: Worth reporting these sorts of numbers in a table?

Pg. 19, ln. 12: ". . .a long transient increase in strength for about 300 years . . ." – Any chance of including a plot of the AMOC strengths of the models from their spin-up phases?; not least to give some idea of interannual variability in the absence of interannual forcing variability (if any)

Pg. 20, ln. 15: Since the plot makes a point of examining the time-series of AMOC, would it be possible to present the RAPID estimates on the same plot?

Pg. 20, ln. 19: ". . . the climatology of de Boyer Montégut et al. (2004)" – This is calculated how?

Pg. 20, ln. 22: On a related point, are model MLDs comparable to the climatology?; e.g. could MLD be calculated for the model in the same way as it's done for the observation-based climatology?; if this is already the case, please make this clear

Pg. 21, ln. 12: "clearly indicates a too deep mixing" – Grammar

Pg. 21, ln. 13-16: Does this have anything to do with sea-ice?; some models can exhibit large polynas in the SO, with the result that mixing, and ventilation, can be extreme

Pg. 22, ln. 5-7: "We note that the Eppley-VGPM algorithm produces global PP estimates at about the mean value . . ." – This might well be true, but in my experience the spatial patterns of different estimates are wildly different, making the choice of such an "intermediate" product less clear

Pg. 22, ln. 12-14: "These large discrepancies are reduced in model version 1.2 . . ." – Is it possible (e.g. via run models that are not shown here) to be sure that the improvements stem from the BGC changes as opposed to the physics changes?

Pg. 23, ln. 5-6: "This missing PP on the shelves . . ." – You could make this clearer by calculating PP in open ocean areas only for VGPM and the model runs

Pg. 23, ln. 18-26: Is it possible to determine a map of nutrient limitations from the model?; it might help diagnose another reason for differences between them

Pg. 24, section 3.3: BGC tracers are the core of the model, while production is just one process within the model; I'd suggest swapping the sections around and making this 3.2; production could come just before export - which is arguably more natural anyway

Pg. 24, section 3.3: Why no chlorophyll?; is this because the fixed chl:C ratio here causes problems?

Pg. 25, ln. 16: "Moreover, nitrogen fixation, . . ." – A map of this perhaps?

Pg. 25, ln. 17: ". . .which occurs in the surface ocean as soon as [NO3] < RN:P[PO4], . . ." – Does this mean that all N2-fixation occurs in the right place?; cf. ostensible temperature limits, etc.

Pg. 26, ln. 2-6: What do the distributions of biogenic opal and CaCO3 export look like?; and how do they compare to observationally-derived estimates (e.g. in total)?

Pg. 27, ln. 15: While iron isn't quite at the stage of having a global observational climatology, Geotraces has some fields that might help; and even in the absence of an observational comparison, it might be helpful to compare the models to elucidate differences; iron's reach is longer than simply total iron concentration

Pg. 28, ln. 22: ". . .results in increased CaCO3 production and a considerable reduction of the alkalinity and DIC biases . . ." – Can this be squared with any observational evidence?; it can certainly be squared with model evidence (cf. Kwiatkowski et al., 2014;

here, a number of models have low CaCO3 production in the tropics and excessive alkalinity and DIC)

Pg. 30, ln. 25: "...the GLODAP data base" – The Khatiwala et al. (2013) estimate of anthropogenic CO2 is probably a better estimate

Pg. 31, ln. 16: As mentioned previously, having a figure that illustrates how each of these schemes remineralises organic matter down the water column would also be helpful (or a plot of how the OM is attenuated)

Pg. 31, ln. 28: "At the end of the spin-up runs ... " –How do these fit with the long spin-ups already done?; also, are these long enough to approach equilibrium?; or is the assumption that they are long enough for only transient drift to remain?

Pg. 33-34: I don't know the answer myself, so it's perhaps cheeky to ask, but could the authors comment on whether direct POC flux measurements or indirect AOU (or other tracer) measurements better constrain export and remineralisation; there may be no good answer at the moment, so the authors' use of both is probably best

Pg. 35, ln. 16: A more general comment – some studies (e.g. Kwon et al., 2009; Kriest, Oschlies & Khatiwala, 2012) examine the tuning of such models of export, whereas the manuscript uses them "as is"; while the authors do mention alternative sinking velocities for STD-fast at one point, they could help here by drawing further attention to this and / or commenting on the tuning of such models (e.g. if they have any unreported experience on the success or otherwise of this)

Pg. 36, ln. 10: So, paradoxically, excessively high and excessively low O2?

Pg. 37, ln. 18: "Part of this problem is the distribution of primary production which is too high in a narrow band along the upwelling ..." – Is this in any way related to the model being an isopycnal model?

Pg. 38, ln. 1: "In the Southern Ocean ..." – Since this paragraph deals with the model as it is, rather than - per the preceding paragraph - the model as it might become, it

should precede the paragraph on shelf improvements

Pg. 38, ln. 9: It's a wholly personal preference, but I think papers finish better with a short, bulletpointed list of the main points / findings

Pg. 38, ln. 9: A general criticism I'd make of the manuscript's validation of the model is that the performance of the model is not properly put within the context of similar models; the CMIP5 archive, for instance, offers a range of similar resolution models that could profitably provide such context; most of which aren't isopycnal models

Table 1: "kmol m−3" – At the risk of both being a pedant and missing the wood for the trees, presumably kmol / m3 is being used here because it is equivalent to molar units (i.e. mol / l); if so, why not just use mol / l?

Table 1: "Laughing gas" – While - appropriately enough - I laughed when I saw this, I don't think it can be called this in the final manuscript; nitrous oxide, perhaps?

Table 2: "Fraction of grazing egested 1−ezoo" – This is a little bit confusing; is the symbol really "1 - ezoo"?; why not "ezoo" and give it a value of 0.2 or 0.1?

Figure 1: Sometimes putting the y-axis on a logarithmic scale is helpful for showing what's happening near-surface

Figure 1: Since the colour map includes white, these panels could do with having the seafloor drawn on; that would help separate places that have zero difference from those that are rock; also, this would help clarify the bathymetry differences between different model grids

Figure 3: What happens with AMOC during the long spin-ups?

Figure 3: "109 kg s-1" – Convert to Sverdrups?; or is this awkward for an isopycnal model?

Figure 3: Observational data from RAPID appropriate for comparison?

Figure 4: "The range given for the observation based estimates is solely due to different criteria used to define MLD and not due to other uncertainties" – Per a previous comment, how does the MLD method used for the models compare to that of the observations?; also, given how variable different MLD methods can be from one another, reporting uncertainty in this way here seems potentially risky

Figure 5: Again, a log-scale y-axis might help here; most of the structure here is in the upper water column

Figure 5: Rotate panel e so that its y-axis is aligned with the x-axes on the panels to the left?; i.e. 90S to the left, 90N to the right

Figure 6: It's a weakness on my part, but I prefer my plots to omit unnecessary grid lines (and have coastlines if possible)

Figure 6: Rotation would make panel e easier to understand (though I appreciate it would not then be aligned as in Figure 5)

Figure 6: Rather than only use VGPM, you could average VGPM with other estimates; it's a poor way of simplifying the diversity in observational estimates of PP, but it can be useful given their spread, and it's not without precedent

Figure 7: Why 40 in one hemisphere and 60 in the other?

Figure 7: Again, why just use VGPM?

Figure 8: In panel c, including the Indian Ocean is complicated by the presence of the monsoon

Figure 9: Rotate panel e again please

Figure 10: Fewer colours in the colour maps here (especially for the delta plots) might make it easier to discern patterns in match-mismatch (e.g. the reds are quite homogeneous)

Figure 12: These panels hint at some odd ventilation feature that elevates N. Pac. silicic acid (which then bleeds into the Arctic); does the same appear in CFC-11?

Figure 13: Add a key if possible; also, different symbols might help with the plot (especially for colour blind readers)

Figure 13: Amend to "Panel (d) shows results for the 500m depth level with error prone grid points located in the tropics (between 20 and 20S) omitted from the analysis"?

Figure 14: Why not go entirely east-west in the Pacific here?; since oxygen is particularly low in the East Pacific, this could be important

Figure 16: Nicely improved!

Figure 17: Uncertainty from Takahashi (and / or other pCO2 products)?

Figure 18: There's quite a bit of a dip in the observationally-derived fluxes; is its origin explained in the main text?

Figure 19: "averaged over the years 1990 to 1998" – Why not a full decade?; being unreasonably suspicious, such odd ranges always raise my eyebrows

Figure 20: The mismatched sizes of bars shown in panel b seem a bad idea; log scale again?

Figure 21: Why zero in the Arctic of panels a, c, e and g?; does the main text say?

Figure 21: A global integrated profile plot of a, c, e and g might be helpful

Figure 22: Bigger dots?; also, a key would be nice; perhaps a little map that shows the regions in the appropriate colour?

---

## Referee Comment (RC2) · Anonymous Referee #2 · 1 Apr 2016

**General comments**

Appreciation of the manuscript

J. Schwinger and co-authors present NorESM-OC, a standalone version of the three-dimensional ocean carbon cycle sub-model of the Norwegian Earth System Model (NorESM1). Two versions are described: NorESM-OC1, the version actually included in NorESM1, and an updated one, NorESM-OC1.2, which can be used on finer grids,

and which includes two newly implemented parametric representations of the particulate transport in the water column.

The paper falls well within the scope of Geoscientific Model Development. It is overall well structured, although in detail, information can not always be found where one would expect to find it (some parameter values are only reported in the discussion section, instead of the model description part). The English language is generally good. There remain a few minor shortcomings that can nevertheless be easily fixed.

The presentation and description of the model is rather complete, the discussion of the results is to some extent merciless: the authors put much emphasis on the biases in their model results. I would be pleased to read a few more sentences about the strengths of their model.

Although I have not found any major problem in the manuscript, I have come across a series of details that could be better explained. I am confident that the authors will be able to address all of the shortcomings mentioned.

General model presentation: scope

The model is clearly global in scope. Time frames for applications are not clearly stated. The paper presents simulation experiments over a few centuries, following 800 or 1000 year spin-ups. Can it also be reasonably used on longer time-scales (a few thousands to a few tens of thousands of years)?

[Figure]

Code: availablility and quality

The code is not openly available, but only upon signing licence agreements (at least two, one for NorESM, one for HAMOCC, but perhaps even more – this is not clear).

As a reviewer, I was nevertheless granted access to the code. It is written in Fortran (much FORTRAN77-style, with some Fortran 90 elements). A C-style pre-processor is required to compile it. Some files are well commented, others almost devoid of comments. The presence of OpenMP compiler directives suggests that it has been prepared for usage on multi-platform shared memory multiprocessing computing facilities, which is definitely a noteworthy feature of interest to prospective users.

Unfortunately, the manuscript does not provide much information about the code, nor about numerical methods adopted in general (except for time-stepping, which seems to follow a leap-frog scheme in combination with a filter with unspecified characteristics though). These are, however, the informations that are typically expected in a "Development and technical paper" in Geoscientific Model Development. I would recommend to add a short, descriptive paragraph about such aspects, the more since the code is not publicly available. Finally, it would also be interesting to read about typical execution times of the three versions.

**Specific comments**

page 4, line 19: "[. . . ] NorESM-OC1 corresponds to the fully coupled NorESM1-ME [. . . ]" would better read "[. . . ] NorESM-OC1 corresponds to the version included in the fully coupled NorESM1-ME [. . . ]"

Although the paper reads as if NorESM-OC1 was a sub-model isolated from NorESM1, its setup is highly similar to the earlier "isopycnic ocean carbon cycle model" of Assmann et al. (2009). How does NorESM-OC1 actually differ from the model described by Assmann et al. (2009)?

page 7, lines 14–17: I presume that the preformed tracer values are set to the mixed-layer concentrations each time that a water parcel leaves the surface mixed-layer. Please rephrase for clarity.

page 8, line 26 – page 9, line 7: the detailed approximation adopted for TA is given, but it would also be interesting to know how the speciation of the acid-base systems is calculated, or, equivalently, how pH is calculated from the TA expression.

page 12, lines 6–8: I may possibly misinterpret this, but I am wondering how the loss of material (and alkalinity) through the sediment is compensated for, if the model does not take into account the influx of carbon, alkalinity and nutrients? Does it operate in some kind of closed-loop configuration (mass an alkalinity that are lost via the deep-sea sediment are re-injected at the top)?

Please clarify how the global inventories of the model ocean are conserved.

page 13, lines 12–13: while I find it rather obvious how $M$ can be a prognostic variable, I do not immediately see how NOS can be handled as such. Please give a few details about how NOS is predicted.

page 15, beginning of section 2.4.1: I recommend to introduce this section by one or two general sentences that summarize which data are required to force the model, before coming to the specific versions of datasets used.

page 16, line 8: Only the fate of the Antarctic freshwater influx is detailed. How is the rest of the freshwater treated?

[Figure]

page 16, line 8: 365 days for v. 1 and 350 days for v. 1.2: why this difference?

page 17, line 8: is atmospheric $pCO_2$ prescribed or is it prognostic?

page 17, lines 13–14: the carbon flux of 0.26 GtC/yr from Mv1 is about 50 times larger than the fluxes for the other two and not truly negligible (it is of about the same order of magnitude as the atmospheric $CO_2$ consumption rate by continental weathering). Is this correct, and if so, what is the reason? Furthermore, the drifts of Mv1.2 and L1.2 are of the order of 5–10%, which is far from negligible over 100 yr. This may point out significant deviation from the sought near-equilibrium state.

page 19, lines 19–23: "[. . . ] if this trend would be removed.": is it possible to remove it? Where does it actually come from?

page 27, line 26: I find these biases rather strong (50% or so). Comments?

page 32, lines 8, 25 and 27: please provide references for the cited numbers (they can be found in the figure captions, but it would be good to have them in the text as well).

page 36, lines 23–29: It is fairly possible that the new sinking parametrisations are simply more efficient in counterbalancing the effects of the (too?) strong Southern Ocean ventilation. Using a model biogeochemical process to reduce biases arising from shortcomings in the model physics can hardly be considered an improvement.

page 37, line 11: "[. . . ] a long standing problem [. . . ]": please provide a reference to support for this statement.

**Technical comments**

Throughout the paper: please consistently use either "parameterisation/parameterise" or "parametrisation/parametrise" as a spelling

page 2, line 14: "[. . . ] scheme prescribing a linear increase of sinking speed [. . . ]" would better read "[. . . ] scheme that uses a linear increase of the sinking speed [. . . ]"

page 3, lines 15–24: please indicate which versions of CESM, CAM-Oslo, MICOM and HAMOCC are used, respectively.

page 3, line 28: "persue" should read "pursue"

page 4, lines 23–24: It is at this point not entirely clear what "on a numerically more efficient grid in 1° and 2° resolution" means. Based upon what we read later on, this should actually read "on a numerically more efficient grid at either 1° or and 2° resolution". Also: are these actual or nominal resolutions? If they are nominal, what are the extremes?

page 7, sect. 2.3: which version of HAMOCC is finally used? 5 or 5.1? Please check and make sure the text is consistent.

page 7, line 26: "seriously" would better read "significantly"

page 7, line 27: "[. . . ] the tracer transport fully consistent [. . . ]" should read "[. . . ] the tracer transport scheme to make it fully consistent [. . . ]"

page 7, line 28: I suggest to replace "on two time-levels" by "on both time levels of the leap-frog scheme"

page 8, line 6: I suggest to replace "is determined by three components:" by "is controlled by three factors:"

page 9, line 15 – page 10, line 2: please include references for the chosen $\alpha$, $\sigma$, $\mu_{phy}$ and $R_{C:Chl}$ values; I have furthermore not been able to find any quantitative information about $k_w$ and $k_{chl}$ – please amend.

page 9, line 20: "based on" should read "calculated from"

page 10, line 14: "by availability" should read "by the availability"

page 10, line 24: while $K_{Si}$ is mentioned here, together with a reference, we have to proceed to page 26 to find the adopted value (where it is mentioned rather accessorily). Why? Please provide the adopted value here, where one would expect to find it.

page 11, line 14: "remineralisation of detritus" should read "remineralisation rate of detritus"

page 11, line 14: "bias" should read "biases" (there are two of these)

page 11, line 27: "decomposed" should read "dissolved"

page 12, line 22 – page 13, line 15: although it is explained how $A$ and $\epsilon$ are derived from $M$ and NOS, it is not clear how $B$ and $C$ are obtained. Please provide a few words of explanation.

page 13, line 4: "described" should read "written as" or "given by"

page 13, line 7 vs. page 14, line 6: $C$ has a double meaning. Please try to avoid.

page 14, line 9: "would be evaluated" should read "would have to be evaluated"

page 14, line 10: "This, however, is" should read "This is, however,"

page 14, line 21: "tripolar grid in 1° and 2° nominal resolution." should read "tripolar grids at either 1° or 2° nominal resolutions."

page 15, line 2: "2000 db" should read "2000 dbar"

page 15, line 2: the way the results are reported, "potential densities" should read "potential density anomalies"

page 15, line 24: "present date" should read "present-day" or "the present"

page 17, line 18: "the data set provided by" can be deleted

page 17, line 23: "At 1 January" should read "On 1st January"

page 18, line 12: "have a too strong" should read "have too strong a"

page 18, line 25: "biased low" should read "biased towards too low values"

page 18, line 27: "[. . . ] whereas the North Pacific is biased cold and fresh [. . . ]" would better read "[. . . ] whereas in the North Pacific there is a cold and fresh bias [. . . ]"

page 20, lines 10–11: "estimated as" should read "estimated at"

page 20, line 15: "for the period" should read "from"

page 22, line 24: "ecosystem" should read "ecosystem model"

page 24, lines 10–11: "[. . . ] has been reported for other models earlier [. . . ]" should read "[. . . ] has previously been reported for other models [. . . ]"

page 25, line 20: "produce" should read "release"

page 28, line 1: "at depths deeper than" should read "at depths greater than"

page 29, line 24: "during" would better read "from"

page 29, line 25: "over" would better read "from"

page 30, line 5: "over 1990–2000" would better read "from 1990 to 2000"

page 31, line 20: "Since already the PP [. . . ] for Lv1.2 is at the low end [. . . ]" should read "Since the PP [. . . ] for Lv1.2 is at the low end already [. . . ]"

page 32, line 16: please provide publication year for Seiter et al.

page 33, line 4: "based on" should read "from"

page 33, line 13: "in terms of" should read "with regard to"

page 33, line 29: "smaller than" should read "lower than" or "below"

page 34, line 18: "more recycling" should read "more intense recycling" or "stronger recycling"

page 34, line 19: "smaller values for export efficiency" should read "lower export efficiencies"

page 34, line 24: "the scheme simulates also" should read "the scheme also leads to"

page 35, lines 7 and 9: the correlation cannot be of 0.32 or 0.22, the correlation *coefficient* can

page 35, line 20: "[. . . ] NorESM-OC1 corresponds to the fully coupled [. . . ]" should read "[. . . ] NorESM-OC1 corresponds to the version included in the fully coupled [. . . ]"

page 35, line 21: "[. . . ] in CMIP5.": please provide a reference to one or two relevant CMIP5 papers.

page 35, line 24: "grid" should read "grids"

page 35, line 24: "in high latitudes" should read "at high latitudes"

page 36, line 24: "[. . . ] a reduced ocean circulation (reduced AMOC) [. . . ] " should read "[. . . ] a more sluggish ocean circulation (weaker AMOC) [. . . ]"

page 37, line 11: "which still is" should read "which is still"

---

## Author Comment (AC1) · 7 Jun 2016

**Authors' response to an anonymous review (reviewer #1) of "Evaluation of NorESM-OC (versions 1 and 1.2), the ocean carbon-cycle stand-alone configuration of the Norwegian Earth System Model (NorESM1)" by Schwinger et al.**

We thank the reviewer for reviewing our manuscript and for his/her constructive and helpful comments. Our detailed response to the points raised is given below (reviewer's comments in italic font, our response in normal font).

My only major comment would be that, while intercomparing the various versions of the model, the manuscript does nothing to contextualise the performance of NorESM-OC within the context of other Earth system models. The CMIP5 archive is something of a treasure trove on this point, and most of the evaluations made in the manuscript could be repeated with output from it. However, I would only suggest adding an overview of this (e.g. Taylor diagrams?), and not amending the manuscript throughout the main task here, and I would agree with the authors on this, is to evaluate performance between traceable versions of NorESM-OC.

We see the reviewers point that it would be interesting for the reader to see how NorESM-OC performs relative to similar models. We therefore propose to add a figure showing the carbon uptake of our model compared to the models used in Le Quéré et al. (2015). We believe that this would satisfactorily meet the reviewer's concern because i) estimation of contemporary C-uptake is and has been an application of major importance for the stand-alone configuration of our model, ii) C-uptake provides a kind of integrated measure since it depends on several aspects of model performance, iii) the models used in Le Quéré et al. (2015) are truly comparable to NorESM-OC also in terms of atmospheric forcing (all these models are forced by reanalysis data). We have added this as Fig. 21 of the revised manuscript.

Pg. 2, ln. 4: "We present simulation results  $\ldots$ " this sentence could perhaps be a little clearer on how many configurations are examined; the mention of multiple resolutions makes it a little opaque

We rephrased this sentence as follows. "We present simulation results of three different model configurations (two different model versions at different grid resolutions) using two different atmospheric forcing data sets."

**Pg. 3, ln. 12: The authors simulate the ocean component of an ESM under reanalysis forcing; did they consider running under atmospheric output from the ESM instead?**

We did not consider this kind of experiments in this study, but it is certainly a very useful application of stand-alone configurations. We added a short sentence on this in the revised manuscript: "Idealised experiments include set-ups where (purposefully manipulated) atmospheric output from fully coupled ESM runs is used to force the stand-alone configuration of the same or another model."

Pg. 8, ln. 1: "We anticipate ... of 10,000 years ..." would I be correct in assuming this "anticipation" is either too computationally expensive to test, or is not actually possible because v1.2 of the model rolls in several changes and not just this one (i.e. it is not possible to separate differences due to the time-stepping change from those from other sources)?

We did tests (1000 years with the low resolution configuration) comparing model runs that differed only in the time-stepping method, and we found that the simulation results were not significantly different. Nevertheless, the global correction that accounts for the non-conservation of tracer mass due to the time-smoothing error in version 1 leads to a small but continuous unphysical re-distribution of tracer mass. By extrapolating these differences we come to the "anticipation" that for time scales of 10,000 years the fully consistent time stepping in version 1.2 would be relevant. The reviewer is correct in the assumption that this would be rather computationally expensive to test by actually running the model for 10,000 years (about a wall-clock year of run time with the low resolution configuration).

Pg. 8, ln. 13: "derived by Wanninkhof (1992)" it might be worth examining Wanninkhof (2014) for updates here; I believe that some of the issues mentioned subsequently are tackled there

Yes, the updated formulation of Wanninkhof (2014) includes a re-fit of the Schmidt-Numbers avoiding the mentioned problems at high SSTs. We decided to adopt the Gröger and Mikolajewicz (2011) formulation to keep our HAMOCC version close to the one used by the Hamburg-group (Ilyina et al., 2013). We mention the updated formulation of Wanninkhof (2014) in the revised manuscript as follows: "We note that the updated air-sea gas exchange formulation provided by Wanninkhof (2014) also includes re-fitted Schmidt numbers, avoiding problems at high SST."

Pg. 10, ln. 15: A remark about the shape of the nutrient limitation relationship would be useful; also something about the relationship between nutrients (e.g. Leibigs Law or something else?)

We added this information as follows. "The nutrient limitation is expressed through a Monod function  $f_{\text{nut}} = X/(K+X)$ , where K is the half-saturation constant for nutrient uptake ( $K = 4 \times 10^{-8}$  kmol P m-3), and

$$X = \min\left(\mathrm{PO}_4, \frac{\mathrm{NO}_3}{R_{\mathrm{N:P}}}, \frac{\mathrm{Fe}}{R_{\mathrm{Fe:P}}}\right).$$

 $R_{\text{N:P}} = 16$  and  $R_{\text{Fe:P}} = 366 \times 10^{-6}$  are the (constant) nitrogen to phosphorus and iron to phosphorus ratios for organic matter used in the model."

Pg. 10, ln. 23: Presumably this means that opal "production" is associated with export only, and is not representative of all of the opal production by diatoms (some of which is dissolved before it can be exported)?

Yes, this is correct. We clarified this by adding the word "export". "The distribution of

calcium carbonate and biogenic silica export production depends on the availability of silicic acid ..."

**Pg. 11, ln. 4: "we performed a re-tuning" by eye?**

We did not apply an objective optimization procedure for the model tuning. Data assimilation schemes are being developed for our biogeochemistry module, but are not yet available. Instead, our model tuning was based on our experience (as to which model parameter could be tuned to achieve a certain effect) and a long series of trial and error simulations to finally arrive at the (mostly) improvements presented in this manuscript. So, yes, the tuning was partly performed "by eye", partly based on statistical measures (biases, correlations, etc. as presented in the manuscript). We clarified this by adding the following sentence. "The tuning was not based on an objective procedure, but relied on the authors' experience and a series of "trial and error" simulations."

Pg. 11, ln. 16: Is there a potential problem here because Si:P ratios in the real world are not constant?; in iron-limited HNLC regions, for instance, Si:P is typically elevated (since diatom cell cycle length is increased affording more time to uptake silicon); the global nature of the "solution" here to DIC/TA biases potentially promises trouble regionally

Indeed, we find that the surface Si distribution is less well reproduced than that of phosphate. In the revised manuscript we added a few sentences on the possible shortcomings in the parameterisation of the Si cycle in connection with elevated Si concentrations in the North Pacific. "We note that the pattern of elevated silicate concentrations in the North Pacific and Arctic is similar to the pattern of  $PO_4$  concentration in the model. While this is in good agreement with observations for phosphate, observed maximum Si concentrations are 50% smaller than modeled maximum concentrations. This might indicate that our ecosystem model is not tuned well enough or that its structure is oversimplified with respect to silica cycling (e.g., fixed Si:P ratio, fixed constant sinking speed)." Although the fixed Si:P is a limitation of the model, we show that surface silica concentrations compare better to observations in model version 1.2, and that DIC/TA biases are significantly reduced by tuning of the fixed Si:P ratio.

Pg. 14, ln. 16: It would perhaps be helpful to illustrate to the reader the depth distribution of sinking particles, as well as remineralisation, produced by these different schemes (e.g. for the same quantity of POC at 100m, whats its fate down a static water column?)

If we understand the reviewer correctly, he/she asks for the general properties of the different schemes. However, in the case of KR02, the sinking speed depends on the available mass and numbers of particles as well as the strength of aggregation, which all is variable in space and time. Kriest and Oschlies (2008) present a general analysis and comparison of several schemes (among them KR02, STD, and Martin curve, which is mostly equivalent to WLIN). In the revised manuscript we add a panel to Fig. 23 showing the normalised POC flux with depth for the simulations presented in the manuscript. We also add a short sentence referring the reader to Kriest and Oschlies (2008). "For a more

detailed discussion of the KR02 scheme in comparison to the assumption of constant or linearly increasing sinking speed, we refer the reader to Kriest and Oschlies (2008)."

*Pg.* 16, *ln.* 10: "... a restoring time scale of 365 days (version 1) and 350 days (version 1.2) ... " any reason why this is 15 days shorter?; presumably a parameterisation oversight?

There was a typo in our original manuscript. The relaxation time scale stated for model version 1.2 should read 300 instead of 350 days. The somewhat shorter relaxation time scale applied in version 1.2 was introduced to approximately counterbalance the weakening of the relaxation flux due to the balancing option. In this context we also found an error in the following sentence describing the balancing option. Correcting these errors and adding the information asked by the reviewer, this paragraph now reads: "In order to stabilise the model solution we apply salinity relaxation towards observed surface salinity with a restoring time scale of 365 days (version 1) and 300 days (version 1.2) for a 50 m thick surface layer. The restoring is applied as a salt flux which is also present below sea ice. In model version 1.2 balancing of the global salinity relaxation flux was added as an option, which allows to keep the global mean salinity constant over long integration times. That is, positive (negative) relaxation fluxes (where "positive" means a salt flux into the ocean) are decreased (increased) by a multiplicative factor if the global total of the relaxation flux is positive (negative). The somewhat shorter relaxation time scale applied in version 1.2 was chosen to approximately counteract the weakening of the relaxation flux due to this balancing procedure."

Pg. 17, ln. 13: "with only small trends of 0.00007, 0.021, and 0.048 Pg Cyr-1 century-1 for Mv1, Mv1.2, and 15 Lv1.2, respectively" any explanation for why the longer duration simulations with v1.2 have markedly higher CO2 trends than the shorter duration v1 model?; one might instinctively expect smaller values for longer simulations; perhaps plot up the net CO2 flux with time?

The explanation for this is a smaller decadal to centennial internal variability in the Mv1 compared to Mv1.2 and Lv1.2, which we attribute to the details of the salinity relaxation. First, we apply a reduced salinity relaxation south of 40 degree in Mv1.2 and Lv1.2, and second, the balancing of the salinity restoring flux (introduced in model version 1.2 leads to a generally more saline surface ocean in Mv1.2 and Lv1.2, i.e. a less stable total water column. This can be clearly seen in Fig. 1 of this response, where we plot the last 200 years of the spin-up runs for the three configurations. Nevertheless, a trend of the order of 0.05 Pg C yr-1 century-1 is still rather small compared to an anthropogenic change in oceanic C inventory of the order of 100 Pg C over less than a century. Further, our ocean sink estimates are calculated relative to a control run, i.e. any offset and trend in the model is taken into account. We added the following text to the revised manuscript in order to better clarify this (also addressing some questions raised by reviewer #2).

"The relatively large uptake of carbon in Mv1 at the end of the spin-up is due to a lower  $CaCO_3$  to POC production ratio found in this configuration. A full equilibration

of the model with respect to this process would require a considerably longer spin-up. The re-tuning of the ecosystem parameterisation, which increases  $CaCO_3$  production in version 1.2, leads to carbon fluxes much closer to zero at the end of the spin-up period. The larger trends in Mv1.2 and Lv1.2 compared to Mv1 despite longer spin-up time is due to larger decadal to centennial scale internal variability in these configurations (i.e., the systematic long term drift is much smaller). We attribute this to the details of the salinity relaxation (balancing of the restoring flux, weaker relaxation south of  $40^{\circ}$  S). We finally note that even these larger trends are tiny compared to the changes in ocean carbon uptake due to anthropogenic carbon emissions, and that we calculate our estimates of anthropogenic carbon uptake relative to a control run to account for offsets and trends due to a not fully equilibrated model."

Pg. 18, In. 8: "Since the unbalanced salinity relaxation flux removes salt ..." - this sentence reads as if it is saying that the reduction in S in Mv1 is due to the relaxation flux being applied across \*all\* model configurations; I think this sentence and the preceding one should be combined into: "While all three configurations include salinity relaxation, this is not balanced in the case of Mv1, with the result that average salinity falls by 0.2 psu during the course of the integration."

Thank you for this suggestion, this has been adopted.

Pg. 18, ln. 14-16: Worth reporting these sorts of numbers in a table?

In the revised manuscript we added two tables. The first one reporting the total numbers for PP and export productions (Table 4), and a table summarising C-uptake and storage for the three model versions/configurations (Table 5).

Pg. 19, ln. 12: "...a long transient increase in strength for about 300 years ..." Any chance of including a plot of the AMOC strengths of the models from their spin-up phases?; not least to give some idea of interannual variability in the absence of interannual forcing variability (if any)

Pg. 20, ln. 15: Since the plot makes a point of examining the time-series of AMOC, would it be possible to present the RAPID estimates on the same plot?

We followed these two suggestions by (i) extending the plot of AMOC strength to include also the part of the historical time period that was forced with CORE normal year forcing (1762–1947), and (ii) by including the RAPID estimate for 2004–2014 (McCarthy et al., 2015) in the plot.

*Pg.* 20, *ln.* 19: "... the climatology of de Boyer Montégut et al. (2004)" This is calculated how?

Pg. 20, ln. 22: On a related point, are model MLDs comparable to the climatology?; e.g. could MLD be calculated for the model in the same way as its done for the observation-based climatology?; if this is already the case, please make this clear

De Boyer Montégut et al. (2004) provide three climatologies using three different threshold criteria, a temperature criterion (0.2 degree), a density criterion (0.03 kg m-3), and a variable density threshold criterion (corresponding to a 0.2 degree decrease in temperature at local temperature and salinity conditions). Our model uses a bulk mixed layer formulation, that is, the mixed layer extent is calculated based on energy considerations, and the mixed layer is not vertically resolved (except for an extra top model layer of 5 to 10 m thickness). Due to the lack of resolution it is not possible to use threshold criteria applied to model output to obtain an equivalent definition of MLD as used in observation based estimates. The best we can do is to compare to a climatology that uses a density criterion, since this is conceptually the most similar definition. In the original manuscript we compared our model results to the mean and the range of the three de Boyer Montégut et al. (2004) climatologies which is (as mentioned also by the reviewer in a comment further down) not the best way to represent uncertainty. For the revised manuscript we propose to compare modeled MLD to the density criterion MLD only. We revised Fig. 4 and the manuscript text accordingly.

"Seasonal cycles of modeled average bulk mixed layer depth (MLD) compared to an observation based MLD climatology (de Boyer Montégut et al., 2004) are shown for several regions in Fig. 4. The climatology uses a threshold criterion for density, that is, MLD is defined by the depth where density has increased by 0.03 kg m-3 relative to its near surface value. We note that the depth of the bulk mixed layer in our model is calculated based on energy gain and dissipation in the surface ocean, and that modelled and observed quantities are therefore not directly comparable. A MLD climatology based on a density criterion is nevertheless suitable for comparison with our model, since the criterion measures stratification directly."

**Pg. 21, In. 12: "clearly indicates a too deep mixing" Grammar**

We rephrased this sentence as follows. "...indicates a deeper than observed mixing compared to CFC-11 profiles from the GLODAP data base."

**Pg. 21, ln. 13-16: Does this have anything to do with sea-ice?; some models can exhibit large polynas in the SO, with the result that mixing, and ventilation, can be extreme**

We do not find large polynyas in the model runs presented here. We clarify this by adding the following sentence to the revised manuscript. "These high concentrations are mainly caused by large fluxes occurring during Antarctic winter north of the ice edge."

Pg. 22, ln. 5-7: "We note that the Eppley-VGPM algorithm produces global PP estimates at about the mean value ..." This might well be true, but in my experience the spatial patterns of different estimates are wildly different, making the choice of such an "intermediate" product less clear

We deleted this sentence from the revised manuscript. As suggest by the reviewer in a comment further down, we have revised our comparison of PP with observations. We now use the average of three different products, namely VGPM, Eppley-VGPM, and CbPM (Behrenfeld and Falkowski, 1997; Westberry et al., 2008) provided by the Ocean Productivity website http://www.science.oregonstate.edu/ocean.productivity/. We plot

the mean and range of these estimates in Figs. 6, 7, and 8. Section 3.2 and the Appendix have been revised accordingly.

Pg. 22, ln. 12-14: "These large discrepancies are reduced in model version  $1.2 \ldots$ " Is it possible (e.g. via run models that are not shown here) to be sure that the improvements stem from the BGC changes as opposed to the physics changes?

Yes, we have done such tests, which show that the changes in the BGC module are indeed the cause for the (by far largest portion) of the differences in PP between model version 1 and 1.2. (see Fig. 2 of this response). We add this information to the revised manuscript as follows. "Sensitivity tests with the same physical model but both versions of the biogeochemistry module (not shown) indicate that changes in the physical fields between model versions do not contribute significantly to this result."

Pg. 23, ln. 5-6: "This missing PP on the shelves ..." You could make this clearer by calculating PP in open ocean areas only for VGPM and the model runs

Thank you for this suggestion, this has been done. We have excluded data from shelf regions in our analysis in the revised manuscript throughout Section 3.2 and in Figures 6, 7, and 8.

Pg. 23, ln. 18-26: Is it possible to determine a map of nutrient limitations from the model?; it might help diagnose another reason for differences between them

This is possible, see Fig. 3 of this response for the configuration Lv1.2. Broadly in agreement with observations nitrate is the limiting nutrient at low latitudes. Iron is the limiting nutrient in the southernmost Southern ocean, in the northernmost North Pacific and parts of the Arctic Ocean in the model. Our model has no iron limitation in the equatorial Pacific and the North Atlantic. Compared to results from nutrient fertilisation experiments compiled by Moore et al. (2013) the iron limited regions in our model appear to be either too small (Southern Ocean, North Pacific) or absent (equatorial Pacific, North Atlantic). Moreover, even if we remove iron limitation completely, model results do not change significantly, that is, in those regions that are iron limited, the limitation is not very strong. Therefore, we do not find that differences in the seasonal cycle of PP and PP patterns can be explained by differences in nutrient limitation. The major differences between the model versions stem from the re-tuning of the ecosystem parameterisation as pointed out in our response above (see also Fig. 2 of this response).

Pg. 24, section 3.3: BGC tracers are the core of the model, while production is just one process within the model; Id suggest swapping the sections around and making this 3.2; production could come just before export - which is arguably more natural anyway

We agree with the reviewer that the order "tracer-production-export" would be more natural from a conceptual viewpoint. However, we describe differences between model versions and configurations here, and the differences in PP is at the origin of much of the differences in the tracer distribution between Mv1, Mv1.2 and Lv1.2. We refer back to the results of section 3.2 frequently in section 3.3, and we think that the manuscript is much easier to understand if organised in this order. We therefore would prefer to leave the order of sections unchanged.

**Pg.* 24, section 3.3: Why no chlorophyll?; is this because the fixed chl:C ratio here causes problems?**

We do not analyse chlorophyll here because it is not of major importance in our NPZDecosystem model. The chlorophyll concentration (calculated by the fixed C:CHl ratio mentioned by the reviewer) is only used in the model to calculate light absorption through phytoplankton. A more realistic variable C:Chl ratio could possibly improve the simulation of light availability, but there is a couple of improvements we think will have a larger impact (e.g. improved vertical resolution in the mixed layer which is currently rather low).

Pg. 25, ln. 16: "Moreover, nitrogen fixation, ..." – A map of this perhaps?

Pg. 25, ln. 17: "... which occurs in the surface ocean as soon as  $[NO_3] < R_{N:P}[PO_4]$ , ..." Does this mean that all N2-fixation occurs in the right place?; cf. ostensible temperature limits, etc.

We added a map (Fig. 13 of the revised manuscript) showing the difference  $[PO_4]$  –  $[NO_3]/16$  and rewrote this paragraph as follows. "The modelled distributions of nitrate and phosphate are similar in terms of biases and general spatial structure since the model uses a fixed stoichiometric ratio  $(R_{\rm N;P} = 16)$  for the composition of organic matter. The difference  $[PO_4] - [NO_3]/R_{N,P}$  (Fig. 13) is positive everywhere in broad agreement with observations, indicating that nitrate is depleted relative to phosphate with respect to the canonical N:P ratio of 16. Large values of this difference are found in the tropical Pacific, in the model as well as in observations, but more pronounced in the model. This pattern is due to the tropical Pacific oxygen minimum zone (OMZ) where NO3 is consumed during denitrification to remineralise organic matter and release  $PO_4$ . The oxygen minimum zones are excessively large in our model, particularly in the tropical Pacific (see Sec. 3.3.2). These results show, that the simple parameterisation of nitrogen fixation, which occurs in the surface ocean as soon as  $[NO_3] < R_{N,P}[PO_4]$ in our model, is active over the whole surface ocean, which is probably unrealistic. This simple parameterisation should be viewed more as a means to keep the model ocean close to the assumed stoichiometric N:P ratio than a realistic parameterisation of nitrogen fixation. At depth, we find major deviations from the similarity of nitrate and phosphate distributions in the OMZ of the tropical Pacific. Here, our model shows a local minimum of nitrate (due to the too strong denitrification caused by too low oxygen values) instead of a local maximum as observed and as it is found for phosphate."

*Pg.* 26, *ln.* 2-6: What do the distributions of biogenic opal and CaCO3 export look like?; and how do they compare to observationally-derived estimates (e.g. in total)?

We have added maps of POC export production as well as maps of  $CaCO_3$  to organic carbon and opal to organic carbon in exported matter (Fig. 9 of the revised manuscript).

We also added a new table summarising the key numbers (PP, exports) for all model versions. Section 3.2 has been expanded by a new subsection which reads as follows.

**"3.2.1 Export production of POC, CaCO3, and opal**

Figure 9 shows the mean export production of POC, as well as the  $CaCO_3$  and opal to organic carbon ratios in exported matter. The spatial pattern of POC export closely resembles the pattern of PP, since the fraction of PP that is exported as POC (about 15–25%, not shown) shows only small spatial variations. Total annual carbon export (averaged over 2003 to 2012, Table 4) is 8.8 (Mv1), 7.1 (Mv1.2), and  $6.1 \text{ Pg C yr}^{-1}$  (Lv1.2).

Since the partitioning between opal and CaCO3 export production is parameterised dependent on available silicate in our model, CaCO3 export dominates over opal export only in regions depleted of surface silicate, which are the subtropical gyres in the Atlantic and the south Pacific. Due to the increased opal to phosphorus uptake ratio  $R_{Si:P}$ in model version 1.2 (clearly visible in Fig. 9 g to i) the CaCO3 production is maintained or even slightly expanded into the western North Pacific despite the much lower PP (and surface silicate consumption) at high latitudes in this model version. We note that the simple parameterisation of opal and CaCO3 export production is qualitatively supported by opal to particulate inorganic carbon (PIC) ratios derived from sediment traps. For example, Honjo et al. (2008, see their Fig. 7) show that high opal/PIC ratios are constrained to ocean regions with high surface silica concentrations, a pattern that is qualitatively reproduced by our model.

Total modelled opal export (between 95 and 120.5 Tmol Si yr-1, Table 4) is within the uncertainty range of the estimate of  $105\pm17$  Tmol Si yr-1 given by Tréguer and De La Rocha (2013). The ratio of CaCO3 export to organic carbon export of 7.1 to 7.9 % is within the range estimated by Sarmiento et al. (2002,  $6\pm3\%$ )."

Pg. 27, ln. 15: While iron isnt quite at the stage of having a global observational climatology, Geotraces has some fields that might help; and even in the absence of an observational comparison, it might be helpful to compare the models to elucidate differences; irons reach is longer than simply total iron concentration

The iron cycle in our model is rather simplistic. The spatial pattern of iron in the surface mainly reflects the aeolian input and some upwelling of iron in the Souhtern Ocean. The maximum iron concentration is determined by relaxation towards a value of 0.6  $\mu$ mol m-3 (when modelled iron is larger than this value) to mimic the process of complexation with ligands. Therefore, below 700 m, the iron concentration is constant at 0.6  $\mu$ mol m-3 everywhere. As mentioned above, the iron limitation in our model is not very strong, and the parameterisation of the iron cycling is definitely something to be improved in the future. For the current model version presented here, we do not think that it makes much sense to discuss the iron cycle in detail.

Rather, we propose to amend the model description as follows. "A fraction of the dust deposition (1%) is assumed to be iron, and part of it is immediately dissolved and available for biological production. To mimic the process of complexation with ligands,

iron concentration is relaxed towards a value of 0.6  $\mu$ mol m-3 when modelled iron is larger than this value. We note that this parameterisation of the iron cycle is rather simplistic. The spatial pattern of iron in the surface mainly reflects the aeolian input and upwelling of iron in the Southern Ocean. At depth, iron concentration is determined by accumulation of remineralised iron and the assumed complexation. Therefore, the iron concentration approaches a constant value of 0.6  $\mu$ mol m-3 at depth larger than approximately 700 m. We find that the resulting iron limitation in our model is rather weak, and given these limitations we will not focus on the iron cycle in this manuscript." In the conclusions, we also mention the iron cycle (together with nitrogen fixation) as parameterisations to be improved in the future. "There are several directions for future model development that can be identified from our results. The parameterisations of iron cycling and nitrogen fixation are simplistic and should be improved for future model versions, particularly since the observational basis for both processes has improved considerably in recent years."

Pg. 28, ln. 22: "... results in increased CaCO3 production and a considerable reduction of the alkalinity and DIC biases ..." Can this be squared with any observational evidence?; it can certainly be squared with model evidence (cf. Kwiatkowski et al., 2014; here, a number of models have low CaCO3 production in the tropics and excessive alkalinity and DIC)

The simple parameterisation used in HAMOCC to set opal and  $CaCO_3$  content of exported material is qualitatively supported by opal to PIC ratios derived from sediment traps (e.g. Honjo et al., 2008, Fig. 7). This information has been added to the new section on export production (Sec. 3.2.1 of the revised manuscript, see above).

We have no direct observational support for improved  $CaCO_3$  export in model version 1.2, but rather use the indirect evidence of improved surface alkalinity. We propose to amend the manuscript as follows.

"Since modelled alkalinity in low latitudes is quite sensitive to  $CaCO_3$  export (see also Kwiatkowski et al., 2014), we take the reduction of alkalinity biases as indirect evidence for an improved distribution of  $CaCO_3$  export in Mv1.2 and Lv1.2."

Pg. 30, ln. 25: "... the GLODAP data base" The Khatiwala et al. (2013) estimate of anthropogenic  $CO_2$  is probably a better estimate

Yes, we agree. We added a comparison of the Khatiwala et al. (2013) estimate with our model results as follows. "Compared to a recent synthesis of anthropogenic carbon storage estimates (Khatiwala et al., 2013,  $155 \pm 31 \text{ Pg C}$  for the year 2010) modelled DICant storage of 186 (Mv1), 175 (Mv1.2), and 159 (Lv1.2) Pg C is higher, but still within the uncertainty range of the synthesis estimate."

Pg. 31, ln. 16: As mentioned previously, having a figure that illustrates how each of these schemes remineralises organic matter down the water column would also be helpful (or a plot of how the OM is attenuated)

In the revised manuscript we added a new panel to Fig. 23 showing the average POC fluxes (normalised to the POC flux at 100 m depth). "The global average sinking speed profiles for the four experiments and the resulting POC fluxes (normalised to the flux at 100 m depth) are shown in Fig. 23a and b, respectively."

Pg. 31, ln. 28: "At the end of the spin-up runs ..." How do these fit with the long spin-ups already done?; also, are these long enough to approach equilibrium?; or is the assumption that they are long enough for only transient drift to remain?

These are not additional spin-up runs. Each model configuration is spun-up for 1000 years. For the comparison of the different POC sinking schemes we use data from the years 1001–1010. The model is generally not in full equilibrium after 1000 years but there is only a small drift remaining. We clarify this by rewording the sentence as follows. "At the end of the spin-up runs (years 1001–1010) we find .... As mentioned above, the model is generally not in full equilibrium after 1000 years, but the remaining drift is small."

Pg. 33-34: I dont know the answer myself, so its perhaps cheeky to ask, but could the authors comment on whether direct POC flux measurements or indirect AOU (or other tracer) measurements better constrain export and remineralisation; there may be no good answer at the moment, so the authors use of both is probably best

We added a short discussion on the advantages/disadvantages of using direct/indirect methods to estimate POC-fluxes as follows. "We have based the evaluation of the different POC sinking schemes on indirect ( $PO_4^{remin}$ ) and direct (sediment trap) measurements. While the indirect method has inherent inaccuracies related to the calculation of AOU and the assumed stoichiometry of remineralisation, the direct measurement of POC fluxes by sediment traps also comes with large systematic uncertainties (see Honjo et al., 2008, and references therein). It is difficult to decide whether one of the two methods provides a more reliable evaluation and we therefore use both approaches here."

Pg. 35, ln. 16: A more general comment some studies (e.g. Kwon et al., 2009; Kriest, Oschlies & Khatiwala, 2012) examine the tuning of such models of export, whereas the manuscript uses them as is; while the authors do mention alternative sinking velocities for STD-fast at one point, they could help here by drawing further attention to this and / or commenting on the tuning of such models (e.g. if they have any unreported experience on the success or otherwise of this)

We do not have any further experience with tuning the POC sinking scheme for our model. The computational constraints do not allow for much more than the runs presented in this study. We added a comment on this as follows. "We finally note that a comprehensive sensitivity analysis or a rigorous tuning of the different POC sinking schemes would require accelerated off-line integration techniques as applied by e.g. Kwon et al. (2009) or Kriest et al. (2012), which we to date have not available for our model."

Pg. 36, In. 10: So, paradoxically, excessively high and excessively low O2?

Yes, exactly. We made this a bit clearer by rephrasing this sentence as follows. "In strong contrast to these positive  $O_2$  biases, the model develops too large oxygen minimum zones along with a too strong accumulation of remineralised phosphate in the tropical oceans, particularly in the Pacific."

Pg. 37, ln. 18: "Part of this problem is the distribution of primary production which is too high in a narrow band along the upwelling ..." Is this in any way related to the model being an isopycnal model?

We do not have any evidence that this could be related to fact that the model is isopycnic. There is a CORE intercomparison paper on Pacific circulation in preparation (Mats Bentsen, personal communication), and our model (as well as the other isopycnal models) do perform as good (or bad) as other models in the equatorial Pacific. Since this work is at an early stage however, we do not have a citable reference for this statement.

Pg. 38, ln. 1: "In the Southern Ocean  $\ldots$ " Since this paragraph deals with the model as it is, rather than - per the preceding paragraph - the model as it might become, it should precede the paragraph on shelf improvements

This paragraph deals with the problems that need to be addressed in the Southern Ocean to improve our model, and as such it also deals with the model as it might become. Since the paragraph concludes with some rather general statements, we would prefer to leave it at this position.

Pg. 38, ln. 9: It's a wholly personal preference, but I think papers finish better with a short, bulletpointed list of the main points / findings

Since this comment is flagged as a personal preference, we hope that it is allowed to respectfully disagree here. We have tried to write a concise 'summary and conclusions' section, which is clearly structured as we believe. We do not immediately see the point in repeating our main findings as a bullet-point list, and would therefore prefer to leave the end of the manuscript unchanged.

Pg. 38, ln. 9: A general criticism I'd make of the manuscript's validation of the model is that the performance of the model is not properly put within the context of similar models; the CMIP5 archive, for instance, offers a range of similar resolution models that could profitably provide such context; most of which arent isopycnal models

Please see our response the general comment at the start of this document.

Table 1: " $kmol m^{-3}$ " At the risk of both being a pedant and missing the wood for the trees, presumably  $kmol/m^3$  is being used here because it is equivalent to molar units (i.e. mol/l); if so, why not just use mol/l?

For some reason kmol m-3 is the "traditional" HAMOCC-unit. It is used in the model descriptions Maier-Reimer et al. (2005) and Ilyina et al. (2013), and therefore we think it might be reasonable to stick to this unit in this manuscript, too.

Table 1: "Laughing gas" While - appropriately enough - I laughed when I saw this, I dont think it can be called this in the final manuscript; nitrous oxide, perhaps?

This has been corrected.

Table 2: "Fraction of grazing egested  $1 - \varepsilon_{zoo}$ " This is a little bit confusing; is the symbol really " $1 - \varepsilon_{zoo}$ "?; why not " $\varepsilon_{zoo}$ " and give it a value of 0.2 or 0.1?

Again, this is a traditional HAMOCC formulation. It has been used by Six and Maier-Reimer (1996), who implemented the ecosystem parameterisation and later on by Maier-Reimer et al. (2005) and Ilyina et al. (2013). We therefore propose to stick to this way of expressing this parameter.

Figure 1: Sometimes putting the y-axis on a logarithmic scale is helpful for showing whats happening near-surface

We changed the vertical scale of the zonal mean plots throughout the manuscript to enlarge the upper 1000 m. In the revised plots the 0-1000 m range occupies about 1/3 of the vertical scale. This solution avoids potential problems of a log scale while still emphasizing the fields at the surface.

Figure 1: Since the colour map includes white, these panels could do with having the seafloor drawn on; that would help separate places that have zero difference from those that are rock; also, this would help clarify the bathymetry differences between different model grids

The background colour of the zonal mean plots has been changed to grey throughout the manuscript, such that sea-floor can be distinguished from regions with zero differences (white colour).

Figure 3: What happens with AMOC during the long spin-ups? Figure 3: " $10^9 \text{ kg s}^{-1}$ " Convert to Sverdrups?; or is this awkward for an isopycnal model?

Figure 3: Observational data from RAPID appropriate for comparison?

For the revised manuscript, we updated Figure 3. We converted units to Sverdrup and an observational estimate from RAPID has been included. Also, we have extended the plot to include the years 1762–1947, which are forced by the CORE normal year forcing. We mention in the main text that during the spin-up "the AMOC shows a long transient increase in strength for about 300 years before stabilising".

Figure 4: "The range given for the observation based estimates is solely due to different criteria used to define MLD and not due to other uncertainties" Per a previous comment, how does the MLD method used for the models compare to that of the observations?; also, given how variable different MLD methods can be from one another, reporting uncertainty in this way here seems potentially risky

Please see our detailed response to the previous comment above. We have revised the

figure and compare our model to one climatology only. This climatology is based on a density criterion, which is conceptually most similar to the MLDs calculated by our model. The sentence about the range of observation based estimates has been deleted from the revised manuscript.

Figure 5: Again, a log-scale y-axis might help here; most of the structure here is in the upper water column

Figure 5: Rotate panel e so that its y-axis is aligned with the x-axes on the panels to the left?; i.e. 90S to the left, 90N to the right

The vertical scale of this figure has been changed as explained above. Panel e has been rotated as requested.

Figure 6: Its a weakness on my part, but I prefer my plots to omit unnecessary grid lines (and have coastlines if possible)

Figure 6: Rotation would make panel e easier to understand (though I appreciate it would not then be aligned as in Figure 5)

We added coast lines to all 2d-plots throughout the revised manuscript. We also rotated panel e as requested. Since we do find the grid lines very helpful to read and interpret this figure, we propose to leave them in the revised plots.

Figure 6: Rather than only use VGPM, you could average VGPM with other estimates; its a poor way of simplifying the diversity in observational estimates of PP, but it can be useful given their spread, and its not without precedent

As explained above, we consider three estimates obtained by different algorithms in the revised manuscript. We plot the mean of the three estimates in panel d), and and the mean and range of zonal means in panel e).

Figure 7: Why 40 in one hemisphere and 60 in the other?

Expressed in PgC yr-1, the production north of 60N is tiny, and the difference between model versions and observations is small. To some degree this is also true in the southern hemisphere, but there is a notable difference between model version 1 and 1.2, so we split the southern hemisphere into one additional subregions. For the northern hemisphere there is no added value in doing so, therefore we propose to leave Fig. 7 unchanged in this respect.

Figure 7: Again, why just use VGPM?

See our answer above. We include the mean and range of three different observational estimates in the revised figure.

Figure 8: In panel c, including the Indian Ocean is complicated by the presence of the monsoon

In the revised manuscript we change Figure 8c to show the tropical Indian Ocean, and

revised the manuscript text as follows. "In the tropics outside the Indian Ocean there is no significant annual cycle in the model as well as in observations. The seasonal variation in PP caused by the monsoon in the Indian Ocean is well captured by the model although there appears to be a small phase shift of about one month relative to the MODIS based PP estimates (Fig. 8c)."

**Figure 9: Rotate panel e again please**

This has been done.

Figure 10: Fewer colours in the colour maps here (especially for the delta plots) might make it easier to discern patterns in match-mismatch (e.g. the reds are quite homogeneous)

In the revised version of the manuscript we reduced the number of colours in the colourmap to 16 (or 32 for difference plots) throughout the manuscript.

Figure 12: These panels hint at some odd ventilation feature that elevates N. Pac. silicic acid (which then bleeds into the Arctic); does the same appear in CFC-11?

CFC-11 does not show a suspicious ventilation feature here. Phosphate shows similar elevated surface values in the North Pacific and Arctic, but for phosphate this is also seen in the observations. We conclude that this might be a problem of our ecosystem parameterisation (fixed stoichiometric ratios, fixed constant sinking speed and/or remineralisation rate of opal, to name a few possible sources). To discuss this, we amended the manuscript text as follows. "We note that the pattern of elevated Si values in the North Pacific and Arctic is similar to the pattern of PO4 concentration in the model. While this is in good agreement with observations for phosphate, observed maximum Si concentrations are 50% smaller than modelled maximum concentrations. This might indicate that our ecosystem model is not tuned well enough or that its structure is oversimplified with respect to silica cycling (e.g., fixed Si:P ratio, fixed constant sinking speed)."

Figure 13: Add a key if possible; also, different symbols might help with the plot (especially for colour blind readers)

This has been done.

Figure 13: Amend to Panel (d) shows results for the 500m depth level with error prone grid points located in the tropics (between 20 and 20S) omitted from the analysis?

This has been done.

Figure 14: Why not go entirely east-west in the Pacific here?; since oxygen is particularly low in the East Pacific, this could be important

We followed this suggestion and plotted a zonal mean  $O_2$  concentration averaged over the entire eastern Pacific. This does not change the general features of the model-observation

comparison (it makes our model look a bit worse though).

Figure 16: Nicely improved!

Thank you.

Figure 17: Uncertainty from Takahashi (and / or other pCO2 products)?

We included  $pCO_2$  from the Takahashi et al. (2009) data product in the plot.

Figure 18: There's quite a bit of a dip in the observationally-derived fluxes; is its origin explained in the main text?

Yes, this is explained in the main text. The dip in the data was worth a Science publication. "From the early 1990s until 2001 the Landschützer et al. (2015) data shows a marked decline followed by a steep increase of carbon uptake, which they attribute to a saturation followed by a reinvigoration of the Southern Ocean carbon sink."

Figure 19: averaged over the years 1990 to 1998 Why not a full decade?; being unreasonably suspicious, such odd ranges always raise my eyebrows

Actually, the time averaging is unnecessary here anyway. Anthropogenic carbon as a time integrated quantity does not need any averaging to arrive at a climatology. The nominal year for the GLODAP anthropogenic carbon product is 1994. We therefore plotted the corresponding model values for 1994 in the revised plot.

Figure 20: The mismatched sizes of bars shown in panel b seem a bad idea; log scale again?

We added a second y-axis on the right hand side of the revised plot which has an appropriate range for the fluxes at 2000 m and at the bottom.

Figure 21: Why zero in the Arctic of panels a, c, e and g?; does the main text say? Figure 21: A global integrated profile plot of a, c, e and g might be helpful

We included a figure with global mean profiles of remineralised  $PO_4$ , and added an explanation for the zero remineralised  $PO_4$  in the Arctic as follows. "The absence of significant amounts of remineralised phosphate in the Arctic basin in our model (Fig. 23) is due to a combination of low PP and too strong ventilation (positive  $O_2$  biases, compare Fig. 16)."

Figure 22: Bigger dots?; also, a key would be nice; perhaps a little map that shows the regions in the appropriate colour?

It is difficult to use much bigger dots, since then they would overlap too much. We increased the size of dots a bit for the revised figure. A map showing the colors of each region has been added.

**Figures**

Figure 1: Air-sea  $CO_2$  flux during the last 200 years of the spin up runs for Mv1 (blue line) Mv1.2 (green line), and Lv1.2 (red line). The dotted lines give a linear fit through the respective time series. Note that for Mv1 the spin-up was only 800 years, and the actual years displayed are 600–800.

---

## Author Comment (AC2) · 7 Jun 2016

**Authors' response to an anonymous review (reviewer #2) of "Evaluation of NorESM-OC (versions 1 and 1.2), the ocean carbon-cycle stand-alone configuration of the Norwegian Earth System Model (NorESM1)" by Schwinger et al.**

We thank the reviewer for reviewing our manuscript and for his/her constructive and helpful comments. Our detailed response to the points raised is given below (reviewer's comments in italic font, our response in normal font).

*The paper falls well within the scope of Geoscientific Model Development. It is overall well structured, although in detail, information can not always be found where one would expect to find it (some parameter values are only reported in the discussion section, instead of the model description part). The English language is generally good. There remain a few minor shortcomings that can nevertheless be easily fixed.*

In the revised manuscript all parameter values are defined in the model description part.

*The presentation and description of the model is rather complete, the discussion of the results is to some extent merciless: the authors put much emphasis on the biases in their model results. I would be pleased to read a few more sentences about the strengths of their model.*

We added a few sentences on this at the beginning of the model description section. "The main benefit of an isopycnal model is the good control on the diapycnal mixing (less numerical diffusion across isopycnic surfaces) that helps to preserve water masses during long model integrations. This is of particular interest for a coupled physical-biogeochemical model, since sharp gradients in tracer concentrations between water masses (e.g. in the thermocline region) can potentially be better represented by the model. The isopycnic framework provides a terrain-following vertical coordinate, and therefore overflow of water masses is modelled without numerical obstacles (although mixing at steep slopes must be parameterised carefully)."

*General model presentation: scope*
*The model is clearly global in scope. Time frames for applications are not clearly stated. The paper presents simulation experiments over a few centuries, following 800 or 1000 year spin-ups. Can it also be reasonably used on longer time-scales (a few thousands to a few tens of thousands of years)?*

We added this information in the introduction as follows: "Although NorESM-OC is computationally less expensive than the fully coupled NorESM, computational constraints limit the time-scale, for which the model can be applied, to a few hundred to thousand years on current hardware. We note, that this limitation is mainly due to the physical ocean model and the costly transport of tracers. By using an efficient offline

method (e.g., Khatiwala et al., 2005; Kriest et al., 2012) it would be possible to apply HAMOCC on time-scales at least one order of magnitude longer, as long as all relevant external inputs are provided (e.g., fluxes due to continental weathering). Similar versions of HAMOCC have been applied on time-scales of 50 000 years (e.g., Heinze et al., 2003, who use an offline setup with annually averaged ocean circulation fields)."

*Code: availablility and quality*

*The code is not openly available, but only upon signing licence agreements (at least two, one for NorESM, one for HAMOCC, but perhaps even more – this is not clear). As a reviewer, I was nevertheless granted access to the code. It is written in Fortran (much FORTRAN77-style, with some Fortran 90 elements). A C-style pre-processor is required to compile it. Some files are well commented, others almost devoid of comments. The presence of OpenMP compiler directives suggests that it has been prepared for usage on multi-platform shared memory multiprocessing computing facilities, which is definitely a noteworthy feature of interest to prospective users.*

*Unfortunately, the manuscript does not provide much information about the code, nor about numerical methods adopted in general (except for time-stepping, which seems to follow a leap-frog scheme in combination with a filter with unspecified characteristics though). These are, however, the informations that are typically expected in a Development and technical paper in Geoscientific Model Development. I would recommend to add a short, descriptive paragraph about such aspects, the more since the code is not publicly available. Finally, it would also be interesting to read about typical execution times of the three versions.*

As recommended we added a subsection on technical aspects at the end of the model description section, which reads:

"**2.5 Technical aspects**

The HAMOCC code was originally written in FORTRAN77. It was later re-written to take advantage of FORTRAN90 elements (e.g. ALLOCATE statements) and to conform to FORTRAN90 free source code format. Certain model options, for example the selection of a POC sinking scheme, are implemented via C-preprocessor directives. HAMOCC should compile on any platform that provides a FORTRAN compiler and a C-preprocessor.

MICOM is parallelised by dividing the global ocean domain horizontally into (logically) rectangular tiles which are processed on one processor core each. Communication between cores is implemented using the Message Passing Interface (MPI) standard. Since HAMOCC is integrated into MICOM as a subroutine call it inherits this parallelism. Note that HAMOCC, in addition, has been parallelised for shared memory systems using the OpenMP standard. This feature is, however, not tested and not supported for the current set-up. For the simulations presented in this manuscript, the ocean component has been run on 190 (Mv1), 309 (Mv1.2), and 155 (Lv1.2) cores on a Cray XE6-200 system, yielding a model throughput of 11 (Mv1), 20 (Mv1.2), and 64 (Lv1.2) simulated model years per (wall-clock) day."

We also better specified the characteristics of the time-smoothing applied with the leap-frog time stepping in MICOM: "To prevent excitation of numerical noise, the MICOM leap frog time-stepping includes a time-smoothing applied to the mid time-level of temperature, salinity and layer thickness fields at each time step, i.e. $x_{\mathrm{mid}} = w_1 x_{\mathrm{mid}} + w_2(x_{\mathrm{old}} + x_{\mathrm{new}})$ with $w_1 = 0.875$ and $w_2 = 0.0625$. Since the physical fields undergo this time smoothing, but the biogeochemical tracers do not, …".

In the "Code availability" section we clarified the number of licenses that is required: "…availability of the code is subject to signing two license agreements — one for the use of NorESM, and, additionally, for the use of HAMOCC signing of the MPI-ESM license agreement is required. …"

**Specific comments:**

*page 4, line 19: "[…] NorESM-OC1 corresponds to the fully coupled NorESM1-ME [...]" would better read "[...] NorESM-OC1 corresponds to the version included in the fully coupled NorESM1-ME [...]"*

This has been adopted.

*Although the paper reads as if NorESM-OC1 was a sub-model isolated from NorESM1, its setup is highly similar to the earlier isopycnic ocean carbon cycle model of Assmann et al. (2009). How does NorESM-OC1 actually differ from the model described by Assmann et al. (2009)?*

We added information on this in Sec. 2.3 as follows: "The model described by Assmann et al. (2010) was the starting point for the development of the ocean biogeochemistry component of NorESM1. Main differences between this model and NorESM-OC1 are updates of mixed layer and eddy diffusivity parameterisations, the use of CICE as ice model, and the details of atmospheric forcing, which followed Bentsen and Drange (2000), wheras NorESM-OC uses the approach of Large and Yeager (2004). The version of HAMOCC employed by Assmann et al. (2010) is very similar to the one in NorESM-OC1 with the notable difference of an update of the carbon chemistry scheme (see below)."

*page 7, lines 14–17: I presume that the preformed tracer values are set to the mixed-layer concentrations each time that a water parcel leaves the surface mixed-layer. Please rephrase for clarity.*

Yes, the preformed tracers are set to the tracer concentration in the mixed layer at each time step. We clarified this by adding "at each time step".

*page 8, line 26 – page 9, line 7: the detailed approximation adopted for TA is given, but it would also be interesting to know how the speciation of the acid-base systems is calculated, or, equivalently, how pH is calculated from the TA expression.*

We added this information as follows: "We use an iterative carbonate alkalinity correction method (Follows et al., 2006; Munhoven, 2013) to calculate the oceanic partial pressure of $CO_2$ ($pCO_2$) prognostically as a function of temperature, salinity, dissolved inorganic carbon (DIC), and TA. Using an initial guess for the hydrogen ion concentration $[H^+]_0$, all non-carbonate contributions to TA are calculated according to Dickson et al. (2007) and added to or subtracted from (1) in order to obtain an initial guess for carbonate alkalinity $CA_0$ ($CA = [HCO_3^-] + 2[CO_3^{2-}]$). Given CA and DIC and the equilibrium constants of the carbonate system $K_1$ and $K_2$, the corresponding hydrogen ion concentration can be calculated:

$$[H^+] = \frac{K_1}{2\,CA}\left(DIC - CA + \sqrt{(DIC - CA)^2 + 4\,CA\,K_2/K_1(2\,DIC - CA)}\right). \quad (2)$$

The hydrogen ion concentration $[H^+]_1$ thus obtained is used to re-iterate these calculations until convergence is reached. Here, we use $[H^+]$ from the previous time step as an initial guess for the calculation of CA, and stop iterations once the relative change $([H^+]_{i+1} - [H^+]_i)/[H^+]_{i+1})$ becomes smaller than $\epsilon = 5 \times 10^{-5}$. Only at the first time step of integration we use $[H^+]_0 = 10^{-8}$ mol kg$^{-1}$. As pointed out by Follows et al. (2006), the changes in $[H^+]$ from one time step to the next are small, and one or a few iterations of this procedure usually suffice."

*page 12, lines 6–8: I may possibly misinterpret this, but I am wondering how the loss of material (and alkalinity) through the sediment is compensated for, if the model does not take into account the influx of carbon, alkalinity and nutrients? Does it operate in some kind of closed-loop configuration (mass an alkalinity that are lost via the deep-sea sediment are re-injected at the top)? Please clarify how the global inventories of the model ocean are conserved.*

There is no compensation for the mass and alkalinity lost to the deep-sea sediment in the model versions presented here, which limits the time-scale of applicability. We clarified this by adding: "The material lost to the sediments is therefore not replaced by some mechanism in the model configurations presented here. The model drift caused by these losses to the sediment is small on the time-scales considered in this manuscript, particularly at the surface. Nevertheless, this simplification limits the applicability of NorESM-OC1 and 1.2 to time-scales of the order of 1000 years. Input of DIC and nutrients in particulate and dissolved forms through rivers is currently implemented into NorESM."

*page 13, lines 12–13: while I find it rather obvious how M can be a prognostic variable, I do not immediately see how NOS can be handled as such. Please give a few details about how NOS is predicted.*

We rewrote this paragraph to accommodate some more details on how NOS is treated as follows. "NOS is treated like other particulate tracers in the model, i.e. NOS is advected and diffused by ocean circulation and treated in HAMOCC's sinking scheme (see below) using the average sinking speed for particle numbers as given in Kriest (2002).

Additionally, aggregation of particles decreases NOS, while photosynthesis, egestion of fecal pellets, and zooplankton mortality increase NOS. The size distribution is affected by these process as follows. Sinking removes preferentially the large particles and leaves behind the smaller ones, thereby steepening the slope $\varepsilon$ of the size spectrum in surface layers. Aggregation "flattens" the slope of the size spectrum, because it reduces the number of particles, but not mass. These processes are parameterised as in scenario "pSAM" of Kriest (2002),...We assume that all other biogeochemical processes do not impact $\varepsilon$ (e.g., photosynthesis increases $M$ and NOS proportionally, such that the slope of the size distribution is not affected), the exception being zooplankton mortality, which flattens the size spectrum through the addition of (large) zooplankton carcasses."

*page 15, beginning of section 2.4.1: I recommend to introduce this section by one or two general sentences that summarize which data are required to force the model, before coming to the specific versions of datasets used.*

We followed this recommendation by adding an introductory sentence to this section: "The atmospheric forcing required to run NorESM-OC comprises temperature, specific humidity and wind at 10 m reference height as well as sea level pressure, precipitation and incident short wave radiation fields."

*page 16, line 8: Only the fate of the Antarctic freshwater influx is detailed. How is the rest of the freshwater treated?*

We clarified this by adding: "All runoff fluxes are mapped to the ocean grid and smeared out over ocean grid cells within 300 km of each discharge point to account for unresolved mixing processes."

*page 16, line 8: 365 days for v1 and 350 days for v1.2: why this difference?*

This was also asked by the first reviewer, and we repeat our response to this question here: There was a typo in our original manuscript. The relaxation time scale stated for model version 1.2 should read 300 instead of 350 days. The somewhat shorter relaxation time scale applied in version 1.2 was introduced to approximately counterbalance the weakening of the relaxation flux due to the balancing option. In this context we also found an error in the following sentence describing the balancing option. Correcting these errors and adding the information asked by the reviewer, this paragraph now reads: "In order to stabilise the model solution, we apply salinity relaxation towards observed surface salinity with a restoring time scale of 365 days (version 1) and 300 days (version 1.2) for a 50 m thick surface layer. The restoring is applied as a salt flux which is also present below sea ice. In model version 1.2 balancing of the global salinity relaxation flux was added as an option, which allows to keep the global mean salinity constant over long integration times. That is, positive (negative) relaxation fluxes (where "positive" means a salt flux into the ocean) are decreased (increased) by a multiplicative factor if the global total of the relaxation flux is positive (negative). The somewhat shorter relaxation time scale applied in version 1.2 was chosen to approximately counteract the weakening of the relaxation flux due to this balancing procedure."

*page 17, line 8: is atmospheric pCO2 prescribed or is it prognostic?*

It is prescribed. We clarified this by rephrasing "...using the CORE normal year forcing and a prescribed atmospheric $CO_2$ concentration of 278 ppm...".

*page 17, lines 13–14: the carbon flux of 0.26 GtC/yr from Mv1 is about 50 times larger than the fluxes for the other two and not truly negligible (it is of about the same order of magnitude as the atmospheric $CO_2$ consumption rate by continental weathering). Is this correct, and if so, what is the reason? Furthermore, the drifts of Mv1.2 and L1.2 are of the order of 5–10%, which is far from negligible over 100 yr. This may point out significant deviation from the sought near-equilibrium state.*

The carbon-flux in Mv1 is due to the too low $CaCO_3$ production, and the time-scale at which the model would equilibrate with regard to this process (under prescribed atmospheric $CO_2$ particularly) is probably of the order of 100000 years. The negligible carbon-uptake in model version 1.2 is a result of the re-tuning of $CaCO_3$ production. If we had implemented riverine inputs to the ocean we would expect to have an outgassing of a few tenths of Pg carbon per year if $CaCO_3$ production was reasonably tuned. The larger drift in Mv1.2 and Lv1.2 is caused by a larger decadal to centenial scale internal variability in these configuration (see response to the first reviewer, Fig. 1). Nevertheless, a trend of the order of 0.05 Pg C century$^{-1}$ is still rather small compared to an anthropogenic change in oceanic C inventory of the order of 100 Pg C over less than a century. Further, our ocean sink estimates are calculated relative to a control run, i.e. any offset and trend in the model is taken into account. We added the following text to the revised manuscript in order to better clarify this.

"The relatively large uptake of carbon in Mv1 at the end of the spin-up is due to a lower $CaCO_3$ to POC production ratio found in this configuration. A full equilibration of the model with respect to this process would require a considerably longer spin-up. The re-tuning of the ecosystem parameterisation, which increases $CaCO_3$ production in version 1.2, leads to carbon fluxes much closer to zero at the end of the spin-up period. The larger trends in Mv1.2 and Lv1.2 compared to Mv1 despite longer spin-up time is due to larger decadal to centennial scale internal variability in these configurations (i.e., the systematic long term drift is much smaller). We attribute this to the details of the salinity relaxation (balancing of the restoring flux, weaker relaxation south of $40°$ S). We finally note that even these larger trends are tiny compared to the changes in ocean carbon uptake due to anthropogenic carbon emissions, and that we calculate our estimates of anthropogenic carbon uptake relative to a control run to account for offsets and trends due to a not fully equilibrated model."

*page 19, lines 19–23: "[...] if this trend would be removed.": is it possible to remove it? Where does it actually come from?*

We agree that this formulation was a bit unclear. The reason for this trend is discussed further down in the text. We rephrased this sentence as follows. "Compared to the other model simulations the annual and decadal scale variability of AMOC strength appears

to be similar, but superimposed there is a negative trend of $8\,\mathrm{Sv}$ over the simulation period (see below)."

Further down we add a sentence to make it clearer that the discussed peculiarity of the salinity relaxation scheme when the balancing of the relaxation flux is activated leads to this negative trend: "This effect is particularly pronounced in the Lv1.2 configuration leading to the negative AMOC trend of $8\,\mathrm{Sv}$ described above."

*page 27, line 26: I find these biases rather strong (50% or so). Comments?*

It is true that the oxygen biases in the model are rather strong. As described in the text, this is due to model deficiencies in the Southern Ocean and in the oxygen minimum zones in combination with the relatively long spin-up time. Part of the problem is also the too low C transport into the deep ocean when the standard sinking is used. In fact, a reduction of the global average $O_2$ bias below 3000 m of 40% is attained with the KR02 and WLIN sinking schemes, as mentioned in Sec. 3.5. Finally, the situation is probably not improved by the fact that the model is isopycnic, since the strong gradients between different water masses with opposite biases are not alleviated by numerical diffusion. We rewrite this paragraph to add these points: "We note that part of the deep ocean oxygen bias is connected to low POC transport into the deep ocean when the standard sinking scheme is used. The large positive bias below 3000 m depth is reduced by 40% in Lv1.2 when using alternative sinking schemes. This is further discussed in Sect. 3.5."

And further down: "We finally speculate that the strong $O_2$ biases of opposite sign in adjacent water masses are probably more pronounced in our isopycnic model than they would be in an $z$-coordinate model, since the strong $O_2$ gradients between these water masses are not alleviated by numerical diffusion."

*page 32, lines 8, 25 and 27: please provide references for the cited numbers (they can be found in the figure captions, but it would be good to have them in the text as well).*

This has been done, each cited number is provided with the respective citation in the revised manuscript.

*page 36, lines 23–29: It is fairly possible that the new sinking parametrisations are simply more efficient in counterbalancing the effects of the (too?) strong Southern Ocean ventilation. Using a model biogeochemical process to reduce biases arising from short-comings in the model physics can hardly be considered an improvement.*

We have multiple lines of evidence for "improvement" based on direct and indirect observations. The direct indication of "improvement" comes from observation based estimates of POC fluxes (Figs. 23 and 26 in the revised manuscript). For example, we show that modeled POC fluxes and sediment trap data compare much better with the KR02 and WLIN sinking schemes. This is independent of any process in the physical model. Even if one has limited confidence in sediment trap derived fluxes because of methodological difficulties, Figs. 22a and b (original manuscript) clearly demonstrate that the STD-slow scheme has a much too low POC flux through 2000 m.

Oxygen biases in the SO and the deep water masses are rather strong as mentioned by the reviewer above, and our results indicate that there are two factors that contribute. First, the too strong ventilation which is also indicated by CFC fields as well as cold and fresh biases in the SO. Second, if the standard sinking scheme is used, a contribution of too little remineralisation in the deep ocean. In Lv1.2 the global average bias below 3000 m is reduced from 0.1 to 0.06 mol m$^3$ with the KR02 and WLIN schemes. Reductions in $O_2$ biases are also seen in regions that are not ventilated by the SO, and where we think that the physical model performs reasonably well (e.g. North Atlantic Deep Waters). Given the evidence from direct POC flux observations, we think that it is justified to consider the reduction of $O_2$ biases an improvement.

*page 37, line 11: "[...] a long standing problem [...]": please provide a reference to support for this statement.*

This has been done. We cite the original work of Najjar et al. (1992) and the more recent work of Dietze and Loeptien (2013).

**Technical comments**

*Throughout the paper: please consistently use either "parameterisation/parameterise" or "parametrisation/parametrise" as a spelling*

All occurrences of "parametrisation/parametrise" have been replaced by "parameterisation/parameterise" in the revised manuscript.

*page 2, line 14: "[...] scheme prescribing a linear increase of sinking speed [...]" would better read "[...] scheme that uses a linear increase of the sinking speed [...]"*

This has been adopted.

*page 3, lines 15–24: please indicate which versions of CESM, CAM-Oslo, MICOM and HAMOCC are used, respectively.*

This has been done except for MICOM, since there are no official version numbers attached to the MICOM developed in Bergen.

*page 3, line 28: "persue" should read "pursue"*

This has been corrected.

*page 4, lines 23–24: It is at this point not entirely clear what "on a numerically more efficient grid in 1° and 2° resolution" means. Based upon what we read later on, this should actually read "on a numerically more efficient grid at either 1° or and 2° resolution". Also: are these actual or nominal resolutions? If they are nominal, what are the extremes?*

We clarified this by rephrasing "...is configured on a numerically more efficient grid at either 1° or 2° nominal resolution...".

We propose to add more detailed information on the grid spacing in Sec. 2.4 as follows:

"We discuss three different model grid configurations, one for NorESM-OC1, which runs on a displaced pole grid with 1.125° nominal resolution and with grid singularities over Antarctica and Greenland. NorESM-OC1.2 has been set up on a numerically more efficient tripolar grid in 1 and 2° nominal resolution. The tripolar grid has its singularities at the South Pole, in Canada, and in Siberia. The nominal resolutions given here indicate the zonal resolution of the grid while the latitudinal resolution is finer and variable. For the displaced pole grid of NorESM-OC1, the latitudinal grid spacing is 0.27° at the equator gradually increasing to 0.54° at high southern latitudes. The tripolar grid of NorESM-OC1.2 is optimised for isotropy of the grid at high latitudes, and the latitudinal spacing varies from 0.25° (0.5°) at the equator to 0.17° (0.35°) at high southern latitudes for the 1° (2°) nominal resolution. Note that over the northern hemisphere both grid types are distorted to accommodate the displaced pole over Greenland or the dual pole structure over Canada and Siberia."

*page 7, sect. 2.3: which version of HAMOCC is finally used? 5 or 5.1? Please check and make sure the text is consistent.*

We clarified in the text that HAMOCC5.1 is used.

*page 7, line 26: "seriously" would better read "significantly"*
*page 7, line 27: "[. . . ] the tracer transport fully consistent [. . . ]" should read "[. . . ] the tracer transport scheme to make it fully consistent [. . . ]"*

Thank you, these suggestions has been followed.

**References**

Assmann, K. M., Bentsen, M., Segschneider, J., and Heinze, C.: An isopycnic ocean carbon cycle model, Geosci. Model Dev., 3, 143–167, doi:10.5194/gmd-3-143-2010, 2010.

Bentsen, M. and Drange, H.: Parameterizing surface fluxes in ocean models using the NCEP/NCAR reanalysis data, RegClim, Regional Climate Development Under Global Warming, General Technical Report no. 4, pp. 149–157, 2000.

Dickson, A., Sabine, C., and Christian, J.: Guide to best practices for ocean $CO_2$ measurements, PICES special publication 3, North Pacific Marine Science Organization (PICES), Sidney, British Columbia, Canada, 2007.

Dietze, H. and Loeptien, U.: Revisiting "nutrient trapping" in global coupled biogeochemical ocean circulation models, Glob. Biogeochem. Cyc., 27, 265–284, doi: 10.1002/gbc.20029, 2013.

Follows, M. J., Ito, T., and Dutkiewicz, S.: On the solution of the carbonate chemistry system in ocean biogeochemistry models, Ocean Modelling, 12, 290–301, 2006.

Heinze, C., Hupe, A., Maier-Reimer, E., Dittert, N., , and Ragueneau, O.: Sensitivity of the marine biospheric Si cycle for biogeochemical parameter variations, Glob. Biogeochem. Cyc., 17, 1086, doi:10.1029/2002GB001943, 2003.

Khatiwala, S., Visbeck, M., and Cane, M. A.: Accelerated simulation of passive tracers in ocean circulation models, Ocean Modelling, 9, 51–69, 2005.

Kriest, I.: Different parameterizations of marine snow in a 1-D model and their influence on representation of marine snow, nitrogen budget and sedimentation, Deep-Sea Res. I, 49, 2133–2162, 2002.

Kriest, I., Oschlies, A., and Khatiwala, S.: Sensitivity analysis of simple global marine biogeochemical models, Global Biogeochem. Cycles, 26, GB2029, doi:10.1029/2011GB004072, 2012.

Large, W. and Yeager, S.: Diurnal to decadal global forcing for ocean and sea-ice models: The data sets and flux climatologies, Tech. Note NCAR/TN-460+STR, National Center of Atmospheric Research, Boulder, Colorado, 2004.

Munhoven, G.: Mathematics of the total alkalinity–pH equation – pathway to robust and universal solution algorithms: the SolveSAPHE package v1.0.1, Geosci. Model Dev., 6, 1367–1388, doi:10.5194/gmd-6-1367-2013, 2013.

Najjar, R. G., Sarmiento, J. L., and Toggweiler, J. R.: Downward transport and fate of organic matter in the ocean: Simulations with a general circulation model, Glob. Biogeochem. Cyc., 6, 45–76, 1992.

---

## Author Response (AR2)

**Technical corrections made to the manuscript "Evaluation of NorESM-OC (versions 1 and 1.2), the ocean carbon-cycle stand-alone configuration of the Norwegian Earth System Model (NorESM1)" by Schwinger et al.**

We thank the editor for evaluating the revised version of our manuscript. All technical corrections requested by the editor have been adressed as detailed below (editor's comments in italic font, our response in normal font).

We also took the oportunity to correct a few additional minor errors in our manuscript as listed at the end of this response.

*p. 8, ll. 172-173: "To prevent excitation of numerical noise": I suggest to replace "excitation" by "amplification"*

This has been done.

*p. 10, ll. 236-237: ". . . the ratios of carbon to phosphorus and nitrogen to phosphorus in organic matter are RC:P = 122 and RN:P = 16, respectively, and RO2:P = 172 moles of O2 . . . " [grammar and suppression of a few words]*

Thank you, this has been adopted.

*p. 10, ll. 239-240: "by Smith" shoudl read "of Smith" or "from Smith"*

This has been corrected.

*p. 12, l. 281: ". . . export poductions depend on . . . " [grammar]*

This has been corrected.

*p. 13, l. 302: ". . . DIC biases . . . "*

This has been corrected.

*p. 14, l. 311: a few words to explain why 115 moles?*

We rephrased this sentence as follows. ". . . it is assumed that $2/3 \times R_{O_2 : P} \approx$ 115 moles of nitrate are consumed by denitrifying bacteria to remineralise an amount of detritus corresponding to one mole of phosphate (i.e., it is assumed that the oxygen from two moles of nitrate substitutes three moles of oxygen during denitrification)."

*p. 20, ll. 511ff: as specified in the introduction to the standard definition of Fortran 90 (ISO/IEC 1539 : 1991 (E)), names are spelled "FORTRAN 77" and "Fortran 90" (yes, differently since the '90 standard)*

All occurences of "FORTAN77" and "FORTRAN90" have been replaced by

"FORTRAN 77" and "Fortran 90", respectively.

*p. 20, ll. 312-313: "... to conform to the Fortran 90 free source code format certain model options, such as, for example, ...*

This has been adopted.

*p. 21, l. 534: there is no such thing as "psu". Salinity is a dimensionless quantity, and as such does not have any "unit" (see, e.g., UNESCO Technical Paper 45 on The International System of Units (SI) in Oceanography). Please discard and check troughout.*

All occurences of "psu" have been removed from the manuscript.

*p. 23, ll. 571-572: "... but superimposed onto a negative trend of 8 Sv ... "*

This has been adopted.

*p. 23, l. 583: "... of the restoring salt flux ... "*
*p. 23, l. 588: "... that the AMOC strength ... "*
*p. 23, l. 591: "... overestimates the AMOC strength ... "*
*p. 23, l. 593: "... period from 2004 to 2012."*

These errors have been corrected.

*p. 25, l. 640: "Sectroradiometer" should read "Spectroradiometer"*

This has been corrected.

*p. 25, l. 642: "Carbon based Production Model"*

This has been corrected.

*p. 26, l. 663: "At low latitudes ... "*

This has been corrected.

*p. 33, l. 862: "... alkalinity at low latitudes ... "*

This has been corrected.

*p. 40, l. 1070: "whether one ... provides a more reliable ... " should read "which one ... provides the most reliable ... "*
*p. 40, l. 1072 "... techniques such as that applied by ... " or "... techniques such as those applied by ... "*

This has been adopted.

*p. 42, l. 1117: "... which one of the new ... "*
*p. 42, l. 1121: "... that the sinking speed ... "*

This has been corrected.

*p. 43, l. 1143: "...will also offers ..." should read "...will also offer ..."*
This has been corrected.

*p. 44, l. 1170: "...shelf when the ..." should read "...shelf where the ..."*
This has been corrected.

*p. 63, 3rd line of the caption: "retrivals" should read "retrievals"*
This has been corrected.

*Figures in general: please enlarge the graphics as much as possible. Some of them are barely readable (e.g., Figs. 9, 10, 14, 15, 17 and others) in the current manuscript version. Please check throughout.*
We increased the font size in Figs. 5,6,9,10,14,15,17,18,19,22,23,26. We also provide Fig. 26 in a better quality (higher resolution).

Further corrections(line and page numbers refer to the updated manuscript):

p. 6, l. 2: "provides a terrain following coordinate" corrected to "provides a smooth representation of bottom topography".

p. 10, l. 15: "HAMOCC5" replaced by "HAMOCC5.1"

p. 14, l. 13: "through" replaced by "by"

p. 16, l. 11: added "at layer $i$".

p. 22, l. 27-28: added "left panel" and "right panel"

p. 27, l. 14-15: moved "Fig. 8d" into the parentheses in the following sentence.

p. 34, l. 12: added "into the ocean"

Table 5, caption: replaced "last but one" by "second to last"

Fig. 7, caption: removed "different".

Fig. 8, caption: "(c) tropics" corrected to "(c) tropical Indian Ocean"

Figs. 12, 24, 25, caption: replaced "phosphorus to oxygen ratio" by "oxygen to phosphorus ratio"